EMBO
reports

# Genotoxic stress triggers Scd6-dependent regulation of translation to modulate the DNA damage response

Gayatri Mohanan [1], Raju Roy[1,4], Hélène Malka-Mahieu[2,3], Swati Lamba[1], Lucilla Fabbri[2,3], Sidhant Kalia[2,3], Anusmita Biswas[1], Sylvain Martineau [2,3], Céline M Labbé[2,3], Stéphan Vagner [2,3✉] & Purusharth I Rajyaguru [1✉]

## Abstract

**The role of mRNA translation and decay in the genotoxic stress response remains poorly explored. Here, we identify the role of yeast RGG motif-containing RNA binding protein Scd6 and its human ortholog LSM14A in genotoxic stress response. Scd6 localizes to cytoplasmic puncta upon cell treatment with various genotoxic agents. Scd6 genetically interacts with SRS2, a DNA helicase with an anti-recombination role in DNA damage repair under HU stress. Scd6 directly interacts with the SRS2 mRNA to repress its translation in cytoplasmic granules upon HU stress in an eIF4G1-independent manner. Scd6-SRS2 interaction is modulated by arginine methylation and the LSm-domain of Scd6, which acts as a cis-regulator of Scd6 arginine methylation. LSM14A regulates the translation of mRNAs encoding key NHEJ (Non-homologous end-joining) proteins such as RTEL1 (SRS2 functional homolog) and LIG4. NHEJ activity in yeast and mammalian cells is regulated by Scd6 and LSM14A, respectively. Overall, this report unveils the role of RNA binding proteins in regulating the translation of specific mRNAs coding for DNA damage response proteins upon genotoxic stress.**

**Keywords** DNA Damage; Genotoxic Stress; mRNA Translation; RNA Binding Protein; RNA Granules
**Subject Categories** DNA Replication, Recombination & Repair; Signal Transduction; Translation & Protein Quality

## Introduction

DNA damage caused by cellular conditions and/or exogenous factors leads to genomic instability. Replication errors, defective DNA damage repair, and nucleotide misincorporation represent cellular events leading to DNA damage (Marnett, 2001). Exogenous factors causing DNA damage include exposure to genotoxins such as hydroxyurea, cisplatin, etc. Living systems employ strategies to regulate gene expression and adapt the cellular proteome to mount an effective genotoxic stress response (GSR). Even if the contribution of transcription to GSR is overall well documented, and the role of post-translational modifications in eliciting an effective response, changes in the fate of cytoplasmic mRNAs in response to GSR remain poorly explored (Mohanan et al, 2021).

A variety of genotoxic insults such as methyl methanesulfonate (MMS), ultraviolet (UV) irradiation, hydroxyurea (HU), cisplatin (CSP), and zeocin have been used to study genotoxic stress response. All these genotoxic stresses are known to induce double-strand breaks through various mechanisms. MMS is an alkylating agent known to modify DNA leading to replication fork stalling. UV leads to the formation of pyrimidine dimers and DNA oxidation. HU inhibits ribonucleotide reductase, leading to dNTP pool depletion and fork arrest in replication. Cisplatin forms platinum-based adducts on DNA, whereas zeocin binds and intercalates between the DNA strands leading to both single and double-strand DNA lesions (Ma et al, 2011; Koç et al, 2004; Rastogi et al, 2010; Azab et al, 2019; Chen and Stubbe, 2005). Specific changes in mRNA translation and decay induced by these stresses that alter proteome diversity and contribute to genotoxic stress response are poorly characterized. RNA-binding proteins (RBPs) play a crucial role in determining the functional states of mRNAs. Several classes of RNA binding domains, such as RNA Recognition Motif (RRM), K homology (KH) domain, Zinc-finger motif and PUMILIO, have been reported to play a role in post-transcriptional gene expression by changing the fate of specific subsets of mRNAs. RGG motif-containing proteins are the second largest class of RBPs (Thandapani et al, 2013; Chowdhury and Jin, 2022). RGG-motifs are characterized by RGG-/RGX repeats that impart properties of low-complexity sequences (LCS) (Thandapani et al, 2013). Consistently, these sequences contribute towards the assembly of higher-order ribonucleoprotein (RNP) condensates (i.e., RNA granules) by undergoing liquid-liquid phase separation (LLPS) (Chong et al, 2018). In yeast, processing bodies (P-bodies or PB) and stress granules (SG) are the major cytoplasmic mRNPs formed in response to several physiological cues, which contain several RNAs and RBPs. PBs are reported to be the sites of mRNA decay, but recent reports also indicate their role in mRNA storage (Wang et al, 2018; Standart and Weil, 2018). SGs are mainly implicated in

[1]Department of Biochemistry, Indian Institute of Science, Bangalore 560012, India. [2]Institut Curie, PSL Research University, CNRS UMR3348, INSERM U1278, F-91405 Orsay, France. [3]Université Paris Sud, Université Paris-Saclay, CNRS UMR3348, INSERM U1278, F-91405 Orsay, France. [4]Present address: University of Pennsylvania, Philadelphia, PA, USA. ✉E-mail: stephan.vagner@curie.fr; rajyaguru@iisc.ac.in

mRNA storage and repression (Buchan and Parker, 2009). *S. cerevisiae* Scd6 is an RBP with RGG-motif sequences that target eIF4G1 to repress translation (Rajyaguru et al, 2012). Scd6 is a known P-body and stress granule resident protein in response to glucose deprivation and oxidative stress (Bhatter et al, 2019). The RGG motif of this protein is important for localization to RNA granules such as P bodies, interaction with eIF4G1, and consequent translation repression activity (Poornima et al, 2016). LSM14A, the human ortholog of Scd6, is a granule-resident protein that has also been implicated in translational control (Yang et al, 2006), however, the translation targets of either Scd6 or LSM14A are unknown. LSM14A contains two RGG motifs as compared to a single RGG motif in Scd6 (Roy and Rajyaguru, 2018). LSM14A plays a crucial role in forming the mRNA silencing complex via its association with DDX6 (Brandmann et al, 2018). Although xRAP55 (Xenopus ortholog of LSM14/Scd6) has been reported to repress translation (Tanaka et al, 2006), the direct role of human LSM14A in translational repression remains to be demonstrated. Interestingly, LSM14A has been reported as a sensor of viral nucleic acid, which plays a crucial role in antiviral response (Li et al, 2012).

The role of LCS-containing proteins in GSR remains poorly explored. Even though the localization of Scd6 to puncta upon hydroxyurea (HU) stress is known (Tkach et al, 2012), the functional relevance of localization to granules, the overall role of the Scd6 family of proteins with RGG motif (if any) and their specific mRNA targets in genotoxic stress response remain unexplored. In this report, we identify a general role of Scd6 in response to several genotoxic stresses. We further tease out the contribution of Scd6/LSM14A in regulating the translation of specific mRNAs (*SRS2, RTEL1*, and *LIG4*) following HU-mediated genotoxic stress.

SRS2 (Suppressor of Rad Six) is a conserved DNA helicase with DNA-dependent ATPase activity implicated in anti-recombination function and NHEJ pathway (Hegde, 2000; Marini and Krejci, 2010). Cells devoid of SRS2 are sensitive to double-strand break-inducing agents such as HU, MMS, ultraviolet light, ionizing radiations and zeocin (Chvadarova et al, 2015; Dhingra et al, 2021; Friedl et al, 2001). RTEL1 (Regulator of Tumor Elongation Helicase 1) is a functional homolog of SRS2 that protects telomere ends by interacting with shelterin complex protein Trf1 (Sarek et al, 2015). RTEL1 like SRS2 has been reported to have anti-recombinase activity (Dixit et al, 2024) and implicated in maintaining genetic stability and tumor avoidance (Barber et al, 2008). We provide mechanistic insight into arginine methylation-mediated regulation of Scd6-*SRS2* mRNA interaction during HU stress, thus identifying a hitherto unknown translation regulation of mRNAs encoding key DNA damage repair proteins.

# Results

## Scd6 localizes to granules in response to several genotoxic stresses

We began by assessing the localization of RGG-motif-containing RNA-binding protein Scd6 upon exposure to different genotoxic stresses such as methyl methanesulphonate (MMS), ultraviolet (UV) radiation, hydroxyurea (HU), cisplatin (CSP) and zeocin in the yeast *S. cerevisiae* (Fig. 1A). Scd6 localized to distinct cytoplasmic puncta upon 60 min of treatment with 0.03% MMS,

50 J/m² UV radiation, 150 µM CSP, and 45 min of treatment with 200 mM HU (Fig. 1B). Interestingly, treatment with 100 µg/ml zeocin for 60 min (Fig. 1B) did not induce localization of Scd6 to puncta. GFP fluorescence intensity calculations revealed a marginal but significant increase in Scd6-GFP levels upon UV irradiation and Zeocin treatment (Fig. 1C), but no changes in protein levels were observed with other genotoxic stresses. Altogether, our live cell imaging data suggests that Scd6 responds to several genotoxic stresses by changing its localization to cytoplasmic foci, which probably influences DNA damage response.

## SRS2 overexpression increases HU sensitivity in the absence of Scd6

To understand the role of Scd6 in genotoxic stress response, we took cue from a study that investigated the synthetic dosage lethality of SRS2 (a 3'–5' DNA helicase involved in DNA damage response) and identified a class of genes involved in RNA metabolism whose deletion led to aggravated SRS2 associated lethality (León Ortiz et al, 2011). We hypothesized that since Scd6 is an RNA-binding protein that localized to puncta in response to several genotoxic stress, it could modulate SRS2 overexpression-associated lethality. Overexpression of SRS2 using a low copy number plasmid compromised cell growth in the presence of MMS and HU but not UV radiations. Deletion of SCD6 strongly compromised the growth of cells upon SRS2 overexpression in the presence of HU. However, the growth defect was milder in the presence of MMS. Growth in the presence of UV, cisplatin and zeocin was not impacted by the overexpression of SRS2 in the absence of Scd6 (Figs. 2A,B and EV1A). In addition, overexpression of SCD6 completely rescued the growth defect caused by SRS2 overexpression in the presence of HU (Figs. 2C and EV1B). Altogether, these results provided an important insight into the role of Scd6 in the HU-mediated stress response, indicating that Scd6 modulated SRS2 in an HU-dependent manner, which was investigated further.

## Scd6 represses SRS2 mRNA translation in HU-dependent manner

Scd6 is a translation repressor, and therefore, we hypothesised that Scd6 might regulate the translation of the *SRS2* mRNA specifically upon HU stress. To test this, we expressed Scd6-GFP in a 2µ plasmid under its own promoter. We looked at its effect on global translation using polysome profiling. Scd6 overexpression or HU treatment did not cause global translation defects (Fig. 2D). To test the role of Scd6 on *SRS2* mRNA translation upon HU stress, we isolated RNA from the sucrose gradient fractions followed by RT-qPCR to estimate the amount of *SRS2* mRNA in polysome fractions. We found that, upon Scd6 overexpression in HU-treated cells, there is a significant decrease in the amount of *SRS2* mRNA present in the polysome fractions (normalized by *SRS2* mRNA present in untranslated fraction) as compared to untreated cells (Fig. 2E). We observed no changes in the distribution in polysome fractions of the *DNL4* (DNA ligase IV) and *RNR4* (Ribonucleotide Reductase 4) mRNAs, which are known to encode proteins involved in DNA damage response (Fig. EV1C). These results indicate that Scd6 specifically represses the translation of the *SRS2* mRNA upon HU stress.

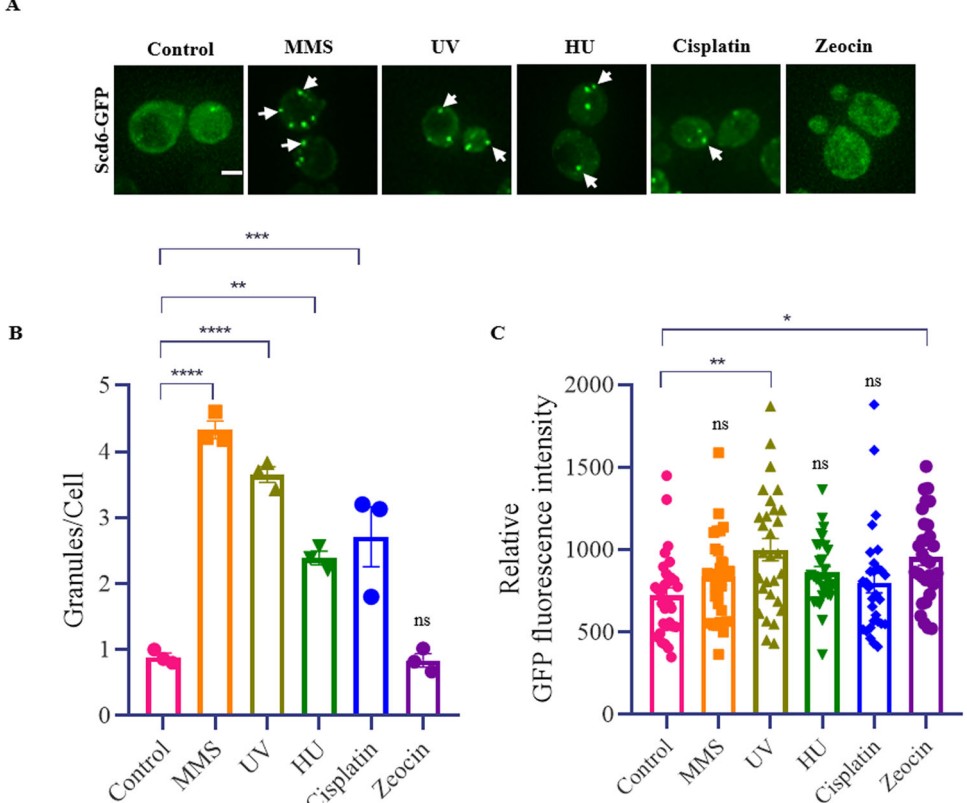

**Figure 1. Scd6 localizes to granules in response to several genotoxic stresses.**

(A) Live cell imaging showing localization of GFP tagged Scd6 expressed from a CEN plasmid under its own promoter, upon treatment with 0.03% methyl methanesulphonate (MMS), 50 J/m$^2$ ultraviolet (UV) radiation, 200 mM hydroxyurea (HU), 150 μM cisplatin and 100 μg/ml zeocin. White arrows indicate cytoplasmic foci. Scale bar = 2 μm. (B) Quantification of Scd6 granules. Statistical significance was calculated using Tukey's multiple comparisons test. Error bars indicate the standard error of mean. $n = 3$ (biological replicates); ≥300 cells were counted (Control vs MMS, ****$p < 0.0001$; Control vs UV, ****$p < 0.0001$; Control vs HU, **$p = 0.0030$; Control vs Cisplatin, ***$p = 0.0006$). (C) Quantification of relative GFP fluorescence intensity from the same experiments. Statistical significance was calculated using Tukey's multiple comparisons test. Error bars indicate the standard error of mean. $n = 3$ (biological replicates); 90 cells were used for GFP fluorescence intensity quantification. (Control vs UV, **$p = 0.0031$; Control vs Zeocin, *$p = 0.0219$). Source data are available online for this figure.

## HU-induced Scd6 puncta are dynamic and sensitive to cycloheximide

We next tested the role of Scd6 localization to puncta in HU stress and its possible connection to *SRS2* mRNA repression. To investigate whether mRNAs were present in the Scd6 puncta formed upon HU treatment, we treated cells with cycloheximide (CHX), which locks the mRNA on polysome, thereby reducing its availability for granule assembly and maintenance. When HU-treated cells were subjected to 0.1 mg/ml of CHX for 5 min at 30 °C (Fig. 2F, upper panel), there was a substantial decrease in Scd6-GFP granules (Fig. 2G, left panel). Since granules are dynamic structures that depend on active translation, CHX treatment decreased Scd6 granule number. This suggests that mRNAs are present in HU-induced Scd6 puncta and that most of the mRNAs present in these granules could be recycled for translation in polysomes. RNA granules disassemble rapidly upon removal of stress (Brengues et al, 2005). The reversible nature of RNA granules is essential as it allows the return of mRNAs and various RBPs to the cytoplasm. To examine if the HU-dependent formation of Scd6 granules is reversible, we performed the experiments described earlier but

followed by a recovery period (Fig. 2F, lower panel). The HU-treated cells were resuspended and grown in HU-free media during recovery. After 75 min of recovery, Scd6 puncta decreased significantly (Fig. 2G, right panel). Based on these observations, we conclude that the HU-induced Scd6 granules are reversible. We further characterize Scd6-containing granules by enriching these from yeast cells and assessing their properties. Scd6-containing granules were resistant to NaCl and EDTA treatment, indicating that hydrophobic interactions stabilize these granules. RNase treatment did not perturb these structures, indicating that the RNA is protected by partner RBPs and/or protein–protein interactions play a major role in the integrity of the granules once the core is formed. As expected, 1% SDS treatment caused a complete loss of Scd6 signal from the pellet fraction and appeared in the soluble fraction. All the above properties, including resistance to RNase treatment, are consistent with reported properties of stress granule core (Jain et al, 2016), confirming that the Scd6 puncta are not protein aggregates but higher-order mRNA-protein complexes harboring Scd6 (Fig. EV2A).

Scd6 is a modular protein, and domain deletion experiments revealed that the absence of the Lsm domain completely abrogated

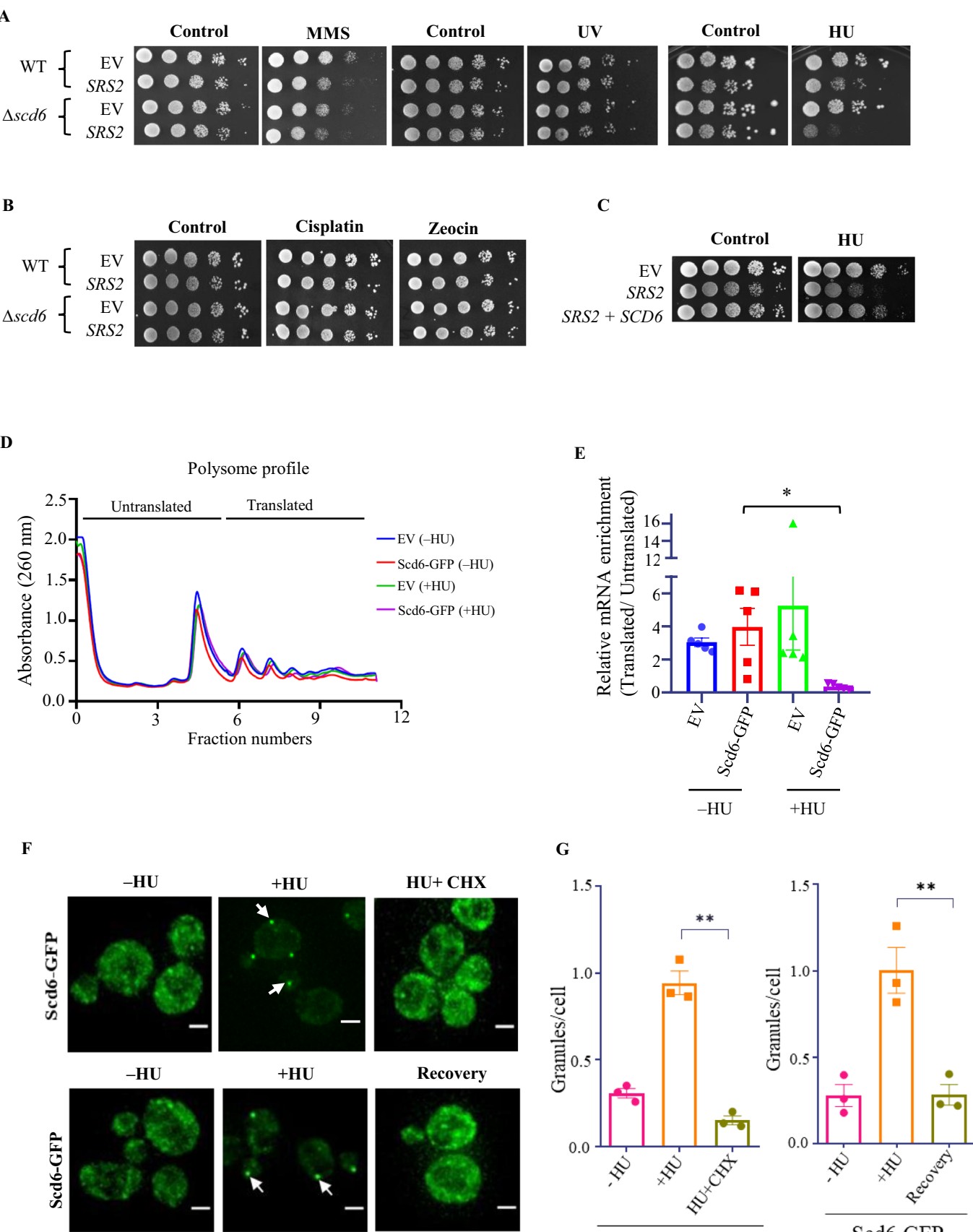

Figure 2. SCD6 genetically interacts with SRS2 and represses its tranlastion.

(A, B) Spot assay of WT and Δscd6 strain either carrying EV, or SRS2 on CEN plasmid, spotted on Uracil dropout agar plate and 2% glucose, in the presence of 0.01% methyl methanesulphonate (MMS), 100 mM HU, 150 μM cisplatin and 100 μg/ml Zeocin or plate treated with 30 J/m$^2$ ultraviolet (UV) radiation after spotting. The plates were incubated at 30 °C. (C) Spot assay with WT either carrying EV, or SCD6 on 2μ plasmid, along with SRS2 on CEN plasmid, spotted on Uracil-Leucine dropout agar plate and 2% glucose, in the presence of 100 mM HU. (D) Representative polysome profiles of 200 mM HU treated or Control cells expressing Scd6-GFP on a 2μ plasmid (or EV). (E) Quantification of SRS2 mRNA in the polysome fractions plotted as relative log2-Fold change ratio of Translated/Untranslated fractions ($n = 5$, biological replicates) using SRS2-specific primer normalized to total mRNA and PGK1 (internal control). Error bars indicate standard error of mean and statistical significance was calculated using unpaired t-test. [Scd6(−HU) vs Scd6(+HU), *$p = 0.0121$]. (F) Live cell image showing the change in localization of endogenously tagged Scd6-GFP after 5 min of 0.1 mg/ml CHX treatment post 55 min of HU treatment and localization after incubating in non-HU-containing media for 75 min (recovery). White arrows indicate cytoplasmic foci. Scale bar = 2 μm. (G) Quantification of granules as granules per cell (Left panel shows granule count for CHX treatment and right panel for recovery). $n = 3$ (biological replicates); ≥300 cells were counted for analysis; ≥100 cells were used for intensity calculations. Error bars indicate the standard error of the mean, and statistical significance was calculated using an unpaired t-test. (HU vs CHX, **$p = 0.0029$; HU vs Recovery, **$p = 0.0080$). Source data are available online for this figure.

Scd6 localization to puncta, whereas RGG motif deletion reduced puncta localization upon HU stress (Fig. EV2B,C). We measured the GFP fluorescence intensity to estimate the expression of WT and domain deletion mutants of both proteins and observed comparable expression levels (Fig. EV2D), indicating that reduced localization to granules is not due to decreased protein levels. Since CHX experiments indicated that these granules contain mRNAs, the localization of Scd6 to these puncta could regulate the fate of specific mRNAs like SRS2 in response to HU.

## The SRS2 mRNA associates with Scd6-containing granules in HU-treated cells

Our results suggest that HU increases the localization of Scd6 to granules and that Scd6 represses the translation of the SRS2 mRNA. We used two distinct yet complementary approaches to check if this repression occurred in granules. The first one used smFISH (single-molecule fluorescence in situ hybridization) to detect the localization of the SRS2 mRNA using Cy5-labeled secondary probes with affinity to specific oligonucleotides targeting the SRS2 mRNA. Upon HU treatment, there was a significantly increased overlap of SRS2 mRNA foci with Scd6 granules (Fig. 3A–C), even though the total number of SRS2 mRNA or Scd6 protein foci did not change significantly (Fig. 3D, left panel and EV2E). Since Scd6 was expressed from a 2μ plasmid, there was an increase in the number of granules per cell to ~3 granules per cell (Fig. EV2E), in the absence of treatment as compared to when expressed from a CEN plasmid (Fig. 1A). Therefore, we calculated the percentage granular localization (granule intensity) of Scd6 upon HU treatment, which showed a significant increase in Scd6 localization to granules (Fig. 3D, right panel). These data point towards an interaction between Scd6 and SRS2 mRNA in granules.

Using a complementary biochemical approach, we assessed the association of SRS2 mRNA and Scd6 protein in the cytoplasmic (lighter) and granule-enriched (heavier) fractions from lysates of cells treated with HU (Fig. 3E) (Wheeler et al, 2017). We observed a significant enrichment of the Scd6 protein and the SRS2 mRNA in granules-enriched fractions from HU-treated cells expressing Scd6-GFP (Fig. 3F), with no significant increase in total SRS2 mRNA levels (EV2F). The enrichment of the SRS2 mRNA in granules-enriched fractions was, however, significantly hampered in HU-treated cells expressing the LSm domain-deletion mutant of Scd6 (Fig. 3F, right panel). The enrichment of this mutant itself is highly defective in granule-enriched fractions as compared to the wild-type Scd6 protein which is consistent with the localization result using live-cell imaging (Fig. EV2B).

If localization to Scd6 granules was important for SRS2 mRNA repression then the inability of SRS2 mRNA to localize to granules could lead to decreased repression activity. To test this, we performed polysome profiling followed by RNA isolation and RT-qPCR to quantify the enrichment of SRS2 mRNA in polysome upon expression of wild type or LSm domain-deletion mutant of Scd6. As observed earlier, Scd6 expression led to increased repression of SRS2 mRNA, whereas a significant reduction in SRS2 mRNA repression was observed when LSm domain-deletion mutant was expressed (Fig. 3G). The results indicate that the Scd6 protein and the SRS2 mRNA co-localize to granules upon HU treatment where SRS2 mRNA is translationally repressed. SRS2 mRNA localization to granules is important for its translation repression.

## Scd6 interacts with the SRS2 mRNA

To investigate the possible HU-dependent association of Scd6 to the SRS2 mRNA, we performed RNA immunoprecipitation (RIP) in lysates of cells expressing Scd6-GFP in the presence of HU (i.e., in the same conditions as the polysome profiling experiment in Fig. 2D) (Fig. 4A,B). The amounts of SRS2 mRNA expression were quantified in the Scd6 immunoprecipitates (IP) and normalized with total SRS2 mRNA and PGK1 control (using PGK1-specific primers). We observed a significant enrichment of SRS2 mRNA in the Scd6 IP from HU-treated cell lysate (Fig. 4C). This enrichment was not visible when the experiment was performed in cells expressing ΔLSm or ΔRGG mutants of Scd6, indicating a role of these protein domains in the Scd6-SRS2 mRNA interaction. Transcripts encoding two key DNA damage response proteins, DNL4 (the yeast homolog of Ligase IV) and RNR4, were also tested. The DNL4 mRNA, but not the RNR4 mRNA, was significantly enriched in the Scd6 IP when cells were exposed to HU (Fig. EV3A), similar to the SRS2 mRNA. According to the polysome profiling results, the DNL4 mRNA was, however, not translationally repressed by Scd6 (Fig. EV1C).

## Scd6 methylation and Scd6-eIF4G1 interaction decrease, whereas Scd6-SRS2 mRNA interaction increases upon HU stress

Scd6 is arginine methylated at RGG-motif, which regulates its repression activity via promoting binding to eIF4G1 protein (Poornima et al, 2016). However, the direct role of arginine methylation in Scd6 interaction with RNA is unknown. We checked if a change in the methylation status of Scd6 could

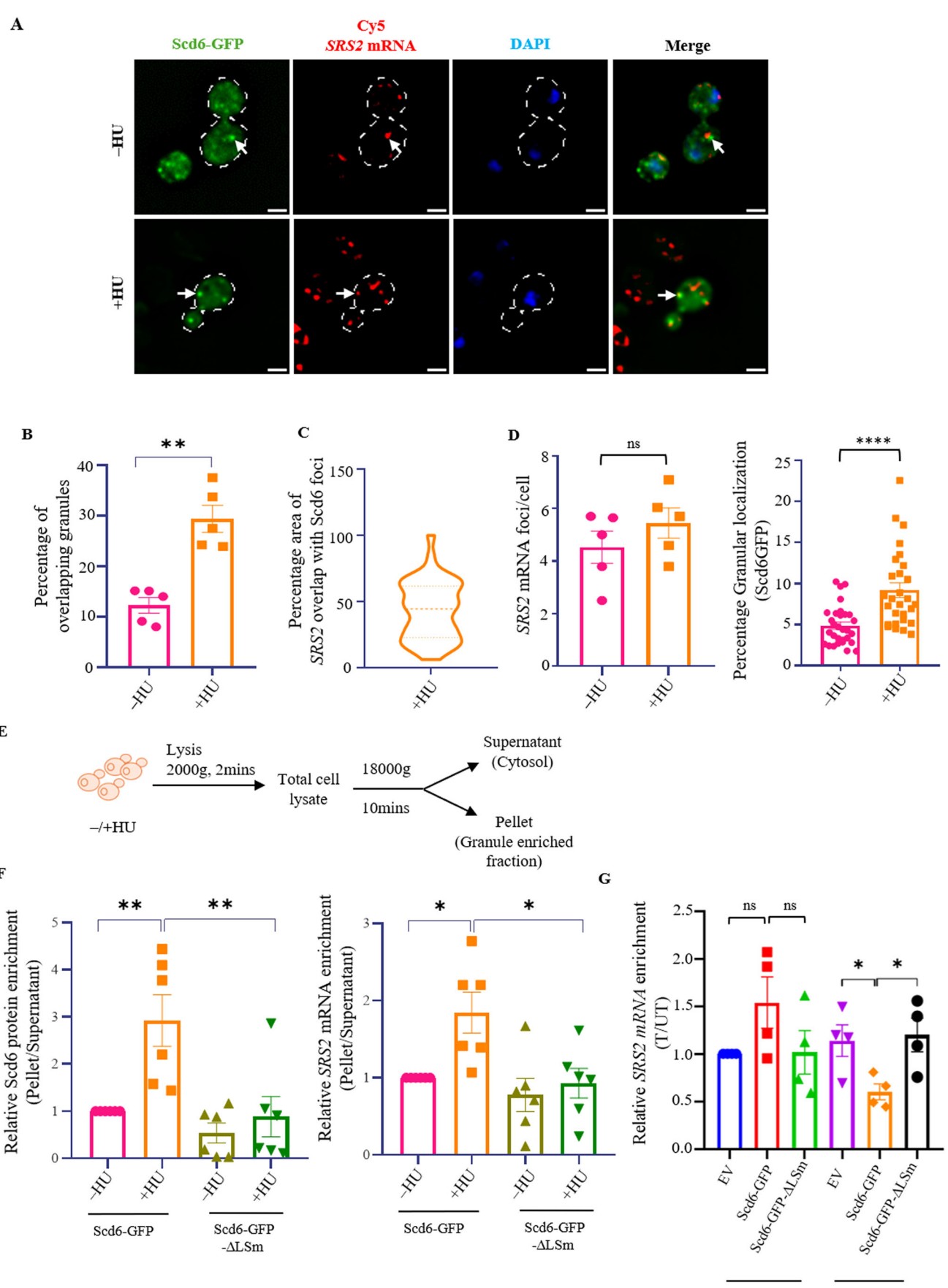

**Figure 3.  SRS2 mRNA localizes to Scd6 granules upon HU stress.**

(A) smFISH showing localization of *SRS2* mRNA and Scd6GFP in control and HU-treated cells. White arrows indicate partially overlapping *SRS2* and Scd6 foci. White dotted line denotes cell boundary. Scale bar = 2 μm. (B) Quantification of the percentage of *SRS2* mRNA foci overlapping with Scd6 granules normalized to total Scd6 granules (*n* = 5 biological replicates, ≥100 cells containing both red and green foci were counted). Statistical significance was calculated using paired t-test (**p = 0.0062). Error bars indicate standard error of mean. (C) Quantification of the degree of overlap calculated as the area of percentage overlap of *SRS2* mRNA foci with Scd6 foci and plotted as a violin plot (*n* = 5 biological replicates, 40 overlapping granules were quantitated). (D) Quantification of the number of *SRS2* foci formed upon HU stress compared to unstressed control (left panel) and percentage granular localization of Scd6 for the same cells (right panel, 30 cells were used for quantification) (*n* = 5 biological replicates). Statistical significance was calculated using paired t-test. Error bars indicate the standard error of the mean. ****p = 0.0001. (E) Schematic showing workflow for granule enrichment from cells expressing Scd6-GFP in the presence or absence of 200 mM HU. (F) Quantification of relative Scd6 protein (left panel) and *SRS2* mRNA (right panel) localization in soluble (supernatant) and the heavier (granule enriched fraction) fraction plotted as a ratio of pellet by supernatant (*n* = 6 biological replicates) Statistical significance was calculated using Tukey's multiple comparisons test. Error bars indicate the standard error of mean [For Scd6 protein, Scd6GFP(−HU) vs Scd6GFP(+HU), **p = 0.0063; Scd6GFP(+HU) vs Scd6GFP-ΔLSm(+HU), **p = 0.0038. For *SRS2* mRNA, Scd6GFP(−HU) vs Scd6GFP(+HU), *p = 0.0295; Scd6GFP(+HU) vs Scd6GFP-ΔLSm(+HU), *p = 0.0172]. (G) Quantification of *SRS2* mRNA in the polysome fractions plotted as relative log2-fold change ratio of Translated/Untranslated fractions (*n* = 4 biological replicates) using SRS2-specific primer normalized to total mRNA and PGK1 (internal control). Statistical significance was calculated using paired t-test. Error bars indicate the standard error of the mean. [EV (+HU) vs Scd6GFP (+HU), *p = 0.0204; Scd6GFP(+HU) vs Scd6GFP-ΔLsm(+HU), *p = 0.0228]. Source data are available online for this figure.

mediate the increased interaction of Scd6 with the *SRS2* mRNA upon HU stress. A western blot using an antibody against mono-methylated arginine (MMA) revealed a signal for Scd6, which was absent, as expected, for the RGG deletion mutant. We observed a significant decrease in Scd6 methylation upon HU stress. Interestingly, this was not the case with the Scd6 mutant lacking the LSm domain, whose methylation levels were significantly more than the WT Scd6 in the presence of HU (Fig. 4D,E). This suggested that the LSm domain could negatively regulate the methylation status of Scd6, thereby promoting RNA binding upon HU stress, which is consistent with the observations that the LSm domain and the RGG motif are required for *SRS2* mRNA interaction (Fig. 4C). These results identify HU-mediated altered methylation of Scd6 as a mechanism to modulate *SRS2* mRNA binding. They also highlight the unexpected role of the Lsm domain in modulating Scd6 methylation.

Since the previously reported mode of Scd6 translation repression is via binding to eIF4G1, which is augmented by Scd6 methylation, we checked the Scd6-eIF4G1 interaction by Scd6-GST immunoprecipitation. We observed reduced Scd6-eIF4G1 interaction upon HU treatment (Fig. 4F,G). A decrease in Scd6 methylation and a concomitant decrease in Scd6-eIF4G1 interaction upon HU indicates a previously unexplored alternate mode of translation repression by Scd6.

Since HU treatment reduced Scd6 methylation (Fig. 4D) and increased Scd6 interaction with the *SRS2* mRNA in vivo (Fig. 4C), we tested if the LSm domain altered Scd6 methylation in vitro and which if, in turn, directly altered RNA binding activity using purified recombinant proteins (Fig. 4H) and in vitro transcribed *SRS2* mRNA. As observed in vivo, methylation of Scd6 by purified recombinant Hmt1, the methyltransferase known to methylate Scd6, was significantly higher in the Scd6ΔLSm mutant (Fig. 4I,J), reinforcing the cis-regulation of Scd6 methylation by the LSm domain. To analyze Scd6 RNA binding activity in vitro, we transcribed 200-mer 5'UTR and 3'UTR of *SRS2* mRNA and performed RNA electromobility shift assay (EMSA) with recombinant purified Scd6. We observed a shift in both the mRNA fragments when incubated with increasing concentrations of Scd6, however, significantly more binding was observed with the 5'UTR fragment of *SRS2* mRNA compared to the 3'UTR fragment (Fig. EV3B,C). Furthermore, to investigate the role of methylation in the RNA binding activity of Scd6, in vitro methylated Scd6 was

used to set up EMSA with the 5'UTR fragment. We confirmed the methylation of Scd6 by western blotting using a mono-methylated arginine-specific antibody (Fig. EV3D). We found that the binding of methylated Scd6 to *SRS2* mRNA was significantly reduced compared to unmethylated Scd6 with Kd values of 2.7 ± 1.3 μM and 2.86 ± 1.1 μM, respectively (Fig. 4K,L). These results provide a direct demonstration that reduced methylation of Scd6 increases *SRS2* mRNA binding, thereby suggesting that the HU-dependent regulation of Scd6 methylation is important for RNA binding.

## LSM14A localizes to puncta upon HU treatment in an RGG-motif-dependent manner and represses translation of specific mRNAs

To analyze whether the observed role of Scd6 in genotoxic stress response was conserved in humans, we focused on LSM14A, the human ortholog of Scd6 (Yang et al, 2006). A direct role of LSM14A in repressing translation remains to be demonstrated. Like Scd6, LSM14A is a modular protein (Fig. 5A) with two RGG domains. Using live cell imaging, we observed that LSM14A localized to puncta upon HU treatment in an RGG-motif-dependent manner (Fig. 5B,C), indicating that it is a conserved feature of the Scd6 family of proteins in yeast and humans. To examine the role of LSM14A in regulating mRNA translation in response to HU treatment, we performed polysome profiling experiments (Fig. 5D). RNAs isolated from translating (heavy) and non-translating (light) fractions of wild type and LSM14A knockdown (siRNA) cells were sequenced to identify mRNAs whose association with polysomes was perturbed by LSM14A (Dataset EV1 and EV2). Several mRNAs were identified to be differentially regulated. To address the role of LSM14 in the HU response, a similar analysis was carried out for LSM14A-depleted cells treated with HU. Again, many mRNAs were identified to be differentially regulated at the translation level, highlighting the important role of LSM14A in translation control under normal conditions and genotoxic stress (Fig. 5E).

We observed that the association of transcripts encoding DNA damage repair proteins, including Ligase IV (NHEJ pathway protein, DNL4 yeast homolog) and RTEL1 (the functional homolog of SRS2), with translating fractions increased upon HU stress in cells depleted for LSM14A (Fig. 5E). RT-qPCR analysis of RNAs extracted from each fraction of the sucrose gradient confirmed this regulation since we

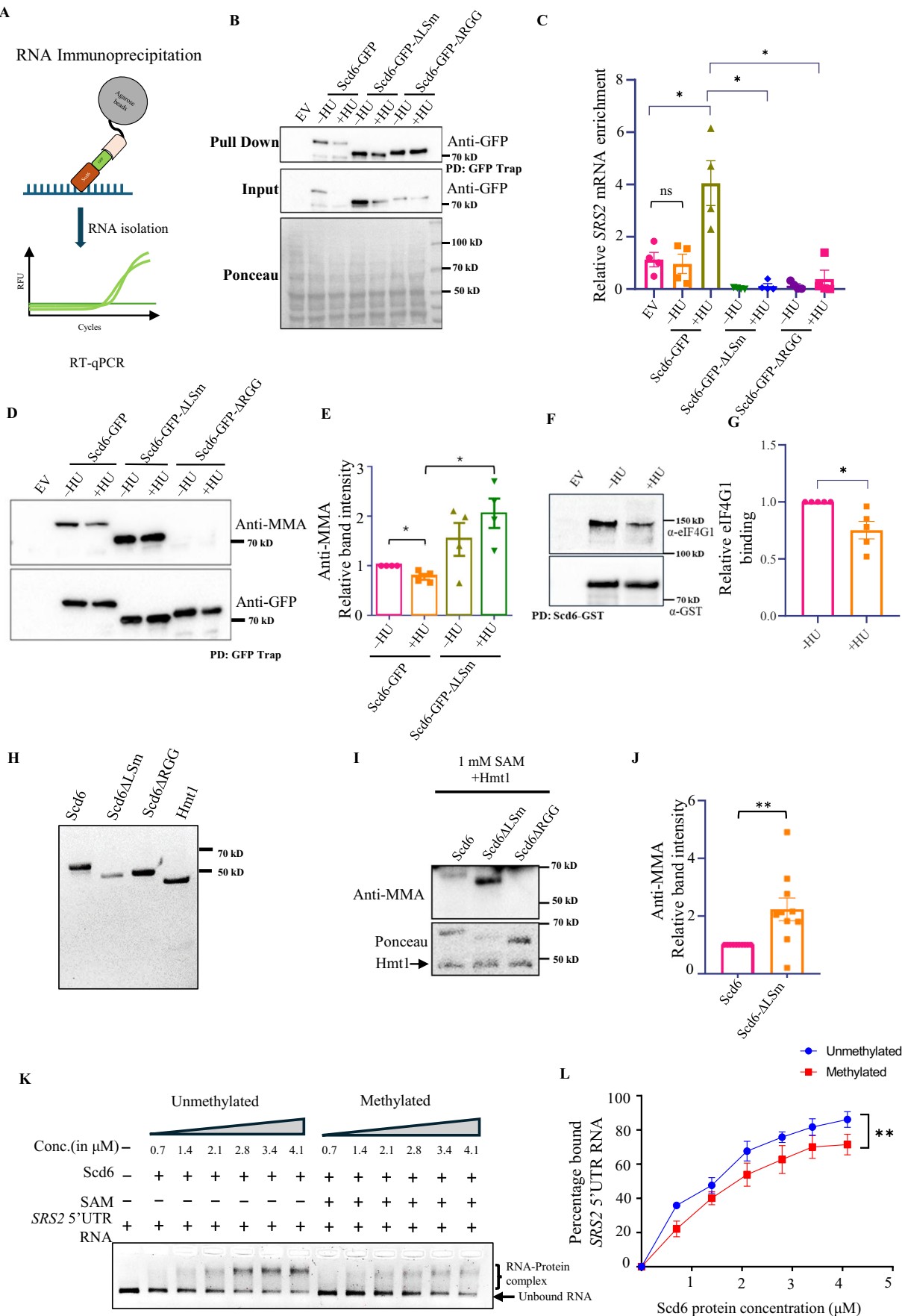

**Figure 4. Scd6 binds to *SRS2 mRNA* upon HU treatment in a manner modulated by methylation.**

(A) Approach for RNA immunoprecipitation (RIP) using GFP-tagged Scd6 (expressed in a 2 μ plasmid, under its own promoter) in untreated (−HU) and 200 mM HU treatment. (B) Representative western blot showing pull down of Scd6-GFP WT and domain deletion mutants used in RIP. (C) Quantification of *SRS2* mRNA enriched in the pull-down fraction, plotted as relative log$_2$-Fold change w.r.t total mRNA ($n = 4$ biological replicates). Statistical significance was calculated using paired t-test. Error bars indicate the standard error of the mean. [EV vs Scd6GFP (+HU), *$p = 0.0452$; Scd6GFP(+HU) vs Scd6GFP-ΔLSm(+HU), *$p = 0.0151$; Scd6GFP(+HU) vs Scd6GFP-ΔRGG(+HU), *$p = 0.0100$]. (D) Western blot showing arginine-mono-methylation of Scd6-GFP WT and domain deletion mutants upon HU stress which is quantified in (E) $n = 4$ biological replicates. Statistical significance was calculated using paired t-test. Error bars indicate the standard error of the mean. Scd6GFP(−HU) vs Scd6GFP(+HU), *$p = 0.0364$; Scd6GFP(+HU) vs Scd6GFP-ΔLSm(+HU), *$p = 0.0348$). (F) Western blot showing interaction of eIF4G1 with Scd6-GST in control and HU treated samples which is quantified in (G) $n = 5$ biological replicates. Statistical significance was calculated using paired t-test. Error bars indicate the standard error of the mean. *$p = 0.0320$. (H) Coomassie Brilliant Blue (CBB) stained gel of purified His-Scd6-FLAG (HSF), HSF-ΔLSm, HSF-ΔRGG and His-Hmt1 used for in vitro experiments. (I) Western blot for in vitro arginine-mono-methylation of recombinant WT and domain deletion mutants of Scd6 incubated with equimolar concentration of recombinant Hmt1 and 1 mM S-adenosyl methionine (SAM) for 120 mins at 37 °C. (J) Quantification of relative in vitro mono-methylation as shown in I ($n = 10$ biological replicates). Statistical significance was calculated using an unpaired t-test. Error bars indicate the standard error of the mean. **$p = 0.0062$. (K) Ethidium Bromide (EtBr) stained agarose gel for electromobility shift assay (EMSA) with increasing concentrations of recombinant Scd6 with or without in vitro methylation incubated with 200-mer 5′UTR fragment of *SRS2 mRNA* (0.94 μM RNA). (L) Quantification of percentage bound *SRS2 mRNA* fragment as a function of protein concentration ($n = 3$ independent replicates). Statistical significance was calculated by paired t-test. Error bars indicate the standard error of the mean. **$p = 0.0017$. Source data are available online for this figure.

observed a slight shift in the distribution of the *Ligase IV* and *RTEL1* mRNAs, but not the *HPRT* mRNA used as a negative control, towards heavy polysome fractions upon HU treatment (Fig. 5F). We also confirmed increased RTEL1 protein levels upon LSM14A knockdown in HU stress (Fig. 5G). Since both Ligase IV and RTEL1 have been implicated in NHEJ, we examined if the altered translation of these genes led to a change in NHEJ activity. We monitored NHEJ activity using a plasmid integration assay. In this assay, linearized DNA with a neomycin selection gene is transfected, and the efficiency of random chromosomal integration of the plasmid DNA by NHEJ is measured by colony formation in G418-containing media. We observed that LSM14A knockdown significantly increases the NHEJ activity upon HU stress (Fig. 5H), indicating that LSM14A-mediated translation regulation is associated with the DNA damage response to HU stress.

Since Srs2 is involved in NHEJ-mediated DNA damage repair (Carter et al, 2009; Hegde, 2000), we assessed the role of Scd6 in NHEJ repair upon HU stress along with MMS and UV stress which induced Scd6 localization to cytoplasmic puncta (Fig. 1A,B). We used a previously reported suicide deletion strain which can directly be used for quantitating NHEJ activity (Karathanasis and Wilson, 2002). To identify if Scd6 mediated regulation of SRS2 has an implication in the NHEJ repair pathway upon HU, MMS and UV stress, we transformed these strains with empty vector or Scd6GFP expressed on a 2μ plasmid. We observed a severe defect in NHEJ-mediated repair in the presence of all three stresses upon Scd6 expression compared to the empty vector (Fig. 5I). Interestingly, the same was not observed in the presence of zeocin, which failed to induce localization of Scd6 to puncta. This indicates that the regulation by Scd6 signficiantly modulates NHEJ activity upon genotoxic stress.

## Discussion

In this report, we have explored and identified a conserved role of RGG motif-containing Scd6 family proteins in genotoxic stress response to HU. Several observations support this conclusion: (i) Scd6 localizes to puncta in response to several genotoxic stress such as methyl methanesulfonate (MMS), ultraviolet (UV) radiations, hydroxyurea (HU) and cisplatin (Fig. 1), (ii) Absence of Scd6 makes the cells sensitive to SRS2 overexpression in the presence of HU and MMS

(Fig. 2), (iii) Scd6 represses *SRS2* mRNA translation upon HU stress (Fig. 2), (iii) HU-induced Scd6 granules are sites of *SRS2* mRNA enrichment and repression (Fig. 3), (iv) Scd6 binds to *SRS2* mRNA in vivo upon HU stress (Fig. 4), (v) LSm domain deletion mutant which is defective in binding to SRS2 mRNA poorly enriches SRS2 mRNA in granules resulting in its poor repression (Fig. 3), (vi) Arginine methylation of Scd6 is reduced upon HU stress which leads to decreased eIF4G1 binding (Fig. 4), (vii) LSm domain deletion mutant is hypermethylated and defective in binding *SRS2* mRNA (Fig. 4), (viii) Methylation reduces Scd6 protein interaction with SRS2 mRNA (Fig. 4), (ix) LSM14A localizes to granules upon HU stress and represses translation of multiple transcripts upon HU stress including *LIG4* and *RTEL1* (Fig. 5), and (x) LSM14A/Scd6 modulate NHEJ activity upon HU, MMS and UV stress (Fig. 5). These results identify a new role (summarized in a model; Fig. 6) of Scd6 family proteins in modulating genotoxic stress response by repressing specific mRNAs.

Scd6 responds to HU, MMS, UV and cisplatin stress by localizing to discrete cytoplasmic puncta. These genotoxins function through different mechanisms, leading to different kinds of damage, which leads to DNA double-strand break. Zeocin, a glycopeptide, intercalates the DNA, leading to single-strand and double-strand DNA cleavage. Surprisingly, Zeocin, which can also lead to double-stranded breaks, does not induce relocalization of Scd6 to puncta. This suggests that the signaling mechanism(s) directing Scd6 localization to puncta are likely different between zeocin and other stressors tested in this study. HU, MMS, UV and cisplatin mainly cause double-strand breaks by their encounter with the replication and repair machinery mainly in the S-phase of the cell cycle, whereas Zeocin can directly cause DNA lesions. This could also indicate a cell-cycle-specific regulation of GSR, which is mediated by Scd6-like proteins. This aspect of regulation needs further analysis and is only a conjecture at this point.

The genetic interaction between Scd6 and SRS2 is stress-specific as it manifests upon exposure to HU (strong) and MMS (weak) but not in the presence of UV and cisplatin. This raises the possibility that Scd6 could modulate the translation and/or decay of distinct stress-specific mRNA subsets in puncta. The specific mRNA targets modulated by Scd6 upon exposure to MMS, UV and cisplatin are yet to be identified. Moreover, identifying any specific RNA sequence or structural motif that could lead to Scd6 binding (either direct or mediated by another RBP) also remains an interesting area of investigation.

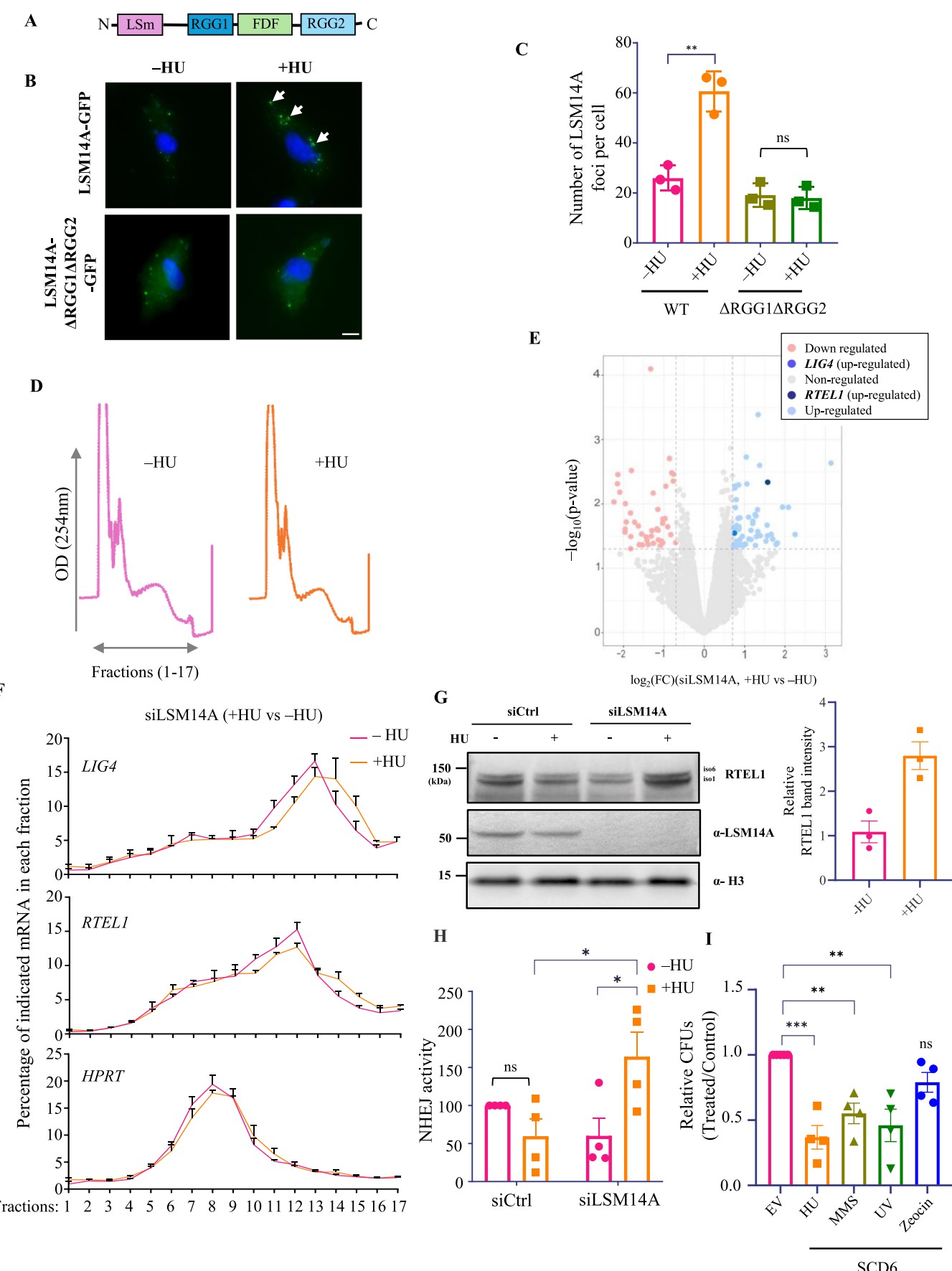

**Figure 5. LSM14A localizes to puncta upon HU stress to regulate specific mRNA translation and modulate NHEJ activity.**

(A) Domain organization of LSM14A. LSM14A contains an N-terminal LSm domain, an FDF motif and two RGG motifs flanking the FDF motif. (B) Localization of LSM14A WT and RGG-deletion mutants upon HU treatment in retinal pigment epithelial cells (RPE). Scale bar = 10 μm. (C) Localization of LSM14A WT and RGG-deletion mutant to granules as depicted in (B), quantified as granules per cell. $n = 3$ biological replicates. Statistical significance was calculated using an unpaired t-test. Error bars indicate the standard error of the mean. **$p = 0.0030$. (D) Polysome profiles of siLSM14A-transfected A2058 cells treated with HU. OD: Optical density. (E) Volcano plots report the $\log_2$ fold change—$\log_2$(FC)—on the x-axis and the minus $\log_{10}$ of the p-value –$\log_{10}$(p-value) on the y-axis obtained from the comparison of translation efficiencies in the +HU vs –HU treated A375 cells (Dataset EV1). Differential analysis between conditions was done using the R package Xtail. Blue dots highlight the translationally up-regulated mRNAs, while red dots highlight the translationally down-regulated mRNAs. (F) Validation of *LIG4* and *RTEL1* mRNA in each polysome fraction comparing siLSM14A untreated (–HU) condition with siLSM14A HU treated condition. The distribution of control *HPRT* mRNA in the polysome fractions is also shown. $n = 3$ biological replicates. (G) Western blot showing endogenous Rtel1 protein levels in A2058 cells and its quantification. Quantification for Rtel1 was done for both bands corresponding to the 2 endogenous isoforms of Rtel1 together normalized to H3. To calculate fold change, the intensity of siCtrl (+/−HU) was used for normalizing siLSM14A (+/−HU) treatment, respectively. Statistical significance was calculated by unpaired t-test. Error bars indicate the standard error of the mean. *$p = 0.0126$. (H) Quantification of plasmid integration efficiencies in A375 cells transfected with the indicated siRNAs and treated with or without HU. Data were normalized to control transfection without HU treatment, which was set to 100%. The statistical significance of the experimental data was determined using Two-Way ANOVA. Error bars indicate standard error of mean, $n = 4$ biological replicates. siControl (+HU) vs siLSM14A (+HU), *$p = 0.0429$; siLSM14A (−HU) vs siLSM14A (+HU), *$p = 0.0438$. (I) Quantification of NHEJ activity in yeast suicide deletion strains expressing either empty vector or Scd6-GFP on a 2μ plasmid under its own promoter upon 100 mM HU, 0.01% MMS, 30 J/m² ultraviolet (UV) radiation and 100 μg/ml zeocin ($n = 4$ biological replicates). Statistical significance was calculated using Tukey's multiple comparisons test. Error bars indicate the standard error of the mean. EV vs HU, ***$p = 0.0002$; EV vs MMS, **$p = 0066$; EV vs UV, **$p = 0.0012$. Source data are available online for this figure.

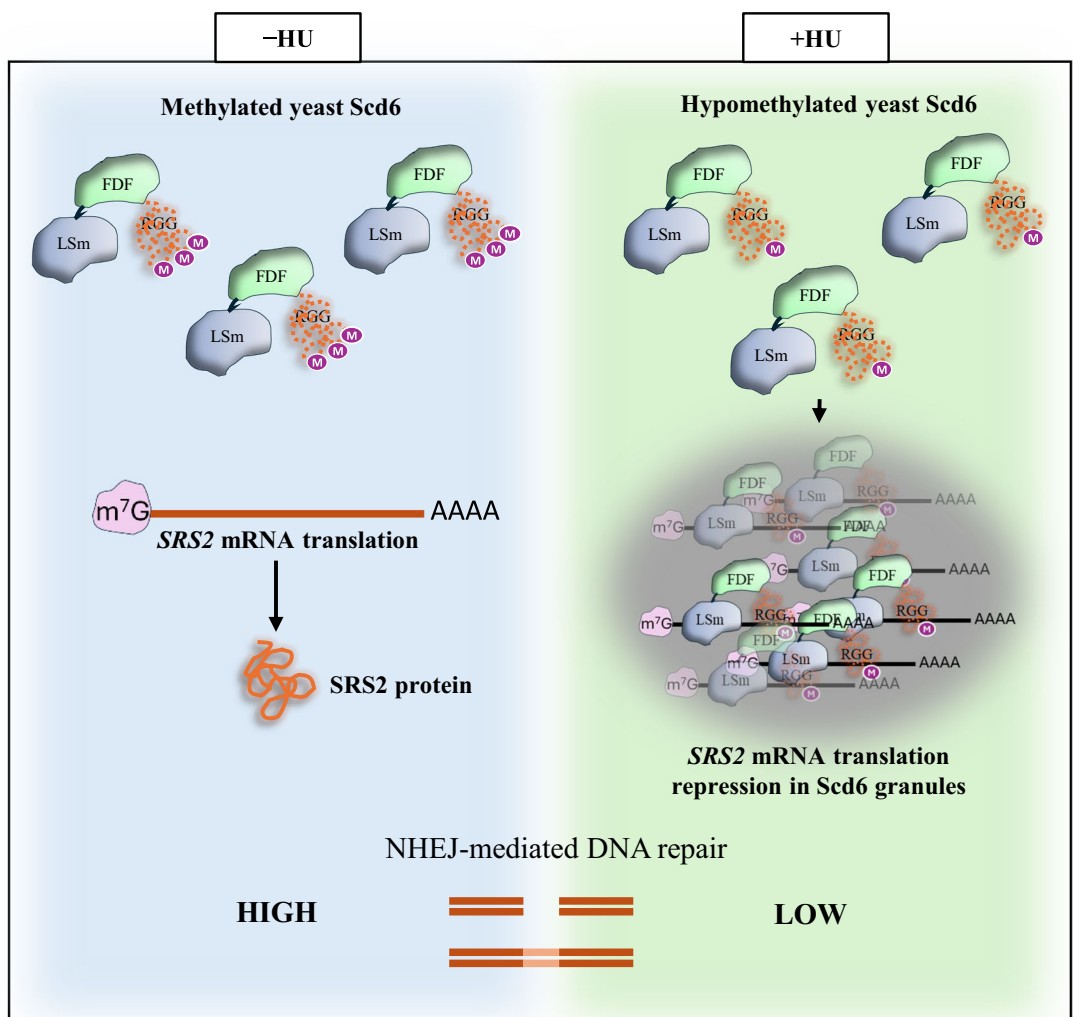

**Figure 6. Scd6 is involved in translation repression of *SRS2* mRNA and modulates NHEJ activity upon HU-induced genotoxic stress.**

In unstressed conditions, *SRS2* mRNA is translated to protein and Scd6 mainly remains cytoplasmic. Upon HU stress, mono-methylation of Scd6 at the arginine residues in the RGG domain decreases, leading to binding of Scd6 to *SRS2* mRNA via its LSm and RGG domain. The Scd6-bound SRS2 mRNA localizes to cytoplasmic granules and remains translationally repressed.

HU is a well-known genotoxin that causes DNA damage by depleting dNTP pools, leading to replication fork arrest, eventually causing DNA double-strand breaks (Singh and Xu, 2016; Koç et al, 2004; Saintigny, 2001). Genome-wide studies have reported changes in the localization of numerous proteins in response to genotoxic stress (Tkach et al, 2012); however, barring a handful of studies, the functional relevance and mechanistic basis of altered localization have not been explored. RNA granules are sites of regulation of mRNA translation and decay. Localization of cytoplasmic RNA binding proteins Scd6, and LSM14A to puncta upon HU treatment suggests that such localization change could lead to the regulation of mRNA fate upon HU. Furthermore, the reversibility of HU-induced Scd6 puncta and their sensitivity to cycloheximide suggests that puncta-resident mRNAs may return to translation once the HU stress is removed.

SRS2 is a 3′–5′ DNA helicase whose expression levels are tightly controlled, which is evident from the observations that either overexpression or deletion of this gene causes sensitivity to several genotoxins (Alex et al, 2018; Bronstein et al, 2018). Even though it has been established that Srs2 levels are tightly regulated in cells (Bronstein et al, 2018), the regulator of this protein remains elusive. No study has addressed the translation regulation of SRS2/RTEL1 transcripts, highlighting that such regulation of DDR genes is a poorly explored avenue. Our work identifies Scd6 as a translation regulator of SRS2, a role conserved in humans.

Our study provides critical mechanistic insights underlying the regulation of SRS2 by Scd6. We observe that Scd6 binds SRS2 mRNA, and this interaction increases with HU stress. Scd6-SRS2 interaction leads to translation repression of SRS2 mRNA in granules, and the inability of SRS2 mRNA to localize to these granules causes de-repression.

Interestingly, individual deletion of the LSm domain or RGG-motif compromises Scd6-SRS2 interaction. Our results suggest that increased methylation levels of RGG-motif mediate the reduced interaction of the Scd6 LSm-domain deletion mutant with SRS2. The following observations support this idea: (a) LSm-domain deletion mutant is hypermethylated as compared to full-length Scd6 (Fig. 4D) and (b) Methylated Scd6 interacts poorly with SRS2 as compared to unmethylated Scd6 (Fig. 4K). The basis for reduced Scd6 methylation and hypermethylation of LSm mutant will be an interesting future direction. Another important observation from this study is the decreased interaction of Scd6-eIF4G1 protein

(Fig. 4F). The well-established mode of Scd6-mediated translation repression is via binding to eIF4G1, which is augmented by Scd6 methylation. A decrease in Scd6-eIF4G1 interaction upon HU stress reveals an alternative mode of Scd6-mediated translation repression, which relies on direct interaction with SRS2 mRNA and its sequestration to puncta upon HU stress. This is further supported by the fact that upon HU stress, eIF4G1 itself does not localize to granules (Tkach et al, 2012).

Our results provide the first evidence for the role of both Scd6 and human LSM14A as a translation regulator upon genotoxic stress. Translation of multiple mRNAs is upregulated in LSM14A knockdown cells exposed to HU. We have validated changes in the translation status of two mRNAs, LIG4 and RTEL1 (Fig. 5F). RTEL1 is the functional homolog of SRS2, which, like SRS2, is an anti-recombinase and a negative regulator of HR-mediated DNA repair (Frizzell et al, 2014; Barber et al, 2008). Mutations in RTEL1 variants have shown a predisposition to various human pathologies (Vannier et al, 2014), and therefore understanding its regulation is a significant step.

LIG4 (DNA Ligase IV) is associated with XRCC4 to promote DNA double-strand break repair via the non-homologous end joining (NHEJ) pathway (Grawunder et al, 1998). LSM14A knockdown increases the translation of LIG4 during HU stress (Fig. 5F). This regulation has functional relevance as we observe that the NHEJ activity, as measured by plasmid integration assay, increases upon the knockdown of LSM14A under HU stress (Fig. 5H). Curiously, DNL4 is enriched in Scd6 pulldown fraction from yeast cells but is not translationally repressed by Scd6 (Figs. EV1C and EV3A). This enrichment could result from indirect interaction mediated by another protein in the complex. Alternatively, it is possible that Scd6 could affect the stability of the DNL4 transcript. SRS2 has also been reported to be involved in the NHEJ pathway (Hegde, 2000). Therefore like Scd6, LSM14A contributes to HU stress response by regulating the translation of a specific mRNA.

Overall, this study establishes RGG motif-containing proteins Scd6 and LSM14A as regulators of the genotoxic stress response by affecting the translation of specific mRNA targets. Identifying the signaling mechanisms that enable genotoxic stress condition-specific repression activity of Scd6 and LSM14A would be a critical future direction.

# Methods

**Reagents and tools table**

| Reagent/Resource | Reference or Source | Identifier or Catalog Number |
|---|---|---|
| **Experimental models (genotype)** | | |
| MATa his3Δ1 leu2Δ0 met15Δ0 ura3Δ0('BY4741') | Poornima et al, 2016 | yPIR1 |
| MATa his3Δ1 leu2Δ0 met15Δ0 ura3Δ0SCD6-GFP (HIS) | Rajyaguru et al, 2012 | yPIR13 |
| MATa his3Δ1 leu2Δ0 met15Δ0 ura3Δ0 hmt1Δ::KanMX | This study | yPIR171 |
| MATa ade2::SD2-::URA3 his3Δ1 leu2Δ0 ura3Δ0 | Karathanasis and Wilson, 2002 | YW714 |
| YW714 MAT yku70Δ::kanMX4 | Karathanasis and Wilson, 2002 | YW713 |

| Reagent/Resource | Reference or Source | Identifier or Catalog Number |
|---|---|---|
| **Recombinant DNA** | | |
| CEN plasmid used as empty vector. *URA3*; Ampicillin resistance gene | | pPIR92 |
| CEN plasmid to express Scd6GFP under its own promoter-*URA3*; Ampicillin resistance gene | Poornima et al, 2019 | pPIR93 |
| CEN plasmid to express Scd6GFP-ΔLSm under its own promoter. Amino acids deleted: 2–93. *URA3*; Ampicillin resistance gene. | Parbin et al, 2019 | pPIR172 |
| CEN plasmid to express Scd6GFP-ΔRGG under its own promoter. Amino acids deleted: 282–348. URA3; Ampicillin resistance gene | Poornima et al, 2019 | pIR64 |
| Multicopy 2μ plasmid used as empty vector. *LEU2*; kanamycin resistance gene | Garg et al, 2022 | pPIR130 |
| Multicopy 2μ plasmid to express Scd6GFP under its own promoter-*LEU2*; kanamycin resistance gene | Garg et al, 2022 | pPIR168 |
| Multicopy 2μ plasmid to express Scd6GFP-ΔLSm under its own promoter. Amino acids deleted: 2–93. *LEU2*; kanamycin resistance gene | This study | pPIR172 |
| Multicopy 2μ plasmid to express Scd6GFP-ΔRGG under its own promoter. Amino acids deleted: 282–348. *LEU2*; kanamycin resistance gene | Garg et al, 2022 | pPIR169 |
| Used to express full-length Scd6 with His- and FLAG tag at N- and C- terminus, respectively, under T7 promoter | Poornima et al, 2019 | pPIR1 |
| Used to express Scd6-ΔLSm with His- and FLAG tag at N- and C- terminus, respectively, under T7 promoter | Parbin et al, 2019 | pPIR188 |
| Used to express Scd6-ΔRGG with His- and FLAG tag at N- and C- terminus, respectively, under T7 promoter | Poornima et al, 2019 | pPIR2 |
| Used to express full-length Hmt1 with His-tag at N-terminus | A kind gift from Anita Corbett | pPIR3 |
| CEN plasmid used as empty vector. *URA3*; Ampicillin resistance gene | | pPIR369 |
| Used to express SRS2 (untagged) under its own promoter. *URA3*; Ampicillin resistance gene | Bronstein et al, 2018 | pPIR369 |
| Mammalian expression vector used to express LSM14A-GFP; kanamycin resistance gene | This study | pPIR98 |
| Mammalian expression vector used to express LSM14A--ΔRGG1ΔRGG2-GFP; kanamycin resistance gene | This study | pPIR101 |
| **Antibodies** | | |
| Anti-Pgk1 | Abcam | ab113687 |
| Anti-myc | Sigma | C3956 |
| Anti-monomethylation arginine | Cell Signalling Technologies | 8711S |
| Anti-His | Cell Signalling Technologies | 27E8 |
| Anti-HisGST | Cell Signalling Technologies | 27E82624 |
| Anti-eIF4G1 | Cocalico Biologicals | |
| Anti-Rabbit | Jackson ImmunoResearch Laboratories | Code No. 111-035-003 |
| Anti-Mouse | Jackson ImmunoResearch Laboratories | Code No. 115-035-003 |
| Monoclonal mouse LSM14A antibody | SantaCruz Biotechnology | sc-398552 |
| Rabbit EIF4G antibody | Cell Signalling Technologies | 2498 |
| Mouse EIF4E antibody | Cell Signalling Technologies | 9742 |
| Secondary mouse AlexaFluor 594 antibody | ThermoFisher | A32744 |
| RTEL1 antibody | A kind gift from Michael Schertzer, I. Curie | |
| LSM14A antibody | Rb, Gmbh | |
| Histone 3 (H3) antibody | Cell Signalling Technologies | |
| **Oligonucleotides and other sequence-based reagents** | | |
| GGAGGAGAGAGCTTTTGGGCTC | | PIR-GM-SRS2-S |
| ACTCAGCAGCTGTTGTATTTCTG | | PIR-GM-SRS2-AS |
| GGTGGGAATCTTTCTTCTCTTCT | | PIR-GM-DNL4-S |
| CCTGTCTCCACTCAAACTATCC | | PIR-GM-DNL4-AS |

| Reagent/Resource | Reference or Source | Identifier or Catalog Number |
|---|---|---|
| ACGAAATCTGGGCTGCTTAC | | PIR-GM-RNR4-S |
| GGAACCCGTAGAAACTCTTACC | | PIR-GM-RNR4-AS |
| TAATACGACTCACTATAGGGGGGCAGAATAGGGCTCCACTCC | | PIR-GM-5UTR_T7-S |
| TGCACTTTCGATCCTTTTAATTTGTCT | | PIR-GM-5UTR-AS |
| TAATACGACTCACTATAGGGAAAATTAAACAACGGTGAAATCATAGTCAT | | PIR-GM-3UTR_T7-S |
| GTATAGAAAAAGAATATTCATTAAGCTATAATAACACACC | | PIR-GM-3UTR-AS |
| ATGTCTTTATCTTCAAAGTTGT | | PIR-GM-PGK1-S |
| GGTTGGCAAAGCAGC | | PIR-GM-PGK1-AS |
| /5Cy5/CACTGAGTCCAGCTCG AAACTTAGGAGG/3Cy5Sp/ | Pizzinga et al | FLAP X-Cy5 |
| AGGTGCTCCTTCAGCTTTTGGTATTCGGCGAG | | PIR-B-Lsm14a-del268-281-S |
| CTCGCCAATACCAAAAGCTGAAGGAGCACCT | | PIR-B-Lsm14a-del268-281-AS |
| GAATCCCACTTCGTCCAAACGAGTTTGCGGATTTTGAATA | | PIR-B-Lsm14a-del404-448-S |
| TATTCAAAATCCGCAAACTCGTTTGGACGAAGTGGGATTC | | PIR-B-Lsm14a-del404-448-AS |

**Oligos used for FISH**

| Probe sequence | Probe position |
|---|---|
| gttaaaaccttagtcttcccCCTCCTAAGTTTCGAGCTGGACTCAGTG | 118 |
| ttcatttcgttagcagctttCCTCCTAAGTTTCGAGCTGGACTCAGTG | 211 |
| actttccttgtgattgaggaCCTCCTAAGTTTCGAGCTGGACTCAGTG | 442 |
| aatagtccattcatctccatCCTCCTAAGTTTCGAGCTGGACTCAGTG | 485 |
| tctgtttcttgatcagcttcCCTCCTAAGTTTCGAGCTGGACTCAGTG | 510 |
| ggcgtcatgattcgaatctaCCTCCTAAGTTTCGAGCTGGACTCAGTG | 575 |
| tttttcttgcttagctctgaCCTCCTAAGTTTCGAGCTGGACTCAGTG | 625 |
| agatgatgatttcccttagcCCTCCTAAGTTTCGAGCTGGACTCAGTG | 784 |
| gatcaggatcacctacgatgCCTCCTAAGTTTCGAGCTGGACTCAGTG | 819 |
| agaaagttgtgcgctaaggcCCTCCTAAGTTTCGAGCTGGACTCAGTG | 862 |
| tgaataggtttggcaatgccCCTCCTAAGTTTCGAGCTGGACTCAGTG | 1119 |
| atgcactttcgatcctttaCCTCCTAAGTTTCGAGCTGGACTCAGTG | 1182 |
| atagggtatccgatgttctaCCTCCTAAGTTTCGAGCTGGACTCAGTG | 1205 |
| tggaatcccagaaactgtggCCTCCTAAGTTTCGAGCTGGACTCAGTG | 1239 |
| cttaatcttctcaccggtagCCTCCTAAGTTTCGAGCTGGACTCAGTG | 1364 |
| acacatcggtagctaacgtgCCTCCTAAGTTTCGAGCTGGACTCAGTG | 1395 |
| gtcggtatgtctagcattatCCTCCTAAGTTTCGAGCTGGACTCAGTG | 1453 |
| ctgaaagacccctagtaaaCCTCCTAAGTTTCGAGCTGGACTCAGTG | 1542 |
| aggttcggatttttcaagctCCTCCTAAGTTTCGAGCTGGACTCAGTG | 1640 |
| tcttttggagtgatgaccttCCTCCTAAGTTTCGAGCTGGACTCAGTG | 1801 |
| gctgcatcagaatgaagggaCCTCCTAAGTTTCGAGCTGGACTCAGTG | 1849 |
| ctttattggattccgactctCCTCCTAAGTTTCGAGCTGGACTCAGTG | 1878 |
| tgacaaacccattcttctcaCCTCCTAAGTTTCGAGCTGGACTCAGTG | 1914 |

| Reagent/Resource | Reference or Source | Identifier or Catalog Number |
|---|---|---|
| tttggcaccgtgaattgtagCCTCCTAAGTTTCGAGCTGGACTCAGTG | | 1940 |
| aaaactaccggccactcaagCCTCCTAAGTTTCGAGCTGGACTCAGTG | | 1963 |
| ggaattataccttcttcgcaCCTCCTAAGTTTCGAGCTGGACTCAGTG | | 1993 |
| tcttgatcttcttcttcgtcCCTCCTAAGTTTCGAGCTGGACTCAGTG | | 2053 |
| aggtattttgcacgagtctgCCTCCTAAGTTTCGAGCTGGACTCAGTG | | 2164 |
| atcgacatcttccacagttaCCTCCTAAGTTTCGAGCTGGACTCAGTG | | 2204 |
| aatcggcttgcaattcttggCCTCCTAAGTTTCGAGCTGGACTCAGTG | | 2227 |
| atcggacatggcttttatcaCCTCCTAAGTTTCGAGCTGGACTCAGTG | | 2261 |
| tggttatagtctttgcgcaaCCTCCTAAGTTTCGAGCTGGACTCAGTG | | 2383 |
| aaatcatcctttctctcctaCCTCCTAAGTTTCGAGCTGGACTCAGTG | | 2415 |
| attgattttggatgggcgtgCCTCCTAAGTTTCGAGCTGGACTCAGTG | | 2585 |
| gtaccttttttctggactttCCTCCTAAGTTTCGAGCTGGACTCAGTG | | 2633 |
| tattgaggtgcatacaccttCCTCCTAAGTTTCGAGCTGGACTCAGTG | | 2692 |
| gaatgaaactcctgcctactCCTCCTAAGTTTCGAGCTGGACTCAGTG | | 2734 |
| cctatcttctcttctcagaaCCTCCTAAGTTTCGAGCTGGACTCAGTG | | 2777 |
| tcgttgatgatcgtggtgatCCTCCTAAGTTTCGAGCTGGACTCAGTG | | 2811 |
| ttctgaaagttcctgcagttCCTCCTAAGTTTCGAGCTGGACTCAGTG | | 2937 |
| ggaggatgcagttcatcaacCCTCCTAAGTTTCGAGCTGGACTCAGTG | | 2974 |
| ggctgatcagaattactgcaCCTCCTAAGTTTCGAGCTGGACTCAGTG | | 3058 |
| cctcctttctattgatatgtCCTCCTAAGTTTCGAGCTGGACTCAGTG | | 3140 |
| caaatctatcacttcctccaCCTCCTAAGTTTCGAGCTGGACTCAGTG | | 3245 |
| aactcagcagctgttgtattCCTCCTAAGTTTCGAGCTGGACTCAGTG | | 3295 |
| ctgtgtctttttcattgctgCCTCCTAAGTTTCGAGCTGGACTCAGTG | | 3386 |
| gacttgatgcaggttcattcCCTCCTAAGTTTCGAGCTGGACTCAGTG | | 3426 |
| ctttttttttcgcacgtgacaCCTCCTAAGTTTCGAGCTGGACTCAGTG | | 3467 |
| **Chemicals, Enzymes and other reagents** | | |
| Hydroxyurea | MedChem Express | HY-B0313 |
| Methyla methanesulphonate | Sigma Aldrich | 129925 |
| Zeocin | ThermoFisher Scientific | R25001 |
| Cisplatin | Sigma Aldrich | 232120 |
| Cyclohehimide | Sigma Aldrich | 01810 |
| RiboLock RNase Inhibitor | Invitrogen | EO0381 |
| GFP-TRAP Magnetic agarose beads | Chromotek | gtma |
| glutathione Sepharose beads | GE Healthcare | 17075605 |
| Ni-NTA agarose | G Biosciences | 786-940 |
| Flag beads | Sigma Aldrich | A2220 |
| Coomassie brilliant blue | USB Chemicals | 32826 |
| S-adenosyl methionine | New England Biolabs | B9003S |
| T7 polymerase | Invitrogen | EP0111 |
| DNAse | Invitrogen | EN0521 |
| Fluoromount-G™ Mounting Medium | Invitrogen | 00-4958-02 |
| Trizol | G Biosciences | 786-652 |
| TRIzol LS reagent | Invitrogen | 10296010 |
| TB Green Premix Ex Taq II | TaKaRa | RR820B |

| Reagent/Resource | Reference or Source | Identifier or Catalog Number |
|---|---|---|
| Illumina TruSeq Stranded mRNA Library preparation kit | Illumina | |
| KAPA library quantification kit | Roche | |
| Lipofectamine RNAiMAX Reagent | Life Technologies | |
| siRNA LSM14A | Dharmacon | |
| **Software** | | |
| ImageJ-win64 | | |
| GraphPad Prism Version 8.0 | | |
| softWoRx 3.5.1 software | | |
| Institut Curie RNA-seq Nextflow pipeline (v3.1.4) | | |
| **Other** | | |
| NovaSeq 6000 instrument | Illumina | |

## Yeast strains and transformation

All plasmids and strains used in this study are listed in reagents and tools table. For transformation, strains were grown to 0.6 $OD_{600}$ in yeast extract and peptone supplemented with 2% glucose (YEPD), and pelleted down. The cells were washed once with water, followed by a single wash with 100 mM Lithium Acetate (LiAc). The cells were resuspended in 100 mM LiAc and aliquoted into 50 μL fractions. The cell suspension was then layered with 240 μL of 50% PEG (v/v), 36 μL of 1 M LiAc, 25 μL of salmon sperm DNA (100 mg/ml) and 100 ng of the respective plasmid DNA and vortexed. The mixture was then incubated at 30 °C for 30 min, followed by 15 min at 42 °C. The cells were then pelleted, resuspended in 100 μL water and plated on synthetic defined media and glucose agar plate. The plates were incubated at 30 °C for 2 days before colonies appeared.

## Drug treatments

Yeast cultures were grown in 10 ml of SD-Ura⁻ or of SD-Leu⁻ supplemented with 2% glucose, to 0.4–0.8 $OD_{600}$ and split into two equal parts. The fractions were treated with 0.03 % MMS, 200 mM HU, 150 μM cisplatin or 100 μg/ml Zeocin and vehicle control and incubated for 60 min (45 min for HU) at 30 °C with constant shaking. For UV treatment for live cell imaging, 500 μl of mid-log phase yeast cultures were evenly spread in a 6-well uncoated plate followed by UV exposure of 50 J/m² energy using stratalinker. The plate was instantly covered, culture transferred to aluminum-foiled centrifuge tube and spotted on coverslip in dark for imaging.

For CHX treatment, after incubation in HU, cells were treated with 0.1 mg/ml CHX solution (dissolved in methanol) or methanol (vehicle control). Cells were kept in the shaker for 5 min, followed by pelleting and live cell imaging.

## Cell imaging

In all cases for yeast live cell imaging, after the incubation period, the cells were immediately harvested (14,000 rpm, 15 s), resuspended in 20 μL of media and spotted on a glass coverslip (No. 1) and observed using live cell imaging. Yeast images were acquired using a Deltavision RT microscope system running softWoRx 3.5.1 software (Applied Precision, LLC), using an Olympus 100×, oil immersion 1.4 NA objective. The Green Fluorescent Protein (GFP) channel had 0.2 or 0.5 s of exposure and 32% or 50% transmittance respectively. A minimum of 80–100 cells were observed for each experiment. Quantification was done as granules per cell. Percentage granular intensity of Scd6 was calculated by using the ratio of GFP fluorescence intensity in granules by total cellular GFP fluorescence intensity using ImajeJ software. Statistical analysis was done using GraphPad Prism Version 8.0.

For localization experiment with ectopically expressed LSM14A-GFP and LSM14AΔRGG1ΔRGG2-GFP, cells were seeded on coverslips pre-coated with 1 μg/ml fibronectin (Sigma) and 20 μg/ml collagen (Sigma) 2 days before drug treatment. After reaching confluency, cells were treated with 10 mM hydroxyurea for 30 min. After the treatment, the coverslips were washed twice in PBS, and allowed to dry for 2–3 min. The cells were then fixed with 4% PFA in PBS for 10 min, followed by permeabilization for 10 min at room temperature in PBS containing 0.1% Tween-20 (PBST). Cells were washed with PBST, followed by 1X PBS. Cells were dried onto the cover slip before adding mounting media containing DAPI. This was followed by fixation onto a glass slide for confocal microscopy (Leica SP8).

## Western blotting

SDS Polyacrylamide gels were transferred onto Immobilon-P Transfer Membrane® (MERCK) using Transfer-Blot® Semi Dry apparatus (BIO-RAD). The transfer was done at 10 V for 1 h. The membrane was then blocked with 5% skimmed milk, followed by washing and incubation with specific antibodies. For re-probing the blot with more than one antibody, the blot was stripped, blocked, and reprobed. All antibodies used in the study are listed in reagents and tools table.

For RTEL detection, A2058 (500,000) cells were transfected with 25 nM siLSM14A (Dharmacon siRNA pool) for 48 h followed by 10 mM HU treatment for 2 h. Post treatment, the cells were scraped and lysed using RIPA lysis buffer followed by 10s ON/OFF sonication for 10 cycles (Bioruptor). 30 μg of protein was loaded after BCA assay (Thermo) in 4–20% gradient gel (Invitrogen) and migrated in MOPS buffer at constant voltage of 150 V for 1 h. Semi

dry transfer (iBlot2) was performed followed by 1 h blocking in 5% milk in TBS-T (0.1%). Secondary rabbit antibodies were used (1:3000) and incubated for 1 h at RT followed by 3 × 5 min washes before being developed using Clarity ECL (bio-rad) solution in Western blot developing machine (Vilber).

## Plating assay and spot assay

For spot assay, freshly grown yeast cells from the agar plate was resuspended in autoclaved deionized water and serially diluted to 10-fold dilutions after normalizing the first dilution to OD600 = 1. 10 µl of each dilution was spotted on an agar plate and incubated at 30 °C for 48–60 h before imaging. For the plating assay, 100 µl of the 4th dilution, i.e., $10^{-4}$ OD$_{600}$, was plated on a Control or drug-containing agar plate (100 mM HU, 0.01% MMS, 150 µg/ml Zeocin) or plated cells were exposed to 30 J/m$^2$. The plates were then incubated at 30 °C for 48–60 h before counting the colony-forming units (CFUs).

## RNA immunoprecipitation

300 ml of yeast cells expressing GFP-tagged Scd6, or Empty Vector (EV), were grown till 0.8 OD$_{600}$ in leucine drop-out synthetic defined media. The Scd6-GFP culture was then split into two, where one was treated with 200 mM HU and the other kept as an untreated control. The cultures were then pelleted and lysed in Lysis buffer [10 mM Tris, 150 mM NaCl, 1 X EDTA-free Protease inhibitor complex, RiboLock RNase Inhibitor, (Catalog No. EO0381)]) by bead beating. The lysate was spun at 5500 rpm for 5 min. The supernatant was then transferred into a fresh microcentrifuge tube, and a small aliquot was removed for the isolation of total RNA. The rest of the lysate was diluted 1:1 with the lysis buffer, and 10 µl of GFP-TRAP Magnetic agarose beads (Cat. No. gtma) were added to a final volume of 1 ml. The pull-down samples were nutated at 4 °C for 90 min. The beads were then separated using a magnetic stand, followed by two washes with the lysis buffer. Finally, the beads were resuspended in 120 µl of lysis buffer. The pull-down efficiency was calculated by western blotting and probing the blot with an anti-GFP antibody (1:5000 dilution, Biolegend)). The rest of the pull-down was used for RNA isolation.

To quantify relative mRNA enrichment, ΔCt values for each gene tested were calculated by subtracting Ct value of the PGK1 primer (internal control) from the test primer. ΔΔCt was calculated by subtracting ΔCt of total RNA from ΔCt of pull-down RNA. The final values were then obtained by normalizing the $2^{(-\Delta\Delta Ct)}$ with Scd6 pull-down intensities.

## Glutathione S transferase (GST) pull-down

100 ml of yeast cells expressing GST-tagged Scd6, or Empty Vector (EV), were grown till 0.8 OD$_{600}$ in uracil drop-out synthetic defined media. The Scd6-GST culture was then split into two, where one was treated with 200 mM HU and the other kept as an untreated control. The cultures were pelleted and lysed in Lysis buffer [10 mM Tris, 150 mM NaCl, 0.5% NP40, 1 X EDTA-free Protease inhibitor complex] by bead beating. The lysate was spun at 5500 rpm for 5 min. The supernatant was then transferred into a fresh microcentrifuge tube. The lysate was diluted 1:1 with the dilution buffer (lysis buffer without NP40), 40 µl was aliquoted separately as input and 40 µl of glutathione

sepharose (GE, catalog no. 17075605) were added to a final volume of 1 ml. The pull-down samples were nutated at 4 °C for 90 min. The beads were washed thrice with lysis buffer at room temperature for 10 min. The beads were finally resuspended in 100 µl of lysis buffer. Following pull down, western analysis was performed by cutting the blot and using anti-GST (CST, catalog no. 2624; 1:1000 dilution) and anti-eIF4G1 (Cocalico Biologicals; 1:1000 dilution) antibodies to probe respective blot pieces.

## Protein purification and in vitro methylation

Recombinant N terminal His-tagged, and C terminal Flag-tagged Scd6 WT (His-Scd6-Flag) and domain deletion mutants were purified from *Escherichia coli* according to standard protocols using Ni-NTA agarose (G Biosciences Cat No. 786-940) This was followed by a second purification with Flag beads (Sigma Aldrich Cat No. A2220) using standard purification protocols. N terminal His-tagged Hmt1 was purified using Ni-NTA agarose beads (G Biosciences Cat No. 786-940). Purified proteins were dialyzed with 20 mM Tris-Cl pH 7.5, 100 mM NaCl, 10% glycerol and 1 mM DTT and purification was confirmed by SDS PAGE and Coomassie brilliant blue staining (Cat No. 32826). Purified His-Scd6-FLAG was methylated by taking 1:1 molar ratio of the protein and purified His-Hmt1 in methylation buffer containing 100 mM Tris–Cl pH 8, 200 mM NaCl, 2 mM ethylenediaminetetraacetic acid (EDTA) and 1 mM Dithiothreitol (DTT) with or without 20 µM S-adenosyl methionine (SAM) (New England Biolabs; catalog no. B9003S). The reaction was incubated for 2 h at 37 °C and used for downstream experiments. The methylation was confirmed by SDS-PAGE followed by western blotting using mono methyl arginine (MMA) antibody (Cell Signaling Technology, catalog no. 8711; 1:1000 dilution).

## In vitro transcription (IVT) and electromobility shift assay (EMSA)

The 5' fragment of the SRS2 gene coding 184 bp of 5'UTR and 16 bases downstream the first base of the ORF (200-mer template) was amplified from *BY4741* genomic DNA by PCR. The 3' fragment of the SRS2 gene coding 163 bp of 3'UTR and 37 bases upstream of the last base of the ORF (200-mer template) was amplified from *BY4741* genomic DNA by PCR. 100 ng of PCR amplified, and column purified DNA templates were transcribed using T7 polymerase (Thermo Cat ID. EP0111) in 100 µl reaction at 37 °C for 2 h using manufacturer's protocol followed by DNAse treatment (Thermo Scientific Cat ID: EN0521). The RNA was then purified using phenol-chloroform and stored in multiple aliquots for later use.

For EMSA, to compare the binding of 5'UTR and 3'UTR, increasing concentrations of purified His-Scd6-Flag (0.26, 0.5, 0.80, 1.1 µM) were incubated with 1.7 µM in vitro transcribed RNA fragments. For EMSA, to compare the RNA binding activity of methylated vs unmethylated Scd6, increasing concentrations of purified His-Scd6-Flag (0.7, 1.4, 2.1, 2.8, 3.4, 4.1 µM; unmethylated or methylated) were incubated with 0.94 µM of in vitro transcribed RNA. The EMSAs were performed in binding buffer [10 mM Tris (pH 8), 50 mM NaCl, 0.05% NP-40, 6% glycerol, 1 mM DTT, 2 µg/ml BSA and 0.5 µg/ml tRNA] at 30 °C for 30 min. The reaction was then loaded onto 2.5% Agarose gel in 1 X MOPS buffer and the mobility shift visualized using ethidium bromide staining.

## Single-molecule fluorescence in situ hybridization (smFISH)

smFISH protocol was adapted and modified from Tsanov et al (2016) and Pizzinga et al (2019). Briefly, WT and Δsrs2 strains transformed with Scd6-GFP plasmid were grown to mid-log phase (OD$_{600}$ 0.5–0.8) in leucine drop-out synthetic defined media and fixed with 3% paraformaldehyde for 45 min, at room temperature, in dark. To make smFISH probes, 200 pmol of an equimolar mix of SRS2-specific oligos (48 oligo pool, IDT; Reagents and tools table) was annealed with 250 pmol of Cy5 labeled X-flap oligo in 1× NEBuffer 3. After fixation, cells were washed with buffer B (1.2 M sorbitol and 100 mM KHPO4, pH 7.5), followed by resuspension in spheroplasting buffer (Buffer B, 2 mM Ribonucleoside Vanadyl Complex, 0.2% β-mercaptoethanol, and 1 mg/ml lyticase) and incubated at 37 °C shaker incubator for 15 min and stored in 70% ethanol at −20 °C. Subsequently, cells were hybridized with 40 pmol of smFISH probes in 100 μl of hybridization buffer (10 mg E. coli tRNA, 0.2 mM Ribonucleoside Vanadyl Complex, 200 μg/ml BSA, 10% dextran sulfate, 10% formamide, and 2× SSC). Cells were then washed in 10% formamide and 2× SSC. The washed cells were resuspended in PBS and mixed with the mounting agent (Fluoromount-G™ Mounting Medium Cat ID. 00-4958-02) with DAPI, and spotted on the slide. The images were acquired using a Deltavision RT microscope system running softWoRx 3.5.1 software (Applied Precision, LLC), using an Olympus 100×, oil immersion 1.4 NA objective at required wavelengths. The images were processed and quantified using ImageJ software. Percentage overlap of SRS2 mRNA foci with Scd6 foci was calculated by measuring the area of both the foci, and the area of overlap using ImageJ software and dividing the overalapping area with SRS2 foci area. The final fraction overlap was then converted into percentages and plotted as a violin plot using GraphPad Prism 8.

## Cell culture

The A2058 and A375 melanoma cell lines used in this study were purchased from the ATCC. Cancer cell lines were maintained at 37 °C and 5% CO$_2$ in a humidified atmosphere and grown in DMEM:F12 or MEM growth media supplemented with 10% FBS, 2 mM glutamine, 50 μ ml$^{-1}$ penicillin and 50 mg ml$^{-1}$ streptomycin (Gibco).

## Polysome fractionation

WT cells expressing either EV or Scd6-GFP were grown till OD$_{600}$ 0.8 and treated with 200 mM HU for 60 min, followed by 30 min of 0.1 mg/ml CHX treatment. The cells were lysed in lysis buffer (10 mM Tris pH 7.4, 100 mM NaCl, 30 mM MgCl$_2$, RNase inhibitor, cycloheximide and EDTA-free protease inhibitor complex), and pre-cleared lysate was loaded onto a 10–50% sucrose density gradient and centrifuged in an SW41 rotor (Beckman-Coulter) at 39,000 r.p.m. for 2 h at 4 °C. The untranslated (till 80S fraction) and translated (polysome fractions) fractions were collected and pooled (Biocomp), followed by RNA isolation from both fractions by TRIzol–chloroform method. Total RNA was also isolated from lysate. RNA samples were then treated with DNase followed by DNA library preparation using random primers. RT-qPCR were performed using gene-specific primers.

Quantification of mRNA enrichment in the polysome fractions (translated fraction) was calculated w.r.t. mRNA in untranslated fractions. ΔCt values for each gene tested were calculated by subtracting Ct value of the PGK1 primer (internal control) from the test primer. ΔΔCt was calculated by subtracting ΔCt of total RNA from ΔCt of translated or untranslated fractions. The final values were then obtained by normalizing the 2$^{(−ΔΔCt)}$ of translated by untranslated polysome fractions.

To understand the translation control effect of LSM14A, A2058 cells were transfected with scrambled siRNA or siRNA specific to LSM14A cells. The cells were stressed with 10 mM hydroxyurea for 2 h, followed by lysis, and polysome profiling. Sucrose density gradient centrifugation was used to separate the sub-polysomal and the polysomal ribosome fractions. Fifteen minutes before collection, cells were incubated at 37 °C with 100 mg/ml CHX added to the culture medium. Next, cells were washed, scraped into ice-cold PBS supplemented with 100 mg/ml CHX, centrifuged at 3000 r.p.m. for 5 min and then collected into 400 ml of LSB buffer (20 mM Tris, pH 7.4, 100 mM NaCl, 3 mM MgCl$_2$, 0.5 M sucrose, 2.4% Triton X-100, 1 mM DTT, 100 U ml RNasin and 100 mg/ml cycloheximide). After homogenization, 400 ml LSB buffer supplemented with 0.2% Triton X-100 and 0.25 M sucrose was added. Samples were centrifuged at 12,000 × g for 10 min at 4 °C. The lysates were loaded onto a 15–50% sucrose density gradient and centrifuged in an SW41 rotor at 38,000 r.p.m. for 2 h at 4 °C. Polysome fractions were monitored and collected using a gradient fractionation system (Isco). Total RNA was extracted from the four heaviest fractions and the input samples using the TRIzol–chloroform method.

## RNA isolation, cDNA preparation, and RT-qPCR

RNA isolation from yeast samples were done using Trizol (G Biosciences, Cat ID 786-652)- chloroform method. Briefly, the lysate/PD fractions were treated with 1 ml Trizol and 200 μl chloroform. After vortexing, the mixture was spun at 14,000 rpm for 20 min, followed by Isopropanol precipitation by flash freezing. It was spun at 15000 rpm at 4 °C for 30 min, followed by one 70% ethanol wash. The pellets were dried and resuspended in the required amount of Diethyl pyrocarbonate (DEPC)-treated water. The RNA was then treated with DNase (Thermo Scientific Cat ID: EN0521), followed by cDNA library preparation with 1 μg of RNA. OligodT was used for creating the libraries. The libraries were then diluted (1:5 for lysates; 1:1 for pull down and polysome fractions) and used as templates for RT-qPCR in 10 μl reaction volume, 40 cycles (TB Green Premix Ex Taq II (Tli RNase H Plus); TaKaRa, Cat ID RR820B). Gene-specific primers (Bioserve, India) were used in the PCR.

For mammalian samples, RNA from the same volume (300 μl) of each of the polysome fractions (17 fractions) was isolated using TRIzol LS reagent (TRIzol™ LS Reagent, Invitrogen, Cat ID 10296010) using the manufacturer's protocol. The cDNAs from each fraction were diluted to 1:10 times from which equal volume (2 μl) cDNA were used along with SYBR green (TB Green Premix Ex Taq II (Tli RNase H Plus); TaKaRa, Cat ID RR820B) and primers (Bioserve, India) for validation of mRNA levels using quantitative real-time PCR (BioRad CFX96 Touch Real-Time PCR Detection System). The % of RNA in each fraction was calculated as described in Panda et al (2017).

## RNA sequencing and bioinformatic analysis

RNA sequencing libraries were prepared from 500 ng to 1 µg of total RNA or mRNA-enriched from heavy polysome fractions (13 to 17) using the Illumina TruSeq Stranded mRNA Library preparation kit which allows to perform a strand of strand-specific sequencing. Nanodrop spectrophotometer was used to assess purity of RNA based on absorbance ratios (260/280 and 260/230) and BioAnalyzer for RNA integrity (RIN>9). A first step of polyA+ selection using magnetic beads is done to focus sequencing on polyadenylated transcripts. After fragmentation, cDNA synthesis was performed and resulting fragments were used for dA-tailing followed by ligation of TruSeq indexed adapters. PCR amplification was finally achieved to generate the final barcoded cDNA libraries. The libraries were equimolarly pooled and subjected to qPCR quantification using the KAPA library quantification kit (Roche). Sequencing was carried out on the NovaSeq 6000 instrument from Illumina based on a 2*100 cycle mode (paired-end reads, 100 bases) to obtain around 30 million clusters (60 million raw paired-end reads) per sample. Finally, Fastq files were generated from raw sequencing data using bcl2fastq pipeline performing data demultiplexing based on index sequences.

RNA-seq data were analyzed with the Institut Curie RNA-seq Nextflow pipeline (v3.1.4). Briefly, raw reads were first trimmed with Trimgalor. Reads were aligned on the human reference genome (hg19) using STAR (STAR_2.6.1a_08-27). Genes abundances were then estimated using STAR, and the Gencode v34 annotation. The read counts on RNAs extracted from polysome fractions (13–17) depend on both mRNA abundance and their translation rates. Differential translation can be characterized by the differences between the changes in total mRNA levels and the levels of mRNAs engaged in ribosomes (heavy polysome fractions). Differential analysis between conditions were done using the R package Xtail (Xiao et al, 2016) on protein-coding genes only (Dataset EV1).

## RNA interference

Cells were transfected with 20 nM of each siRNA against LSM14A (Dharmacon) using Lipofectamine RNAiMAX Reagent (Life Technologies) following the supplier's instructions.

## Random plasmid integration assay

A375 cells were seeded in a 6 cm dish and transfected with siRNAs (30 nM) the following day (day 2). At day 3, the cells were transfected once more with siRNAs and, later at day 3, the cells were treated with or without 10 mM Hydroxyurea for 30 min and subsequently transfected with with 2 µg gel-purified BamHI-EcoRI-linearized pEGFP-C1 plasmid for 48h.

On day 5, cells were collected, counted, seeded, and grown in medium lacking or containing 0.5 mg/mL G418. Transfection efficiency of the linearized pEGFP-C1 was determined by flow cytometry. The cells were incubated at 37 °C to allow colony formation for 2 weeks by refreshing media every 3–4 days. The cells were then stained with a 20% ethanol solution containing 0.5% crystal violet. Random plasmid integration events (number of G418-resistant colonies) were normalized by the plating efficiency (number of colonies without G418) and by transfection efficiency.

## NHEJ suicide deletion assay

The assay was adapted and modified from Karathanasis and Wilson, 2002. Briefly, the suicide deletion strain YW714 (a kind gift from Professor Thomas Wilson) was transformed with either empty vector or Scd6-GFP expressed on a 2µ plasmid. Stationary phase cells were plated on synthetically defined leu⁻ ade⁺, leu⁻ ade⁻ or leu⁻ ade⁻ drop out agar plates supplemented with 100 mM HU and either 2% sucrose or a combination of 1% sucrose + 1% galactose as carbon source and to induce double-strand breaks. The plates were incubated at 30 °C for 3–4 days till visible colonies appeared. Colonies were counted from all the plates and plotted as a ratio of HU treated and untreated CFUs normalized to empty vector.

## Data availability

The datasets generated in this study have been deposited in the Gene Expression Omnibus repository (GEO) under accession numbers GSE274294.

The source data of this paper are collected in the following database record: biostudies:S-SCDT-10_1038-S44319-025-00443-3.

## Peer review information

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

## Acknowledgements

The authors thank Vagner and Rajyaguru labs for their critical inputs and suggestions during this work. We are grateful to Thomas Wilson for providing us with the *S. cerevisiae* suicide deletion strains. We would also like to thank

Martin Kupiec for kindly gifting pRS316-SRS2 plasmid. The authors thank Won-Ki Huh (Seoul National University) for providing us with the *S. cerevisiae* Scd6-myc strain. We thank Mark Ashe for helping us with the FISH experiment. We are grateful to K Muniyappa for providing crucial resources. We thank Poornima Gopalakrishna for creating the RGG-deletion mutants of LSM14A. This work was supported by the following: CEFIPRA grant for SV and PIR (7003-H/7003-2). PIR acknowledges support from India Alliance DBT-Wellcome trust (IA/I/12/2/500625) and BT/PR51975/BMS/85/23/2024 from the Department of Biotechnology to PIR and the DST-FIST program to the Department of Biochemistry. GM and SL were supported by a fellowship from MHRD, India. RR was supported by a fellowship from DBT, India. AB was supported by DST fellowship. This work was supported by grants from Institut Curie, Gustave Roussy, INSERM, CNRS and Equipe labellisée Ligue Nationale Contre le Cancer (LNCC) (to SV). High-throughput sequencing was performed by the ICGex NGS platform of the Institut Curie (Virginie Raynal and Sylvain Baulande) supported by the grants ANR-10-EQPX-03 (Equipex) and ANR-10-INBS-09-08 (France Génomique Consortium) from the Agence Nationale de la Recherche ("Investissements d'Avenir" program), by the ITMO-Cancer Aviesan (Plan Cancer III) and by the SiRIC-Curie program (SiRIC Grant INCa-DGOS-465 and INCa-DGOS- Inserm_12554). Data management, quality control and primary analysis were performed by the Bioinformatics platform of the Institut Curie.

## Author contributions

**Gayatri Mohanan**: Data curation; Formal analysis; Validation; Investigation; Visualization; Methodology; Writing—original draft; Writing—review and editing. **Raju Roy**: Data curation; Investigation; Methodology. **Hélène Malka-Mahieu**: Data curation; Investigation; Methodology. **Swati Lamba**: Formal analysis; Investigation. **Lucilla Fabbri**: Data curation; Formal analysis; Investigation. **Sidhant Kalia**: Formal analysis; Investigation. **Anusmita Biswas**: Formal analysis. **Sylvain Martineau**: Formal analysis; Investigation. **Céline M Labbé**: Formal analysis; Investigation. **Stéphan Vagner**: Conceptualization; Resources; Data curation; Formal analysis; Supervision; Funding acquisition; Investigation; Visualization; Writing—review and editing. **Purusharth I Rajyaguru**: Conceptualization; Resources; Data curation; Formal analysis; Supervision; Funding acquisition; Investigation; Visualization; Methodology; Writing—original draft; Project administration; Writing—review and editing.

Source data underlying figure panels in this paper may have individual authorship assigned. Where available, figure panel/source data authorship is listed in the following database record: biostudies:S-SCDT-10_1038-S44319-025-00443-3.

## Disclosure and competing interests statement

The authors declare no competing interests.

# Expanded View Figures

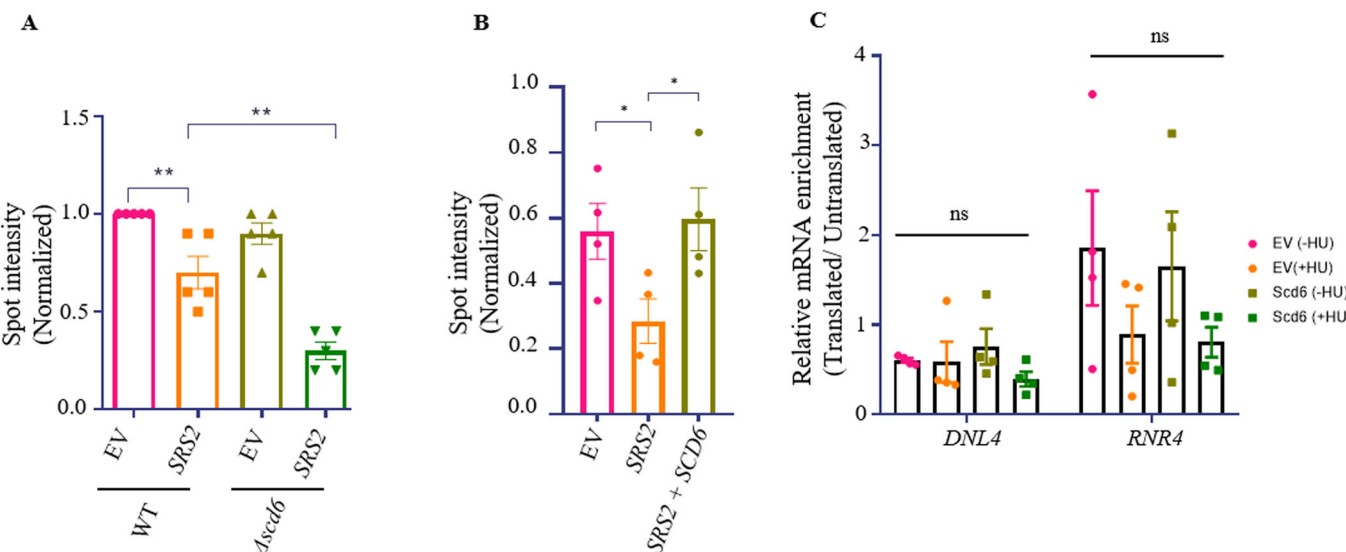

**Figure EV1. Quantitiation of the SRS2-SCD6 genetic interaction.**

(A) Quantification of growth assay (in Fig. 2A) by measuring the intensity of the second spot and normalizing the values of spots on HU plate to the respective control values for each strain ($n = 5$ biological replicates). Error bars indicate standard error of mean and statistical significance was calculated using unpaired t-test. WT EV vs WT SRS2, **$p = 0.0071$; WT SRS2 vs Δscd6 SRS2, **$p = 0.0029$. (B) Quantification of growth assay (in Fig. 2C) by measuring the intensity of the third spot and normalizing the values of spots on HU plate to the respective control values for each strain ($n = 4$ biological replicates). Error bars indicate standard error of mean and statistical significance was calculated using unpaired t-test. EV vs SRS2, *$p = 0.0448$; SRS2 vs SRS2+Scd6, *$p = 0.0382$. (C) Quantification of *DNL4* and *RNR4* mRNA in the polysome fractions plotted as relative $\log_2$-Fold change ratio of Translated/Untranslated fractions ($n = 4$ biological replicates) using gene-specific primers. Error bars indicate standard error of mean and statistical significance was calculated using unpaired t-test.

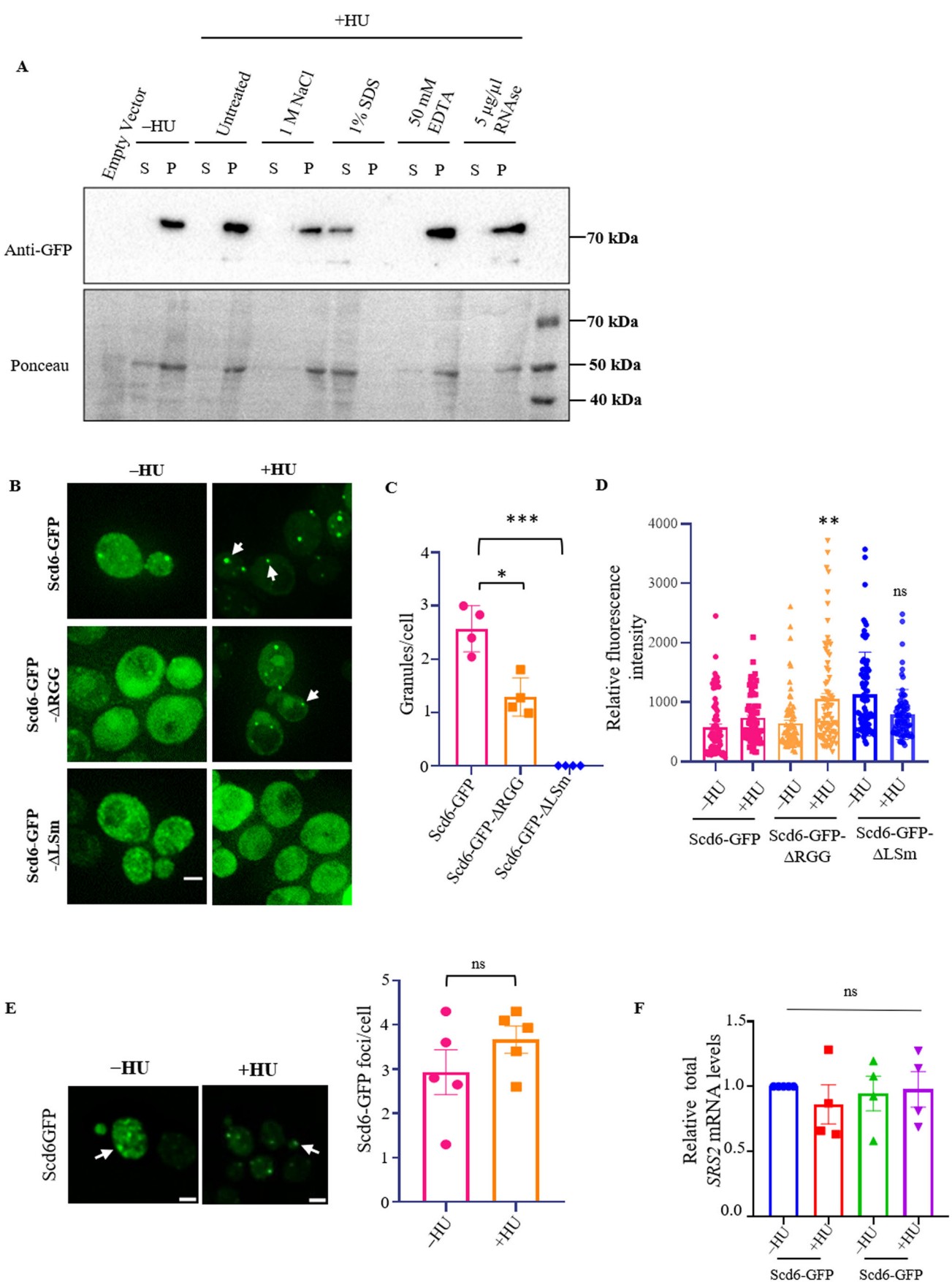

◀  **Figure EV2.  Scd6 localizes to higher order mRNP condensates upon HU stress in LSm and RGG-motif dependent manner.**

(A) Western blot showing partitioning of granule enriched Scd6GFP into soluble (S) or pellet (P) fraction upon treatment with 1M NaCl, 1% SDS, 50 mM EDTA or 5 μg/μl RNAse. (B) Live cell imaging showing localization of Scd6 and its domain deletion mutants upon HU treatment. White arrows indicate cytoplasmic foci. Scale bar = 2 μm. (C) Quantification of number of granules per cell shown in (B) ($n = 4$ biological replicates). Statistical significance was calculated using an unpaired t-test. Error bars indicate the standard error of the mean. Scd6-GFP vs Scd6GFP-ΔRGG, *$p = 0.0039$; Scd6-GFP vs Scd6GFP-ΔLSm ***$p = 0.0001$. (D) Quantification of relative GFP intensity as a measure of Scd6-GFP and mutants protein expression ($n = 3$ biological replicates, ≥70 cells were counted). Statistical significance was calculated using Tukey's multiple comparisons test. Error bars indicate the standard error of the mean. Scd6GFP-ΔLSm (−HU) vs Scd6GFP-ΔLSm (+HU), **$p = 0.0037$. (E) Live cell imaging showing granular localization of Scd6-GFP expressed on a 2 μ plasmid (left panel) and quantification of granule per cell localization of Scd6GFP (right panel) ($n = 5$ biological replicates, ≥100 cells counted). White arrows indicate cytoplasmic foci. Scale bar = 2 μm. (F) Quantification of relative total *SRS2* mRNA levels from lysate (left panel), and relative enrichment of Scd6 protein and *SRS2* mRNA in soluble (supernatant) and the heavier (granule-enriched fraction) fraction (right panel) (related to Fig. 3F) $n = 4$ biological replicates. Statistical significance was calculated using an unpaired t-test. Error bars indicate standard error of mean.

**A**

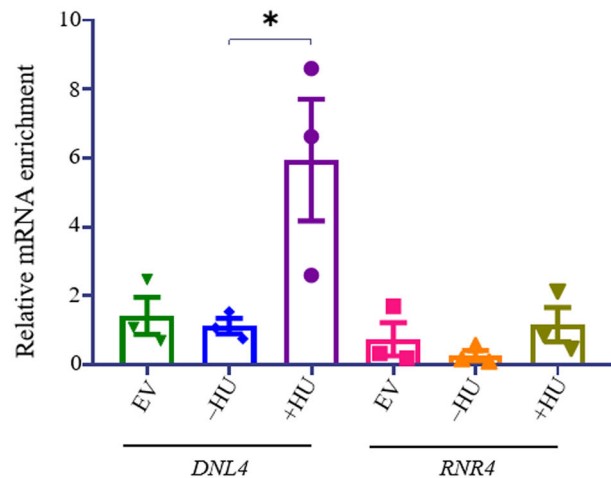

**B**

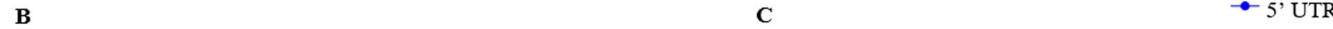

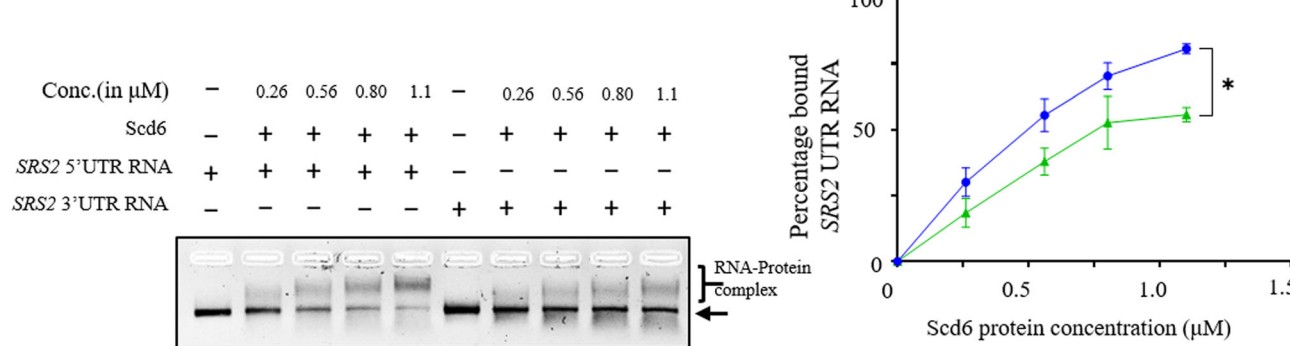

**C**

**D**

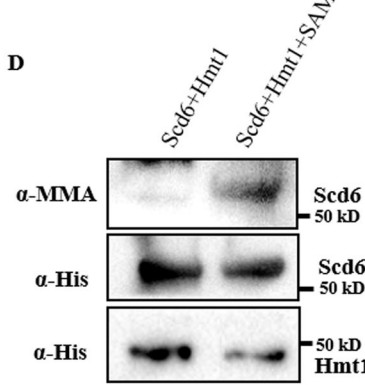

**Figure EV3.  Scd6 binds to specific mRNAs.**

(A) Quantification of mRNA enrichment in RNA immunoprecipitation with *DNL4* and *RNR4* specific primers ($n = 3$ biological replicates). Statistical significance was calculated using unpaired t-test. Error bars indicate standard error of mean. *$p = 0.0351$. (B) Ethidium Bromide (EtBr) stained agarose gel for electromobility shift assay (EMSA) with increasing concentrations of recombinant Scd6 incubated with 200-mer 5′UTR or 3′ UTR fragment of *SRS2 mRNA* (1.7 μM RNA) and its quantification in (C) ($n = 3$ technical replicates) Black arrow denotes unbound RNA. Statistical significance was calculated using a paired t-test. Error bars indicate standard error of mean. *$p = 0.0261$. (D) Western blot showing methylation of purified His-Scd6-Flag in the presence of His-Hmt1 and 1 mM S-adenosyl methionine (SAM).

