## [Peer Review File · EMBO Reports]

Genotoxic stress triggers Scd6-dependent regulation of translation to modulate the DNA damage response

Gayatri Mohanan, Raju Roy, Helene Malka-Mahieu, Swati Lamba, Lucilla Fabbri, Sidhant Kalia, Anusmita Biswas, Sylvain Martineau, Celine M. Labbe, Stephan Vagner, and Purusharth Rajyaguru

Corresponding author(s): Purusharth Rajyaguru (rajyaguru@iisc.ac.in) , Stephan Vagner (Stephan.Vagner@curie.fr)

Review Timeline:

Transfer Date:	17th Nov 24
Editorial Decision:	22nd Nov 24
Appeal Received:	4th Feb 25
Editorial Decision:	4th Feb 25
Revision Received:	8th Feb 25
Editorial Decision:	28th Feb 25
Revision Received:	25th Mar 25
Accepted:	27th Mar 25

Editor: Achim Breiling

Transaction Report: This manuscript was transferred to EMBO reports following peer review at Review Commons.

**Review
COMMONS**

Review #1

1. Evidence, reproducibility and clarity:

Evidence, reproducibility and clarity (Required)

This manuscript demonstrates that RGG containing proteins (Sbp1, Scd6/LSM14A) localize to granules upon treatment with hydroxyurea, Scd6 binds and regulates SRS2 in a manner that is regulated by arginine methylation of the RGG motif, that the RGG and LSM motifs are crucial for the process, and that NHEJ is influenced by LSM14A, suggesting the functional significance of this regulation. Overall the data generally support the conclusions that are drawn. In terms of overall presentation, the manuscript is a bit of a dense read and some of the supplementary figures should be moved to the main body of the manuscript to reflect their importance to the story. I do believe the manuscript will have impact to field, particularly to the specialized RGG protein and stress response niche. I do have several suggestions to polish the manuscript/study:

Major Points:

1. Fig. 1 F/G: were the delta RGG and LSM variants expressed at an equivalent level to the WT protein in these experiments?
2. Fig. 3G: The 6 data points for the delta LSM variant are literally spread evenly up and down the graph, making these data appear highly questionable as to whether one can draw a definitive conclusion from them.

Minor Points:

1. Abstract: the acronym NHEJ likely will need to be defined for the general reader.
2. Introduction, first paragraph: change gene expression to 'transcription' in the phrase 'Even if the contribution of gene expression to GSR..' as I assume this is what is meant here. Gene expression consists of synthesis, processing, translation and decay.
3. Pg. 3 Introduction: Since they are liquid-liquid phase condensates and ribonucleoproteins (RNPs) refers to any protein-RNA interaction, I think that referring to PBs and SGs as mRNPs is a bit misleading (especially the 'major mRNPs').
4. Pg. 3 Introduction: are PBs truly 'sites' of mRNA decay as stated? There are papers in the literature that would argue otherwise.
5. Pg. 3, three lines from bottom. Change LSM14 to LSM14A
6. Pg. 4 top - What is an 'LCS' - containing protein? The acronym has not been defined

7. Fig. S1 - there are a lot of important data in this figure that demonstrate the coordinated movement of Scd6 and Sbp1 to granules. They should be moved into the main body of the manuscript in my opinion. Likewise, a whole section of the Results is dedicated to Fig. S2 - thus I would suggest moving these data into the main body of the manuscript to assist the reader.

8. Fig. 1F should be flipped in the figure with panel G since G is discussed in the results section before F

9. Be sure to define all acronyms for the reader.

10. Pg. 8/Fig. S3D/4A: It would be interesting to complete the story and determine the functional relationship of Scd6 to the DNL4 mRNA

11. Fig. 3H/I: It might be optimal to calculate and compare Kd's for the methylated and unmethylated variants. Also the labels at the top of 3H do not line up with the wells of the EMSA gel.

2. Significance:

Significance (Required)

Overall the data generally support the conclusions that are drawn. In terms of overall presentation, the manuscript is a bit of a dense read and some of the supplementary figures should be moved to the main body of the manuscript to reflect their importance to the story. I do believe the manuscript will have impact to field, particularly to the specialized RGG protein and stress response niche

3. How much time do you estimate the authors will need to complete the suggested revisions:

Estimated time to Complete Revisions (Required)

(Decision Recommendation)

Between 3 and 6 months

No

Review #2

1. Evidence, reproducibility and clarity:

Evidence, reproducibility and clarity (Required)

The authors describe a novel role for the RGG-box containing RNA-binding protein Scd6 and its human ortholog LSM14A in the genotoxic stress to HU treatment. The proteins accumulate in granules upon HU treatment and may be involved in the translational repression of mRNAs coding for relevant factors mediating the HU response. For instance, they show that yeast Scd6 interact with SRS2 mRNA and represses translation during HU response. Interestingly, they also show that interaction with RNAs is dependent on arginine methylation in the LSM domain, which greatly affects interaction with the RNA though specificity needs to be shown. At the end, the authors performed initial analysis with the human ortholog LSM14A, indicating that the cellular function of the protein is evolutionary conserved.

The study is interesting and adds new aspects for understanding RGG-containing proteins and their molecular functions. While it has been previously known that Scd6 accumulates in granules and acts as a translational repressor via eIF4G, this study reveals the regulation of a particular target (SRS2) in context of a relevant stress response. The experiments are usually well performed, though some controls and descriptions need to be added.

****Major points:****

1. Fig. 2A, B. While there seems to be an effect on the lag phase, it could be revealing if the authors pls. calculate the doubling times for the strains and treatments (taking through the exponential growth phase). Furthermore, it would be good if the authors can show the rescue of phenotypes for deletion strains (ie. reintroduction of respective gene on ARS-CEN based plasmids or (if not available) with the OE plasmids).
2. Fig. 3H. The authors tested the 5'UTR of SRS2 for interaction with recombinant Scd6. Firstly, it is unclear why the authors have chosen the 5'UTR for investigation? Can the authors explain. Secondly, the affinities are relatively low (μM) and the gel shift assay lacks a negative control. The authors should test an unrelated RNA fragment of approximately the same size to control for specificity (negative control). It is unclear whether the protein could interact with any RNA-fragment through charged RNA backbone. Thirdly, it would be good if the authors could show a Coomassie gel for the recombinant protein used in those assays.
3. The Materials and Methods section lacks important information and requires further

details to evaluate the study (see below 10 - 17).

****Minor points:****

Results:

4. The numbering of Figure S1, S2 is confused in the first part of the results section. The authors should check numbering. In general, numbering should follow in the order of the text - pls. check.

5. Pg. 5. CHX treatment leads to a decrease in Scd6-GFP and SBP-1 GFP granules. Essentially, CHX blocks translation elongation so the result indicates that puncta depend on active translation. The authors may want to add this liaising point towards the claim that mRNAs could be present in those puncta. How this results integrates with data shown in Fig. 5S5.

6. Fig. 2H. It would be helpful to the reader, if the authors could mark the respective fraction in the polysomes taken for analysis of relative enrichments. How was this relative enrichment was calculated needs further description.

7. Fig. S5B. 1% SDS treatment cause absence for Scd6 signal from the pellet fraction. Based on this result, I am not clear how based on this result they can claim for presence of higher order mRNA-protein complexes? Why does it exclude the possibility for Scd6 aggregates accumulating in the pellet? The authors need to explain/ modify this statement. Related to earlier findings that showed dependency of puncta upon CHX treatment, one wonders how this result matches to this earlier observation (ie. EDTA should disassemble ribosomes)? Can the authors explain?

8. Fig. 5E, F. For the RNA-seq, the authors compared polysomes with free RNAs (up to 80S) and found enrichment of LIG4 and RTEL1. However, the polysomal profiling mainly shows a slight shift of those mRNAs in higher polysomes; while there is no difference compared to free fractions. How can this be explained?

On the line, the authors should indicate clearly what fractions were pooled for RNA seq analysis. It is also not clear how the authors quantified percentage of RNA in individual fractions (have they spiked-in an RNA?)

- this needs to be stated in the M&M section.

9. At the end, it may be beneficial to the reader if the authors could provide a simple scheme depicting the model develop during this study.

10. Supplemental Data set (.xls)

The adjusted p-values are clustered and >0.05 . Can the authors check and describe how those were calculated. How does it match with Volcano plots.

Materials and Methods:

11. A list of primers should be given with specification of their use.
12. The plasmids constructed for (over)expression of proteins/ production of recombinant proteins should be added. If published, references should be added accordingly.
13. RIP: the media for growing yeast cells should be added. Check also other section if defined.
14. RT-qPCR is not sufficiently described. RT kit needs specification, PCR reaction cycles should be given.
15. Quantification of mRNA levels in polysomes is unclear. How was the distribution of mRNA profiles determined? Have the authors added some RNA spikes to fractions?
16. The calculation for the enrichments in IPs is not described conclusively and should be added.
17. Polysomes fractionation (mammalian). It is indicated that the resultant supernatant was adjusted to 5M NaCl and 1 M MgCl₂. This seems to be very high - is this a typo? OR why such high concentrations have been chosen?

2. Significance:

Significance (Required)

The study appears solid and well done, except some weaknesses that need to be addressed (see above section).

Overall, the interplay with translation could be better investigated and the involvement for interactions with eIF4G/ ribosomes could be better investigated but possibly beyond this study.

Essentially, the study adds a nice mix of experimental approaches to manifest the linkage between granules formation, translation and the physiological implications for genotoxic stress response. However, it is not clear how the chosen conditions reflect any natural conditions (yeast may never be exposed to 100-200 mM HU) and hence, it is unclear how far the observations reflect nature and occur like that. It is certainly a limitation that only one particular stress condition was investigated and it is unclear whether it is also seen with other genotoxic stress inducing agents.

Audience: The study will attract the interest of researchers working with cytoplasmic RBPs, translation and stress granules. The latter topic is currently on a high wave as many

researchers jumped on this topic. The study may though remain within this circle as the analysis is rather specialised and contrained on one stress (genotoxic) and wider implications of the findings in a physiological context are unclear.

3. How much time do you estimate the authors will need to complete the suggested revisions:

Estimated time to Complete Revisions (Required)

(Decision Recommendation)

Between 1 and 3 months

Yes

Review #3

1. Evidence, reproducibility and clarity:

Evidence, reproducibility and clarity (Required)

In this manuscript, Rajyaguru, Vagner and colleagues address the role of yeast Scd6 in translation regulation during hydroxyurea (HU) stress response. They first show that Scd6 associates to puncta upon HU-treatment, in a manner that depends on the RGG and LSm domains of the protein. They then show that deletion of Scd6 has a mild effect on cell survival under HU-treatment, and over-expression of SCD6 restores the defects observed upon over-expression of SRS2 (a known regulator of the DNA damage response-DDR), suggesting that SRS2 is an SCD6 target, and that SCD6 negatively regulates SRS2. The authors further show that SCD6 binds to SRS2 mRNA and inhibits its translation. Binding to mRNA is somewhat regulated by methylation of the RGG domain, and methylation seems to be controlled by the LSm domain. Finally, the role of SCD6 as a translation regulator during HU-response is conserved, because its mammalian homolog, LSM14A, stimulates the translation of DDR factors under similar conditions. Altogether, this is a nice report showing a function of Scd6/LSM14A in regulation of translation upon HU treatment. However, there are some contradictions that need to be resolved, and the role of puncta in

the whole picture is not clear.

****Major comments:****

1. Page 5 and Fig S2E-F: The CLHX experiment to conclude that mRNA is present in Sdc6 and Sbp1 puncta is rather indirect. The fact that RNase treatment of a granule-enriched pellet has no effect (Fig S5B) does not help. The authors should perform RNase treatment of intact cells and see that the puncta disappear.
2. Fig 2A-F: The effects of Scd6 and Sbp1 deletion upon HU-treatment are very small. A more convincing effect is observed upon over-expression of both SRS2 and SCD6. What is the effect of over-expression of SDC6 and SBP1 alone (i.e. without SRS2 over-expression)?
3. Fig 2E: Why is there an opposite effect of deletion of Scd6 and Sbp1 in the SRS2 over-expression background?
4. Page 7, top: '...indicating that Scd6 regulated the expression of SRS2 in a HU-dependent manner.' In my opinion, the results so far suggest that Scd6 and SRS2 are somehow functionally connected during HU-treatment. To substantiate the statement of the authors, they should provide a Western blot showing that the levels of SRS2 change upon Scd6 KO or OE during HU-treatment. This will also substantiate the results shown in Figs 2G-H.
5. Figs 3: How are the localization of Scd6 protein and SRS2 mRNA to granules, and the levels of SRS2 protein, in cells exposed to HU after deletion of Hmt1? This would substantiate a role of Hmt1 in vivo.
6. Fig 3C: Is the increased interaction of SRS2 mRNA with Scd6 due to increased levels of SRS2 mRNA upon HU treatment? See also comment below.
7. Fig 4A: There seems to be an enrichment of SRS2 mRNA both in the granule-enriched pellet and in the supernatant upon HU treatment in the Scd6-GFP context, suggesting increased SRS2 mRNA levels altogether. The enrichment in granules upon HU is difficult to see, as one should measure the distribution of the mRNA in the pellet relative to the supernatant. Can the authors represent the ratio pellet/supernatant normalized to a control transcript? A similar calculation can be done for the protein normalized to a control protein.
8. Fig 4B: Increased juxtaposition of SRS2 mRNA and Scd6 granules upon HU treatment does not really mean increased colocalization. Granules are likely significantly apart such that increased interactions between the two partners are not explained by increased juxtaposition. Please, comment, tune-down and provide examples where increased granule juxtaposition is associated with increased interaction.
9. Fig 4D: These results are in direct contradiction with those shown in Fig 1C.
10. Fig 5E: Can the authors provide a GO analysis of the up- and down- regulated transcripts?

****Minor comments:****

11. Figures S1 and S2 seem to be swapped. Please make sure that Figures and panels are arranged in the order they are mentioned in the main text.
12. Page 5, sentence: 'our results argue for the role of Scd6 and Sbp1 in HU-mediated stress response'. I do not agree, as no functional assays showing that these proteins affect HU-mediated stress response have been provided at this point of the story. Please, delete.
13. Page 6: The authors state 'Since $\Delta scd6$ and $\Delta sbp1$ showed tolerance to chronic HU exposure...'. Where is this shown?
14. Fig 2F: The rescue by SCD6 OE is not complete, as mentioned in the main text.
15. Figure 2G-H: Please, indicate in the figure what the authors consider 'translated' and 'untranslated' fractions.

2. Significance:

Significance (Required)

The findings provide a new function for Scd6/LSM14A in regulation of translation upon HU treatment. There are limitations regarding the strength of effects in some cases, and the integration of the role of granule formation. These findings are useful for scientists working on genotoxic stress, RNA-binding proteins and/or translation.

3. How much time do you estimate the authors will need to complete the suggested revisions:

Estimated time to Complete Revisions (Required)

(Decision Recommendation)

Between 1 and 3 months

Yes

Revision Plan

Manuscript number: RC-2024-02640

Corresponding author(s): Purusharth I, Rajyaguru; Stephan Vagner

1. General Statements

This section is optional. Insert here any general statements you wish to make about the goal of the study or about the reviews.

In the manuscript titled, "RGG motif-containing Scd6/LSM14A proteins regulate the translation of specific mRNAs in response to hydroxyurea-induced genotoxic stress" we elucidate a conserved role of an RNA-binding protein with low-complexity sequences (RGG-motifs) in genotoxic stress response. This work uncovers HU-stress mediated translation regulation of SRS2, Ligase IV and RTEL1 transcripts by Scd6 (yeast)/LSM14 (human). It further identifies RNP condensates and arginine methylation as sites and means of this regulation.

We heartily thank all three reviewers for their overall encouraging comments about the significance of this manuscript. Specifically, we appreciate their view that the manuscript provides new functional insights into the role of RGG-motif-containing RNA-binding protein in genotoxic stress response. They further agree that such knowledge will impact and interest the general audience of RNA biology and stress biology.

We have carefully noted all the comments raised by three reviewers. We have addressed almost all the comments, including several by performing new experiments. The new results and their analysis have helped us improve the manuscript, allowing us to provide a stronger mechanistic and functional insight underlying the findings presented in this work. We thank the reviewers for their insightful comments. Below, we provide a point-by-point response to each of the comments.

2. Description of the planned revisions

Insert here a point-by-point reply that explains what revisions, additional experimentations and analyses are planned to address the points raised by the referees.

Reviewer 3

Major Comment 4: Page 7, top: '...indicating that Scd6 regulated the expression of SRS2 in a HU-dependent manner.' In my opinion, the results so far suggest that Scd6 and SRS2 are somehow functionally connected during HU-treatment. To substantiate the statement of the authors, they should provide a Western blot showing that the levels of SRS2 change upon Scd6 KO or OE during HU-treatment. This will also substantiate the results shown in Figs 2G-H.

Response: We thank the reviewer for this comment. Detecting Srs2 protein has been technically challenging. The SRS2 construct used in this study is untagged. Unfortunately, the commercial

Revision Plan

SRS2 antibody has been discontinued. We requested several groups who have used SRS2 antibody in their past studies but they have either closed down their labs or are unable to find an aliquot to share. We have tried tagging SRS2 with 6xHis/1XFLAG/3xFLAG tags at N and C-terminal, but unfortunately, the protein was undetectable in the Western blot analysis using either of the tag-specific antibodies. We have also tried western blot analysis using SRS2-GFP strain, but the protein does not get detected by anti-GFP antibody, probably because of very low expression.

Since we will not be able to provide western blots for Srs2 protein levels due to technical challenges, we shall provide western blots for RTEL1 (human homolog of Srs2) protein levels upon Lsm14A knockdown in the presence and absence of HU. This will validate the polysome data we have of RTEL1 regulation by LSM14A, and would, by extension, substantiate the SRS2 polysome data.

Major Comment 5: Figs 3: How are the localization of Scd6 protein and SRS2 mRNA to granules, and the levels of Srs2 protein, in cells exposed to HU after deletion of Hmt1? This would substantiate a role of Hmt1 in vivo.

Response: We will provide the data for Scd6 protein localization and SRS2 mRNA localization in granule enriched fraction upon HU treatment in $\Delta hmt1$ background. This experiment is ongoing.

3. Description of the revisions that have already been incorporated in the transferred manuscript

Reviewer 1

Major Comment 1: Fig. 1 F/G: were the delta RGG and LSM variants expressed at an equivalent level to the WT protein in these experiments?

Response: We thank the reviewer for this comment. We have quantified the total fluorescence intensity of GFP from the existing microscopy images for WT and domain deletion mutants for both Scd6 and Sbp1 (Now Figure 3A and 3D). This result (added as a new figure panel Fig 3C and 3F) indicates that the levels of Scd6 Δ RGG mutant is more whereas Scd6 Δ Lsm protein levels are comparable than WT. Similarly, Sbp1 Δ RGG mutant expression is comparable to WT in the given experimental conditions.

Major Comment 2: Fig. 3G: The 6 data points for the delta LSM variant are literally spread evenly up and down the graph, making these data appear highly questionable as to whether one can draw a definitive conclusion from them.

Revision Plan

Response: We agree with the reviewer that the data points are varied. To address the scatter in data, we have performed additional experiments and added those to the existing results. Even though there is a spread in the points, except for one data point, all others show an increase in methylation of LSM domain deletion mutant compared to WT, which is statistically significant. The old blot and graph (Old Figure 3F and 3G) have now been replaced with new ones (Figure 5F and 5G) which look more convincing. The result and conclusion derived from it remain unchanged.

Minor Comments

Comment 1: Abstract: the acronym NHEJ likely will need to be defined for the general reader.

Response: The acronym has been expanded in the abstract and explained in the introduction.

Comment 2: Introduction, first paragraph: change gene expression to 'transcription' in the phrase 'Even if the contribution of gene expression to GSR..' as I assume this is what is meant here. Gene expression consists of synthesis, processing, translation and decay.

Response: The required change has been made.

Comment 3: Pg. 3 Introduction: Since they are liquid-liquid phase condensates and ribonucleoproteins (RNPs) refer to any protein-RNA interaction, I think that referring to PBs and SGs as mRNPs is a bit misleading (especially the 'major mRNPs').

Response: The statement has been rewritten.

Comment 4: Introduction: are PBs truly 'sites' of mRNA decay as stated? There are papers in the literature that would argue otherwise.

Response: The statement has been modified with more citations.

Comment 5: Pg. 3, three lines from bottom. Change LSM14 to LSM14A

Response: The addition has been done.

Comment 6: Pg. 4 top - What is an 'LCS' - containing protein? The acronym has not been defined

Response: The acronym has been defined now. We have also defined acronyms wherever they were missing.

Comment 7: Fig. S1 - there are a lot of important data in this figure that demonstrate the coordinated movement of Scd6 and Sbp1 to granules. They should be moved into the main

Revision Plan

body of the manuscript in my opinion. Likewise, a whole section of the Results is dedicated to Fig. S2 - thus I would suggest moving these data into the main body of the manuscript to assist the reader.

Response: We thank the reviewer for pointing this out. Figure S1 has now been added to the main body of the manuscript as Figure 2. Figure S2 has now been added to Figure 1 and new Figure 3. This rearrangement has improved the flow of the manuscript.

Comment 8: Fig. 1F should be flipped in the figure with panel G since G is discussed in the results section before F

Response: Figure 1F and 1G are now Figure 3A and 3D and in the same order as mentioned in the text.

Comment 9: Be sure to define all acronyms for the reader.

Response: All acronyms in the manuscript have been defined wherever applicable.

Comment 11: Fig. 3H/I: It might be optimal to calculate and compare Kd's for the methylated and unmethylated variants. Also, the labels at the top of 3H do not line up with the wells of the EMSA gel.

Response: We have calculated the Kd's for the EMSA, and it has been added to the results section. We have also aligned the labels at the top of the EMSA gel (now Figure 5I) to match with the wells.

Reviewer 2

Major Comment 1: Fig. 2A, B. While there seems to be an effect on the lag phase, it could be revealing if the authors pls. calculate the doubling times for the strains and treatments (taking through the exponential growth phase). Furthermore, it would be good if the authors can show the rescue of phenotypes for deletion strains (ie. reintroduction of respective gene on ARS-CEN based plasmids or (if not available) with the OE plasmids.

Response: We thank the reviewer for this remark. We have calculated the doubling times for the strains in the tested conditions and added in the text. We have analyzed the effect of complementing the deletion strains with the respective genes on the CEN plasmid. We observe that $\Delta scd6$ shows tolerance to HU stress as previously seen, which gets rescued almost completely upon complementation with WT SCD6. This result has been included in the manuscript as a new figure panel (Figure S1A) . $\Delta sbp1$ also shows marginal tolerance to HU stress, but complementation with WT SBP1 only slightly rescues the phenotype, which is not statistically significant (Figure S1B). This result highlights a more important role of Scd6 as compared to Sbp1 in genotoxic stress response.

Revision Plan

Major Comment 2 (part 1): Fig. 3H. The authors tested the 5'UTR of SRS2 for interaction with recombinant Scd6. Firstly, it is unclear why the authors have chosen the 5'UTR for investigation? Can the authors explain.

Response: We thank the reviewer for this important comment. During experimentation and analysis, we assayed Scd6 binding to two different fragments of SRS2 mRNA: 5' and 3'UTR of same lengths (200 bases). We used the UTR fragments because there are numerous reports indicating the role of UTRs in the regulation by RNA binding proteins (<https://doi.org/10.1093/bfpg/els056>, <https://doi.org/10.1126/science.aad9868>, <https://doi.org/10.1093/jxb/erae073>). RNA EMSAs with purified Scd6 and *in vitro* transcribed UTR RNA fragments revealed a significantly better binding of Scd6 with the 5' UTR fragment of SRS2 mRNA compared to the 3' UTR. Therefore, we proceeded with the 5' UTR fragment for further analysis. We have now added this as a supplementary figure panel and explanation in the manuscript text (Figure S2B).

Major Comment 2 (part 2): Secondly, the affinities are relatively low (μM), and the gel shift assay lacks a negative control. The authors should test an unrelated RNA fragment of approximately the same size to control for specificity (negative control). It is unclear whether the protein could interact with any RNA fragment through a charged RNA backbone.

Response: Our *in vivo* data suggests that the binding of Scd6 with SRS2 mRNA is condition and RNA-specific and is regulated by methylation (now Figure 5C, S2A and 5E). As the reviewer mentioned, Scd6, in principle, could bind to any RNA molecule given the affinity of an RNA-binding protein (with positively charged amino acids such as arginine) to RNA molecule. Nevertheless, the significant difference in the binding of Scd6 to the 5'UTR and 3'UTR fragments itself acts as a relative control for EMSA. The aim of the *in vitro* experiment (EMSA) was to establish the difference, if any, in the binding affinities of unmethylated vs methylated Scd6, like the *in vivo* data, where we observe significantly increased binding to SRS2 mRNA upon decreased Scd6 methylation.

Major Comment 2 (part 3): Thirdly, it would be good if the authors could show a Coomassie gel for the recombinant protein used in those assays.

Response: The Coomassie gel which was provided as part the supplementary data (now Figure S2C), have now been added as another gel image to the main figure (Figure 5H), next to the EMSA, for better clarity.

Major Comment 3: Methods and Materials: The Materials and Methods section lacks important information and requires further details to evaluate the study (see below 11 – 17)

Response: The comment has been duly noted.

Revision Plan

Minor Comments

Results:

Comment 4: The numbering of Figure S1, S2 is confused in the first part of the results section. The authors should check numbering. In general, numbering should follow in the order of the text - pls. check.

Response: Based on the comment#7 by Reviewer 1, Figure S1 and S2 have now been added to the main figure, and the changes in the text have been made accordingly.

Comment 5: Pg. 5. CHX treatment leads to a decrease in Scd6-GFP and SBP-1 GFP granules. Essentially, CHX blocks translation elongation so the result indicates that puncta depend on active translation. The authors may want to add this liaising point towards the claim that mRNAs could be present in those puncta. How this results integrates with data shown in Fig. S5B.

Response: We thank the reviewer for this comment. Since granules are dynamic structures that depend on active translation, CHX treatment leads to the dissociation of Scd6 and Sbp1 granules. This indicate that most of the mRNAs present in these granules could be recycled for translation in polysomes. This strategy has been used in multiple research articles for similar deductions ([10.1091/mbc.E08-05-0499](https://doi.org/10.1091/mbc.E08-05-0499), <https://doi.org/10.1083/jcb.151.6.1257>, <https://doi.org/10.1093/nar/gku582>). We have now modified the text in the manuscript to accommodate this point. It has been previously reported that core components of stress granules, once formed are stable and resistant to RNase, EDTA and NaCl treatment *ex vivo* (<https://doi.org/10.1016/j.cell.2015.12.038>), even when these structures have RNA. Figure S5B (now S3C) indicates that the granule enriched fraction derived from untreated and treated cells indeed behaves like stress granule cores and not protein aggregates allowing us to proceed with downstream experiments.

Comment 6: Fig. 2H. It would be helpful to the reader, if the authors could mark the respective fraction in the polysomes taken for analysis of relative enrichments. How was this relative enrichment was calculated needs further description.

Response: The modification has been made (now Figure 4G) and added to the methods and materials.

Comment 7: Fig. S5B. 1% SDS treatment cause absence for Scd6 signal from the pellet fraction. Based on this result, I am not clear how based on this result they can claim for

Revision Plan

presence of higher order mRNA-protein complexes? Why does it exclude the possibility for Scd6 aggregates accumulating in the pellet? The authors need to explain/ modify this statement. Related to earlier findings that showed dependency of puncta upon CHX treatment, one wonders how this result matches to this earlier observation (ie.EDTA should disassemble ribosomes)? Can the authors explain?

Response: The very stable β -zipper interactions present in prion like domains, which leads to aggregation, is resistant to 1-2% SDS treatment (<https://doi.org/10.1016/j.cell.2015.12.038>). Hence, we think that solubilization upon 1% SDS treatment indicates that these are not aggregates. EDTA and NaCl are capable of disrupting interactions, which are stabilized mainly by electrostatic forces. Our observations (now Figure S3C) indicate that Scd6 could be part of the more stable mRNP condensate core structure and are therefore resistant to these treatments. Such observations have been previously reported, for example, stress granules in yeast are not affected by EDTA and NaCl treatments (<https://doi.org/10.1016/j.cell.2015.12.038>).

Comment 8 (part 1): Fig. 5E, F. For the RNA-seq, the authors compared polysomes with free RNAs (up to 80S) and found enrichment of LIG4 and RTEL1. However, the polysomal profiling mainly shows a slight shift of those mRNAs in higher polysomes; while there is no difference compared to free fractions. How can this be explained?

Response: We observed a shift from lower polysome fractions (11-12-13) (not from free fractions) to higher polysome fractions (14-15) indicating an increased number of ribosomes translating the RTEL1 mRNA.

Comment 8 (part 2): On the line, the authors should indicate clearly what fractions were pooled for RNA seq analysis. It is also not clear how the authors quantified percentage of RNA in individual fractions (have they spiked-in an RNA?) - this needs to be stated in the M&M section.

Response: We have now added the requested information in the Materials and Methods section. Fractions 13 to 17 were pooled for RNAseq analysis. The % of RNA in each fraction was calculated as described in Panda AC et al. Bio Protoc . 2017 Feb 5;7(3):e2126. doi: [10.21769/BioProtoc.2126](https://doi.org/10.21769/BioProtoc.2126)

Comment 9: At the end, it may be beneficial to the reader if the authors could provide a simple scheme depicting the model developed during this study.

Response: We thank the reviewer for this comment. We have included a model derived from our study as a new figure (Figure 8).

Revision Plan

Comment 10: Supplemental Data set (.xls) The adjusted p-values are clustered and >0.05 . Can the authors check and describe how those were calculated. How does it match with Volcano plots.

Response: The adjusted p-values are indeed >0.05 . The p-values (and not the adjusted p-values) are plotted in the Volcano plot (now Fig. 7E)

Materials and Methods:

Comment 11: A list of primers should be given with specification of their use.

Response: The list has been added in the supplementary files (Table S3)

Comment 12: The plasmids constructed for (over)expression of proteins/ production of recombinant proteins should be added. If published, references should be added accordingly.

Response: The list has been added in the supplementary files (Table S4)

Comment 13: RIP: the media for growing yeast cells should be added. Check also other section if defined.

Response: The information has been added wherever required.

Comment 14: RT-qPCR is not sufficiently described. RT kit needs specification, PCR reaction cycles should be given.

Response: The information has been added

Comment 15: Quantification of mRNA levels in polysomes is unclear. How was the distribution of mRNA profiles determined? Have the authors added some RNA spikes to fractions?
See above.

Response: The % of RNA in each fraction was calculated as described in Panda AC et al. Bio Protoc . 2017 Feb 5;7(3):e2126. doi: [10.21769/BioProtoc.2126](https://doi.org/10.21769/BioProtoc.2126). Details have now been added in the Mat and Meth section.

Comment 16: The calculation for the enrichments in IPs is not described conclusively and should be added.

Response: The calculation has now been elaborated and added to the methods and materials section.

Comment 17: Polysomes fractionation (mammalian). It is indicated that the resultant

Revision Plan

supernatant was adjusted to 5M NaCl and 1 M MgCl₂. This seems to be very high - is this a typo? OR why such high concentrations have been chosen?

Response: The sentence has been removed. There is no need for such adjustment.

Review 3

Major Comment 2: Fig 2A-F: The effects of Scd6 and Sbp1 deletion upon HU-treatment are very small. A more convincing effect is observed upon over-expression of both SRS2 and SCD6. What is the effect of over-expression of SCD6 and SBP1 alone (i.e. without SRS2 over-expression)?

Response: We thank the reviewer for this comment. The effects are indeed small but consistent and reproducible with two different kinds of assays (growth curve and plating assay, now Figure 4A-C). Overexpression of Scd6 or Sbp1 alone when expressed from a CEN/2u plasmid does not have any phenotype in the presence of HU (Figure S1A and S1B). Although, it has been previously reported that *galactose*-inducible Scd6 causes a severe growth defect (<https://doi.org/10.1093/nar/gkw762>), we performed spot assays with galactose inducible Scd6 and Sbp1 on control and HU plates, but did not see any difference in the extent of growth upon HU treatment. This data has now been presented as Figure S1C.

Major Comment 3: Fig 2E: Why is there an opposite effect of deletion of Scd6 and Sbp1 in the SRS2 over-expression background?

Response: We thank the reviewer for this comment; however, we respectfully disagree with the idea that overexpression of SRS2 yields opposite phenotypes in SCD6 and SBP1 deletion backgrounds. Figure 2E (now Figure 4E) gives the impression that SRS2 overexpression in SBP1 deletion grows significantly more for two reasons. There was an increased spotting of $\Delta sbp1$ cells overexpressing SRS2 (row#6) as compared to $\Delta scd6$ cells overexpressing SRS2 (row#4), which is evident in the plate without HU (left panel). Additionally, there is also reduced spotting of wild-type cells overexpressing SRS2 (row#2) as compared to $\Delta scd6$ cells overexpressing SRS2 (row#4). We have now replaced these panels with another image with better loadings. Quantitation of five experiments (Figure S1F) indicates that $\Delta sbp1$ grows slightly better in both EV and SRS2 over-expression background, but the increase is not statistically significant. We interpret this data to suggest that SRS2 is not a direct target of Sbp1.

Revision Plan

Another protein perhaps performs the specific role of Sbp1 in assisting Scd6 in genotoxic stress response in $\Delta sbp1$ background.

Major Comment 6: Fig 3C: Is the increased interaction of SRS2 mRNA with Scd6 due to increased levels of SRS2 mRNA upon HU treatment? See also comment below.

Response: Based on RT-qPCR of total RNA, SRS2 mRNA levels do not seem to increase, which has now been added as a Supplementary figure (Figure S3D, left panel). Moreover, quantification of SRS2 mRNA from the FISH data also does not support an increase in mRNA levels (Figure 6D, left panel).

Major Comment 7: Fig 4A: There seems to be an enrichment of SRS2 mRNA both in the granule-enriched pellet and in the supernatant upon HU treatment in the Scd6-GFP context, suggesting increased SRS2 mRNA levels altogether. The enrichment in granules upon HU is difficult to see, as one should measure the distribution of the mRNA in the pellet relative to the supernatant. Can the authors represent the ratio pellet/supernatant normalized to a control transcript? A similar calculation can be done for the protein normalized to a control protein.

Response: As mentioned earlier, RT-qPCR data with SRS2 mRNA levels in total lysate has been added to supplementary data (Figure S3D, left panel). Based on RT-qPCR of total RNA, SRS2 mRNA levels do not seem to increase.

The quantification of SRS2 mRNA and Scd6 protein enrichment is done such that the supernatant and pellet fractions are separately normalized to their respective controls (Scd6GFP, untreated sample) and therefore do not represent the mRNA distribution but relative mRNA enrichment. However, as per the recommendation by the reviewer, the data has been replotted as a ratio of supernatant and pellet with the addition of two more data points and has been added in the main figure (Figure 6E). The data concludes increased enrichment of SRS2 mRNA in granules upon HU treatment. The previous data has been included in the supplementary data as Supplementary figure (Figure S3D, right panel).

Major Comment 8: Fig 4B: Increased juxtaposition of SRS2 mRNA and Scd6 granules upon HU treatment does not really mean increased colocalization. Granules are likely significantly apart such that increased interactions between the two partners are not explained by increased juxtaposition. Please, comment, tune-down and provide examples where increased granule juxtaposition is associated with increased interaction.

Response: We believe that the usage of term 'juxtaposition' is leading to misinterpretation of the data. Therefore, we have replaced it with 'percentage area overlap' analysis to demonstrate that the SRS2 mRNA foci indeed overlap/localize with Scd6GFP foci up to an average of 43.5% in HU stress. This analysis has been added as an additional panel (Figure 6C), indicating that the SRS2 mRNA interacts with Scd6 in the granules. Even though the granules do not overlap/localize completely, the observed area of granule overlap (43.5%) is functionally

Revision Plan

effective as it leads to the physical interaction of Scd6 and SRS2 (Figure 6E & 5C) and, consequently, repression (Figure 4H). The FISH data, granule enrichment, and RNA immunoprecipitation data demonstrate Scd6 protein and SRS2 mRNA interaction in granules.

Major Comment 9: Fig 4D: These results are in direct contradiction with those shown in Fig 1C.

Response: We thank the reviewer for this comment. Figure 1C (now Figure 1B and 1C) demonstrates that Scd6 localization to puncta, when expressed from a CEN plasmid, significantly increases upon HU stress. The same trend is visible in Figure 4D (now Figure 6D) where Scd6 is expressed from a 2 μ plasmid; however, it is not significant. The data in 1C and 4D (now 1C and 6D respectively) are rather inconsistent with each other than being contradictory. Nevertheless, we understand this reviewer's concern and address it below.

The initial localization experiments were performed using Scd6 expressed from CEN plasmid or genomically tagged Scd6. Since both these versions of Scd6 are not detectable using western blotting, we used Scd6 expressed from 2 μ plasmid. Localization to condensates by liquid-liquid phase separation is a concentration-driven phenomenon. Therefore, when Scd6 is expressed from a 2 μ plasmid amounting to increased protein levels, its localization to puncta increases even in the absence of stress, which is visible in the quantitation provided in the figure (Figure 6D) as compared to Figure 1C. We have now analyzed the percentage granular localization (granule intensity) of Scd6 (2 μ), which significantly increases upon HU stress (Figure S3A). Thus although number of Scd6 granules does not increase upon HU stress when expressed from a 2 μ plasmid, there is significant increase in localization of Scd6 to granule upon HU stress (Figure S3A).

Major comment 10: Fig 5E: Can the authors provide a GO analysis of the up- and down-regulated transcripts?

Response: We have now provided a GO analysis (Table S2). However, due to the low number of regulated genes, only a few GO terms with weak scores appeared in the analysis.

Minor comments:

Comment 11: Figures S1 and S2 seem to be swapped. Please make sure that Figures and panels are arranged in the order they are mentioned in the main text.

Response: We thank the reviewer for pointing it out. Based on the comment#7 by Reviewer 1, Figure S1 and S2 have now been added to the main figure, and the changes in the text have been made accordingly. We have ensured that the order of figures matches the text.

Comment 12: Page 5, sentence: 'our results argue for the role of Scd6 and Sbp1 in HU-mediated stress response'. I do not agree, as no functional assays showing that these proteins

Revision Plan

affect HU-mediated stress response have been provided at this point of the story. Please, delete.

Response: We have removed the sentence from the existing paragraph.

Comment 13: Page 6: The authors state 'Since Dscd6 and Dsbp1 showed tolerance to chronic HU exposure...'. Where is this shown?

Response: The growth curve in Figure 2A and 2B (now Figure 4A and 4B) and the plating assay in Figure 2C (now Figure 4C) was done with hydroxyurea in the media/plate. Hence, we state that deletion of either SCD6 or SBP1 shows tolerance to chronic (or continuous) HU stress.

Comment 14: Fig 2F: The rescue by SCD6 OE is not complete, as mentioned in the main text.

Response: We have now included the quantification of the spot assay in 2F (now Figure 4F) to show that the rescue by SCD6 overexpression is complete (Fig S1G).

Comment 15: Figure 2G-H: Please, indicate in the figure what the authors consider 'translated' and 'untranslated' fractions.

Response: The fractions have now been labelled to indicate the missing information in Figure 2G (now Figure 4G).

4. Description of analyses that authors prefer not to carry out

Review 1

Minor Comment 10: Pg. 8/Fig. S3D/4A: It would be interesting to complete the story and determine the functional relationship of Scd6 to the DNL4 mRNA

Response: It is indeed an interesting observation and is currently being pursued as part of another story. We believe it is beyond the scope of the current manuscript.

Revision Plan

Review 3

Major Comment 1: Page 5 and Fig S2E-F: The CLHX experiment to conclude that mRNA is present in Scd6 and Sbp1 puncta is rather indirect. The fact that RNase treatment of a granule-enriched pellet has no effect (Fig S5B) does not help. The authors should perform RNase treatment of intact cells and see that the puncta disappear.

Response: We thank the reviewer for this comment. Cycloheximide treatment is a well-accepted assay to detect the presence of mRNA in granules. Since granules are dynamic structures, and these depend on active translation, CHX treatment leads to the dissociation of Scd6 and Sbp1 granules. This indicates that granule assembly depends on the availability of mRNA derived from translating ribosomes. The observation that Scd6 puncta are sensitive to cycloheximide but not to RNase A treatment is not surprising. It indeed is consistent with the properties of some of the condensates reported in the literature. For example, stress granule cores that are sensitive to cycloheximide, like Scd6 puncta, are resistant to RNase treatment in lysate, indicating that once formed, these structures are quite stable (<https://doi.org/10.1016/j.cell.2015.12.038>). It is interpreted to suggest that the RNAs in these condensates are protected by the RNA-binding proteins. Also, subsequently, in the study, we do RNA immunoprecipitation and granule enrichment experiments and show specific RNA enrichment with Scd6 (Figure 5C, 6A).

Dear Dr. Rajyaguru,

Thank you for the transfer of your manuscript to EMBO reports. I now read the manuscript, describing roles for the RGG-box containing RNA-binding protein Scd6 and its human ortholog LSM14A in the genotoxic stress response to hydroxyurea (HU), the referee reports from Review Commons, and your p-b-p-response (revision plan) and discussed the paper with my colleagues.

We appreciate that these findings are of interest to the immediate field. However, considering that both proteins have been shown to localize to stress granules and to act as translational repressors, we do not think the study provides the advance and broader impact we are looking for in an EMBO reports paper, although we appreciate that you now identify with SRS2 a specific target. However, as the wider implications of these findings in a physiological context remain unclear (as also indicated by referee #2), we rather think that these data would fit much better into a more specialized journal (as also hinted at by the referees) and have therefore decided not to proceed with the manuscript.

That being said, I think your paper would also be a very good fit for our open-access journal Life Science Alliance, a broad scope Open Access journal published in partnership between the EMBO-, Rockefeller University-, and Cold Spring Harbor Laboratory Presses). I have shared your manuscript, your revision plan and the accompanying reviews with LSA Executive Editor, Eric Sawey, who is interested in these findings, and would like to invite further consideration of this manuscript at LSA, revised according to your Revision Plan.

We encourage you to use the link below to transfer your manuscript to LSA. You do not need to revise the manuscript before transferring it to LSA. Once you transfer, Dr. Sawey will email you an invitation to revise and resubmit, listing the same revision requests as mentioned above.

For more details of this service, and to transfer your manuscript to another EMBO title please click on Link Not Available

Please feel free to reach out at e.sawey@life-science-alliance.org if you have any questions about LSA, the transfer process or the revisions requested.

Yours sincerely,

Rev_Com_number: RC-2024-02640

New_manu_number: EMBOR-2024-60803V1-T

Corr_author: Rajyaguru

Title: RGG motif-containing proteins regulate the translation of specific mRNAs in genotoxic stress

Revision Plan

Manuscript number: RC-2024-02640

Corresponding author(s): Purusharth I, Rajyaguru; Stephan Vagner

1. General Statements

This section is optional. Insert here any general statements you wish to make about the goal of the study or about the reviews.

In the manuscript titled, "RGG motif-containing Scd6/LSM14A proteins regulate the translation of specific mRNAs in response to hydroxyurea-induced genotoxic stress" we elucidate a conserved role of an RNA-binding protein with low-complexity sequences (RGG-motifs) in genotoxic stress response. This work uncovers HU-stress mediated translation regulation of SRS2, Ligase IV and RTEL1 transcripts by Scd6 (yeast)/LSM14 (human). It further identifies RNP condensates and arginine methylation as sites and means of this regulation.

We heartily thank all three reviewers for their overall encouraging comments about the significance of this manuscript. Specifically, we appreciate their view that the manuscript provides new functional insights into the role of RGG-motif-containing RNA-binding protein in genotoxic stress response. They further agree that such knowledge will impact and interest the general audience of RNA biology and stress biology.

We have carefully noted all the comments raised by three reviewers. We have addressed almost all the comments, including several by performing new experiments. The new results and their analysis have helped us improve the manuscript, allowing us to provide a stronger mechanistic and functional insight underlying the findings presented in this work. We thank the reviewers for their insightful comments. Below, we provide a point-by-point response to each of the comments.

2. Description of the planned revisions

Insert here a point-by-point reply that explains what revisions, additional experimentations and analyses are planned to address the points raised by the referees.

Reviewer 3

Major Comment 4: Page 7, top: '...indicating that Scd6 regulated the expression of SRS2 in a HU-dependent manner.' In my opinion, the results so far suggest that Scd6 and SRS2 are somehow functionally connected during HU-treatment. To substantiate the statement of the authors, they should provide a Western blot showing that the levels of SRS2 change upon Scd6 KO or OE during HU-treatment. This will also substantiate the results shown in Figs 2G-H.

Response: We thank the reviewer for this comment. Detecting Srs2 protein has been technically challenging. The SRS2 construct used in this study is untagged. Unfortunately, the commercial

Revision Plan

SRS2 antibody has been discontinued. We requested several groups who have used SRS2 antibody in their past studies but they have either closed down their labs or are unable to find an aliquot to share. We have tried tagging SRS2 with 6xHis/1XFLAG/3xFLAG tags at N and C-terminal, but unfortunately, the protein was undetectable in the Western blot analysis using either of the tag-specific antibodies. We have also tried western blot analysis using SRS2-GFP strain, but the protein does not get detected by anti-GFP antibody, probably because of very low expression.

Since we will not be able to provide western blots for Srs2 protein levels due to technical challenges, we shall provide western blots for RTEL1 (human homolog of Srs2) protein levels upon Lsm14A knockdown in the presence and absence of HU. This will validate the polysome data we have of RTEL1 regulation by LSM14A, and would, by extension, substantiate the SRS2 polysome data.

Major Comment 5: Figs 3: How are the localization of Scd6 protein and SRS2 mRNA to granules, and the levels of Srs2 protein, in cells exposed to HU after deletion of Hmt1? This would substantiate a role of Hmt1 in vivo.

Response: We will provide the data for Scd6 protein localization and SRS2 mRNA localization in granule enriched fraction upon HU treatment in $\Delta hmt1$ background. This experiment is ongoing.

3. Description of the revisions that have already been incorporated in the transferred manuscript

Reviewer 1

Major Comment 1: Fig. 1 F/G: were the delta RGG and LSM variants expressed at an equivalent level to the WT protein in these experiments?

Response: We thank the reviewer for this comment. We have quantified the total fluorescence intensity of GFP from the existing microscopy images for WT and domain deletion mutants for both Scd6 and Sbp1 (Now Figure 3A and 3D). This result (added as a new figure panel Fig 3C and 3F) indicates that the levels of Scd6 Δ RGG mutant is more whereas Scd6 Δ Lsm protein levels are comparable than WT. Similarly, Sbp1 Δ RGG mutant expression is comparable to WT in the given experimental conditions.

Major Comment 2: Fig. 3G: The 6 data points for the delta LSM variant are literally spread evenly up and down the graph, making these data appear highly questionable as to whether one can draw a definitive conclusion from them.

Revision Plan

Response: We agree with the reviewer that the data points are varied. To address the scatter in data, we have performed additional experiments and added those to the existing results. Even though there is a spread in the points, except for one data point, all others show an increase in methylation of LSM domain deletion mutant compared to WT, which is statistically significant. The old blot and graph (Old Figure 3F and 3G) have now been replaced with new ones (Figure 5F and 5G) which look more convincing. The result and conclusion derived from it remain unchanged.

Minor Comments

Comment 1: Abstract: the acronym NHEJ likely will need to be defined for the general reader.

Response: The acronym has been expanded in the abstract and explained in the introduction.

Comment 2: Introduction, first paragraph: change gene expression to 'transcription' in the phrase 'Even if the contribution of gene expression to GSR..' as I assume this is what is meant here. Gene expression consists of synthesis, processing, translation and decay.

Response: The required change has been made.

Comment 3: Pg. 3 Introduction: Since they are liquid-liquid phase condensates and ribonucleoproteins (RNPs) refer to any protein-RNA interaction, I think that referring to PBs and SGs as mRNPs is a bit misleading (especially the 'major mRNPs').

Response: The statement has been rewritten.

Comment 4: Introduction: are PBs truly 'sites' of mRNA decay as stated? There are papers in the literature that would argue otherwise.

Response: The statement has been modified with more citations.

Comment 5: Pg. 3, three lines from bottom. Change LSM14 to LSM14A

Response: The addition has been done.

Comment 6: Pg. 4 top - What is an 'LCS' - containing protein? The acronym has not been defined

Response: The acronym has been defined now. We have also defined acronyms wherever they were missing.

Comment 7: Fig. S1 - there are a lot of important data in this figure that demonstrate the coordinated movement of Scd6 and Sbp1 to granules. They should be moved into the main

Revision Plan

body of the manuscript in my opinion. Likewise, a whole section of the Results is dedicated to Fig. S2 - thus I would suggest moving these data into the main body of the manuscript to assist the reader.

Response: We thank the reviewer for pointing this out. Figure S1 has now been added to the main body of the manuscript as Figure 2. Figure S2 has now been added to Figure 1 and new Figure 3. This rearrangement has improved the flow of the manuscript.

Comment 8: Fig. 1F should be flipped in the figure with panel G since G is discussed in the results section before F

Response: Figure 1F and 1G are now Figure 3A and 3D and in the same order as mentioned in the text.

Comment 9: Be sure to define all acronyms for the reader.

Response: All acronyms in the manuscript have been defined wherever applicable.

Comment 11: Fig. 3H/I: It might be optimal to calculate and compare Kd's for the methylated and unmethylated variants. Also, the labels at the top of 3H do not line up with the wells of the EMSA gel.

Response: We have calculated the Kd's for the EMSA, and it has been added to the results section. We have also aligned the labels at the top of the EMSA gel (now Figure 5I) to match with the wells.

Reviewer 2

Major Comment 1: Fig. 2A, B. While there seems to be an effect on the lag phase, it could be revealing if the authors pls. calculate the doubling times for the strains and treatments (taking through the exponential growth phase). Furthermore, it would be good if the authors can show the rescue of phenotypes for deletion strains (ie. reintroduction of respective gene on ARS-CEN based plasmids or (if not available) with the OE plasmids.

Response: We thank the reviewer for this remark. We have calculated the doubling times for the strains in the tested conditions and added in the text. We have analyzed the effect of complementing the deletion strains with the respective genes on the CEN plasmid. We observe that $\Delta scd6$ shows tolerance to HU stress as previously seen, which gets rescued almost completely upon complementation with WT SCD6. This result has been included in the manuscript as a new figure panel (Figure S1A) . $\Delta sbp1$ also shows marginal tolerance to HU stress, but complementation with WT SBP1 only slightly rescues the phenotype, which is not statistically significant (Figure S1B). This result highlights a more important role of Scd6 as compared to Sbp1 in genotoxic stress response.

Revision Plan

Major Comment 2 (part 1): Fig. 3H. The authors tested the 5'UTR of SRS2 for interaction with recombinant Scd6. Firstly, it is unclear why the authors have chosen the 5'UTR for investigation? Can the authors explain.

Response: We thank the reviewer for this important comment. During experimentation and analysis, we assayed Scd6 binding to two different fragments of SRS2 mRNA: 5' and 3'UTR of same lengths (200 bases). We used the UTR fragments because there are numerous reports indicating the role of UTRs in the regulation by RNA binding proteins (<https://doi.org/10.1093/bfpg/els056>, <https://doi.org/10.1126/science.aad9868>, <https://doi.org/10.1093/jxb/erae073>). RNA EMSAs with purified Scd6 and *in vitro* transcribed UTR RNA fragments revealed a significantly better binding of Scd6 with the 5' UTR fragment of SRS2 mRNA compared to the 3' UTR. Therefore, we proceeded with the 5' UTR fragment for further analysis. We have now added this as a supplementary figure panel and explanation in the manuscript text (Figure S2B).

Major Comment 2 (part 2): Secondly, the affinities are relatively low (μM), and the gel shift assay lacks a negative control. The authors should test an unrelated RNA fragment of approximately the same size to control for specificity (negative control). It is unclear whether the protein could interact with any RNA fragment through a charged RNA backbone.

Response: Our *in vivo* data suggests that the binding of Scd6 with SRS2 mRNA is condition and RNA-specific and is regulated by methylation (now Figure 5C, S2A and 5E). As the reviewer mentioned, Scd6, in principle, could bind to any RNA molecule given the affinity of an RNA-binding protein (with positively charged amino acids such as arginine) to RNA molecule. Nevertheless, the significant difference in the binding of Scd6 to the 5'UTR and 3'UTR fragments itself acts as a relative control for EMSA. The aim of the *in vitro* experiment (EMSA) was to establish the difference, if any, in the binding affinities of unmethylated vs methylated Scd6, like the *in vivo* data, where we observe significantly increased binding to SRS2 mRNA upon decreased Scd6 methylation.

Major Comment 2 (part 3): Thirdly, it would be good if the authors could show a Coomassie gel for the recombinant protein used in those assays.

Response: The Coomassie gel which was provided as part the supplementary data (now Figure S2C), have now been added as another gel image to the main figure (Figure 5H), next to the EMSA, for better clarity.

Major Comment 3: Methods and Materials: The Materials and Methods section lacks important information and requires further details to evaluate the study (see below 11 – 17)

Response: The comment has been duly noted.

Revision Plan

Minor Comments

Results:

Comment 4: The numbering of Figure S1, S2 is confused in the first part of the results section. The authors should check numbering. In general, numbering should follow in the order of the text - pls. check.

Response: Based on the comment#7 by Reviewer 1, Figure S1 and S2 have now been added to the main figure, and the changes in the text have been made accordingly.

Comment 5: Pg. 5. CHX treatment leads to a decrease in Scd6-GFP and SBP-1 GFP granules. Essentially, CHX blocks translation elongation so the result indicates that puncta depend on active translation. The authors may want to add this liaising point towards the claim that mRNAs could be present in those puncta. How this results integrates with data shown in Fig. S5B.

Response: We thank the reviewer for this comment. Since granules are dynamic structures that depend on active translation, CHX treatment leads to the dissociation of Scd6 and Sbp1 granules. This indicate that most of the mRNAs present in these granules could be recycled for translation in polysomes. This strategy has been used in multiple research articles for similar deductions ([10.1091/mbc.E08-05-0499](https://doi.org/10.1091/mbc.E08-05-0499), <https://doi.org/10.1083/jcb.151.6.1257>, <https://doi.org/10.1093/nar/gku582>). We have now modified the text in the manuscript to accommodate this point. It has been previously reported that core components of stress granules, once formed are stable and resistant to RNase, EDTA and NaCl treatment *ex vivo* (<https://doi.org/10.1016/j.cell.2015.12.038>), even when these structures have RNA. Figure S5B (now S3C) indicates that the granule enriched fraction derived from untreated and treated cells indeed behaves like stress granule cores and not protein aggregates allowing us to proceed with downstream experiments.

Comment 6: Fig. 2H. It would be helpful to the reader, if the authors could mark the respective fraction in the polysomes taken for analysis of relative enrichments. How was this relative enrichment was calculated needs further description.

Response: The modification has been made (now Figure 4G) and added to the methods and materials.

Comment 7: Fig. S5B. 1% SDS treatment cause absence for Scd6 signal from the pellet fraction. Based on this result, I am not clear how based on this result they can claim for

Revision Plan

presence of higher order mRNA-protein complexes? Why does it exclude the possibility for Scd6 aggregates accumulating in the pellet? The authors need to explain/ modify this statement. Related to earlier findings that showed dependency of puncta upon CHX treatment, one wonders how this result matches to this earlier observation (ie.EDTA should disassemble ribosomes)? Can the authors explain?

Response: The very stable β -zipper interactions present in prion like domains, which leads to aggregation, is resistant to 1-2% SDS treatment (<https://doi.org/10.1016/j.cell.2015.12.038>). Hence, we think that solubilization upon 1% SDS treatment indicates that these are not aggregates. EDTA and NaCl are capable of disrupting interactions, which are stabilized mainly by electrostatic forces. Our observations (now Figure S3C) indicate that Scd6 could be part of the more stable mRNP condensate core structure and are therefore resistant to these treatments. Such observations have been previously reported, for example, stress granules in yeast are not affected by EDTA and NaCl treatments (<https://doi.org/10.1016/j.cell.2015.12.038>).

Comment 8 (part 1): Fig. 5E, F. For the RNA-seq, the authors compared polysomes with free RNAs (up to 80S) and found enrichment of LIG4 and RTEL1. However, the polysomal profiling mainly shows a slight shift of those mRNAs in higher polysomes; while there is no difference compared to free fractions. How can this be explained?

Response: We observed a shift from lower polysome fractions (11-12-13) (not from free fractions) to higher polysome fractions (14-15) indicating an increased number of ribosomes translating the RTEL1 mRNA.

Comment 8 (part 2): On the line, the authors should indicate clearly what fractions were pooled for RNA seq analysis. It is also not clear how the authors quantified percentage of RNA in individual fractions (have they spiked-in an RNA?) - this needs to be stated in the M&M section.

Response: We have now added the requested information in the Materials and Methods section. Fractions 13 to 17 were pooled for RNAseq analysis. The % of RNA in each fraction was calculated as described in Panda AC et al. Bio Protoc . 2017 Feb 5;7(3):e2126. doi: [10.21769/BioProtoc.2126](https://doi.org/10.21769/BioProtoc.2126)

Comment 9: At the end, it may be beneficial to the reader if the authors could provide a simple scheme depicting the model developed during this study.

Response: We thank the reviewer for this comment. We have included a model derived from our study as a new figure (Figure 8).

Revision Plan

Comment 10: Supplemental Data set (.xls) The adjusted p-values are clustered and >0.05 . Can the authors check and describe how those were calculated. How does it match with Volcano plots.

Response: The adjusted p-values are indeed >0.05 . The p-values (and not the adjusted p-values) are plotted in the Volcano plot (now Fig. 7E)

Materials and Methods:

Comment 11: A list of primers should be given with specification of their use.

Response: The list has been added in the supplementary files (Table S3)

Comment 12: The plasmids constructed for (over)expression of proteins/ production of recombinant proteins should be added. If published, references should be added accordingly.

Response: The list has been added in the supplementary files (Table S4)

Comment 13: RIP: the media for growing yeast cells should be added. Check also other section if defined.

Response: The information has been added wherever required.

Comment 14: RT-qPCR is not sufficiently described. RT kit needs specification, PCR reaction cycles should be given.

Response: The information has been added

Comment 15: Quantification of mRNA levels in polysomes is unclear. How was the distribution of mRNA profiles determined? Have the authors added some RNA spikes to fractions?
See above.

Response: The % of RNA in each fraction was calculated as described in Panda AC et al. Bio Protoc . 2017 Feb 5;7(3):e2126. doi: [10.21769/BioProtoc.2126](https://doi.org/10.21769/BioProtoc.2126). Details have now been added in the Mat and Meth section.

Comment 16: The calculation for the enrichments in IPs is not described conclusively and should be added.

Response: The calculation has now been elaborated and added to the methods and materials section.

Comment 17: Polysomes fractionation (mammalian). It is indicated that the resultant

Revision Plan

supernatant was adjusted to 5M NaCl and 1 M MgCl₂. This seems to be very high - is this a typo? OR why such high concentrations have been chosen?

Response: The sentence has been removed. There is no need for such adjustment.

Review 3

Major Comment 2: Fig 2A-F: The effects of Scd6 and Sbp1 deletion upon HU-treatment are very small. A more convincing effect is observed upon over-expression of both SRS2 and SCD6. What is the effect of over-expression of SCD6 and SBP1 alone (i.e. without SRS2 over-expression)?

Response: We thank the reviewer for this comment. The effects are indeed small but consistent and reproducible with two different kinds of assays (growth curve and plating assay, now Figure 4A-C). Overexpression of Scd6 or Sbp1 alone when expressed from a CEN/2u plasmid does not have any phenotype in the presence of HU (Figure S1A and S1B). Although, it has been previously reported that *galactose*-inducible Scd6 causes a severe growth defect (<https://doi.org/10.1093/nar/gkw762>), we performed spot assays with galactose inducible Scd6 and Sbp1 on control and HU plates, but did not see any difference in the extent of growth upon HU treatment. This data has now been presented as Figure S1C.

Major Comment 3: Fig 2E: Why is there an opposite effect of deletion of Scd6 and Sbp1 in the SRS2 over-expression background?

Response: We thank the reviewer for this comment; however, we respectfully disagree with the idea that overexpression of SRS2 yields opposite phenotypes in SCD6 and SBP1 deletion backgrounds. Figure 2E (now Figure 4E) gives the impression that SRS2 overexpression in SBP1 deletion grows significantly more for two reasons. There was an increased spotting of $\Delta sbp1$ cells overexpressing SRS2 (row#6) as compared to $\Delta scd6$ cells overexpressing SRS2 (row#4), which is evident in the plate without HU (left panel). Additionally, there is also reduced spotting of wild-type cells overexpressing SRS2 (row#2) as compared to $\Delta scd6$ cells overexpressing SRS2 (row#4). We have now replaced these panels with another image with better loadings. Quantitation of five experiments (Figure S1F) indicates that $\Delta sbp1$ grows slightly better in both EV and SRS2 over-expression background, but the increase is not statistically significant. We interpret this data to suggest that SRS2 is not a direct target of Sbp1.

Revision Plan

Another protein perhaps performs the specific role of Sbp1 in assisting Scd6 in genotoxic stress response in $\Delta sbp1$ background.

Major Comment 6: Fig 3C: Is the increased interaction of SRS2 mRNA with Scd6 due to increased levels of SRS2 mRNA upon HU treatment? See also comment below.

Response: Based on RT-qPCR of total RNA, SRS2 mRNA levels do not seem to increase, which has now been added as a Supplementary figure (Figure S3D, left panel). Moreover, quantification of SRS2 mRNA from the FISH data also does not support an increase in mRNA levels (Figure 6D, left panel).

Major Comment 7: Fig 4A: There seems to be an enrichment of SRS2 mRNA both in the granule-enriched pellet and in the supernatant upon HU treatment in the Scd6-GFP context, suggesting increased SRS2 mRNA levels altogether. The enrichment in granules upon HU is difficult to see, as one should measure the distribution of the mRNA in the pellet relative to the supernatant. Can the authors represent the ratio pellet/supernatant normalized to a control transcript? A similar calculation can be done for the protein normalized to a control protein.

Response: As mentioned earlier, RT-qPCR data with SRS2 mRNA levels in total lysate has been added to supplementary data (Figure S3D, left panel). Based on RT-qPCR of total RNA, SRS2 mRNA levels do not seem to increase.

The quantification of SRS2 mRNA and Scd6 protein enrichment is done such that the supernatant and pellet fractions are separately normalized to their respective controls (Scd6GFP, untreated sample) and therefore do not represent the mRNA distribution but relative mRNA enrichment. However, as per the recommendation by the reviewer, the data has been replotted as a ratio of supernatant and pellet with the addition of two more data points and has been added in the main figure (Figure 6E). The data concludes increased enrichment of SRS2 mRNA in granules upon HU treatment. The previous data has been included in the supplementary data as Supplementary figure (Figure S3D, right panel).

Major Comment 8: Fig 4B: Increased juxtaposition of SRS2 mRNA and Scd6 granules upon HU treatment does not really mean increased colocalization. Granules are likely significantly apart such that increased interactions between the two partners are not explained by increased juxtaposition. Please, comment, tune-down and provide examples where increased granule juxtaposition is associated with increased interaction.

Response: We believe that the usage of term 'juxtaposition' is leading to misinterpretation of the data. Therefore, we have replaced it with 'percentage area overlap' analysis to demonstrate that the SRS2 mRNA foci indeed overlap/localize with Scd6GFP foci up to an average of 43.5% in HU stress. This analysis has been added as an additional panel (Figure 6C), indicating that the SRS2 mRNA interacts with Scd6 in the granules. Even though the granules do not overlap/localize completely, the observed area of granule overlap (43.5%) is functionally

Revision Plan

effective as it leads to the physical interaction of Scd6 and SRS2 (Figure 6E & 5C) and, consequently, repression (Figure 4H). The FISH data, granule enrichment, and RNA immunoprecipitation data demonstrate Scd6 protein and SRS2 mRNA interaction in granules.

Major Comment 9: Fig 4D: These results are in direct contradiction with those shown in Fig 1C.

Response: We thank the reviewer for this comment. Figure 1C (now Figure 1B and 1C) demonstrates that Scd6 localization to puncta, when expressed from a CEN plasmid, significantly increases upon HU stress. The same trend is visible in Figure 4D (now Figure 6D) where Scd6 is expressed from a 2 μ plasmid; however, it is not significant. The data in 1C and 4D (now 1C and 6D respectively) are rather inconsistent with each other than being contradictory. Nevertheless, we understand this reviewer's concern and address it below.

The initial localization experiments were performed using Scd6 expressed from CEN plasmid or genomically tagged Scd6. Since both these versions of Scd6 are not detectable using western blotting, we used Scd6 expressed from 2 μ plasmid. Localization to condensates by liquid-liquid phase separation is a concentration-driven phenomenon. Therefore, when Scd6 is expressed from a 2 μ plasmid amounting to increased protein levels, its localization to puncta increases even in the absence of stress, which is visible in the quantitation provided in the figure (Figure 6D) as compared to Figure 1C. We have now analyzed the percentage granular localization (granule intensity) of Scd6 (2 μ), which significantly increases upon HU stress (Figure S3A). Thus although number of Scd6 granules does not increase upon HU stress when expressed from a 2 μ plasmid, there is significant increase in localization of Scd6 to granule upon HU stress (Figure S3A).

Major comment 10: Fig 5E: Can the authors provide a GO analysis of the up- and down-regulated transcripts?

Response: We have now provided a GO analysis (Table S2). However, due to the low number of regulated genes, only a few GO terms with weak scores appeared in the analysis.

Minor comments:

Comment 11: Figures S1 and S2 seem to be swapped. Please make sure that Figures and panels are arranged in the order they are mentioned in the main text.

Response: We thank the reviewer for pointing it out. Based on the comment#7 by Reviewer 1, Figure S1 and S2 have now been added to the main figure, and the changes in the text have been made accordingly. We have ensured that the order of figures matches the text.

Comment 12: Page 5, sentence: 'our results argue for the role of Scd6 and Sbp1 in HU-mediated stress response'. I do not agree, as no functional assays showing that these proteins

Revision Plan

affect HU-mediated stress response have been provided at this point of the story. Please, delete.

Response: We have removed the sentence from the existing paragraph.

Comment 13: Page 6: The authors state 'Since Dscd6 and Dsbp1 showed tolerance to chronic HU exposure...'. Where is this shown?

Response: The growth curve in Figure 2A and 2B (now Figure 4A and 4B) and the plating assay in Figure 2C (now Figure 4C) was done with hydroxyurea in the media/plate. Hence, we state that deletion of either SCD6 or SBP1 shows tolerance to chronic (or continuous) HU stress.

Comment 14: Fig 2F: The rescue by SCD6 OE is not complete, as mentioned in the main text.

Response: We have now included the quantification of the spot assay in 2F (now Figure 4F) to show that the rescue by SCD6 overexpression is complete (Fig S1G).

Comment 15: Figure 2G-H: Please, indicate in the figure what the authors consider 'translated' and 'untranslated' fractions.

Response: The fractions have now been labelled to indicate the missing information in Figure 2G (now Figure 4G).

4. Description of analyses that authors prefer not to carry out

Review 1

Minor Comment 10: Pg. 8/Fig. S3D/4A: It would be interesting to complete the story and determine the functional relationship of Scd6 to the DNL4 mRNA

Response: It is indeed an interesting observation and is currently being pursued as part of another story. We believe it is beyond the scope of the current manuscript.

Revision Plan

Review 3

Major Comment 1: Page 5 and Fig S2E-F: The CLHX experiment to conclude that mRNA is present in Scd6 and Sbp1 puncta is rather indirect. The fact that RNase treatment of a granule-enriched pellet has no effect (Fig S5B) does not help. The authors should perform RNase treatment of intact cells and see that the puncta disappear.

Response: We thank the reviewer for this comment. Cycloheximide treatment is a well-accepted assay to detect the presence of mRNA in granules. Since granules are dynamic structures, and these depend on active translation, CHX treatment leads to the dissociation of Scd6 and Sbp1 granules. This indicates that granule assembly depends on the availability of mRNA derived from translating ribosomes. The observation that Scd6 puncta are sensitive to cycloheximide but not to RNase A treatment is not surprising. It indeed is consistent with the properties of some of the condensates reported in the literature. For example, stress granule cores that are sensitive to cycloheximide, like Scd6 puncta, are resistant to RNase treatment in lysate, indicating that once formed, these structures are quite stable (<https://doi.org/10.1016/j.cell.2015.12.038>). It is interpreted to suggest that the RNAs in these condensates are protected by the RNA-binding proteins. Also, subsequently, in the study, we do RNA immunoprecipitation and granule enrichment experiments and show specific RNA enrichment with Scd6 (Figure 5C, 6A).

Dear Dr. Rajyaguru,

Thanks for your letter asking me to re-consider my decision on your manuscript and for the online meeting presenting the new data that you have obtained in the meantime. Looking again through your manuscript, the referee reports from Review Commons (attached again below), your revision plan, and considering the new data (adding more genotoxic stress inducing agents, further mechanistic insights and functional read-out), I think that a manuscript revised accordingly will address the referee concerns adequately and will also have the advance and broader impact we are looking for.

I thus invite you to revise your manuscript accordingly with the understanding that all concerns must be addressed in the revised manuscript and in a final detailed point-by-point response, as indicated in your revision plan, your letter and in our online meeting.

Acceptance of your manuscript will depend on a positive outcome of another round of review using the same set of referees. It is EMBO reports policy to allow a single round of major revision only and acceptance of the manuscript will therefore depend on the completeness of your responses included in the next, final version of the manuscript.

Revised manuscripts should be submitted within three months of a request for revision. Please contact me to discuss the revision further (also by video chat) if you have questions or comments regarding the revision, or should you need additional time.

- 1) a .docx formatted version of the final manuscript text (including legends for main figures, EV figures and tables), but without the figures included. Figure legends should be compiled at the end of the manuscript text.
- 2) individual production quality figure files as .eps, .tif, .jpg (one file per figure), of main figures (up to 8) and EV figures (up to 5). Please upload these as separate, individual files upon re-submission.

- 3) one final .docx formatted letter INCLUDING the reviewers' reports and your detailed point-by-point responses to their comments. As part of the EMBO Press transparent editorial process, the point-by-point response is part of the Review Process File (RPF), which will be published alongside your paper.

- 4) a complete author checklist, which you can download from our author guidelines (<https://www.embopress.org/page/journal/14693178/authorguide>). Please insert page numbers in the checklist to indicate where the requested information can be found in the manuscript. The completed author checklist will also be part of the RPF.

5) that primary datasets produced in this study (e.g. RNA-seq, CHIP-seq, structural and array data) are deposited in an appropriate public database. If no primary datasets have been deposited, please also state this in a dedicated section (e.g. 'No primary datasets have been generated and deposited'), see below.

The accession numbers and database should be listed in a formal "Data Availability" section (placed after Materials & Methods) that follows the model below. This is now mandatory (like the COI statement). Please note that the Data Availability Section is restricted to new primary data that are part of this study. This section is mandatory. As indicated above, if no primary datasets have been deposited, please state this in this section

Data availability

8) Regarding data quantification and statistics, please make sure that the number "n" for how many independent experiments were performed, their nature (biological versus technical replicates), the bars and error bars (e.g. SEM, SD) and the test used to calculate p-values is indicated in the respective figure legends (also for potential EV figures and all those in the final Appendix). Please also check that all the p-values are explained in the legend, and that these fit to those shown in the figure. Please provide statistical testing where applicable. Please avoid the phrase 'independent experiment', but clearly state if these were biological or technical replicates. Please also indicate (e.g. with n.s.) if testing was performed, but the differences are not significant. In case n=2, please show the data as separate datapoints without error bars and statistics. See also: <http://www.embopress.org/page/journal/14693178/authorguide#statisticalanalysis>

9) Please also note our reference format:

10) We updated our journal's competing interests policy in January 2022 and request authors to consider both actual and perceived competing interests. Please review the policy <https://www.embopress.org/competing-interests> and update your competing interests if necessary. Please name this section 'Disclosure and Competing Interests Statement' and put it after the Acknowledgements section.

11) We now use CRediT to specify the contributions of each author in the journal submission system. CRediT replaces the author contribution section. Please use the free text box to provide more detailed descriptions and do NOT provide an author contributions section in the revised manuscript text file. See also guide to authors: <https://www.embopress.org/page/journal/14693178/authorguide#authorshipguidelines>

12) Please add scale bars of similar style and thickness to microscopic images, using clearly visible black or white bars (depending on the background). Please place these in the lower right corner of the images themselves. Please do not write on or near the bars in the image but define the size in the respective figure legend.

13) Please make sure that all the funding information is also entered into the online submission system and that it is complete and similar to the one in the acknowledgement section of the manuscript text file.

14) All Materials and Methods need to be described in the main text using our 'Structured Methods' format, which is required for all research articles. According to this format, the Materials and Methods section should include a Reagents and Tools Table (listing key reagents, experimental models, software and relevant equipment and including their sources and relevant identifiers), uploaded as separate file, followed by a Methods and Protocols section in which we encourage the authors to describe their methods using a step-by-step protocol format with bullet points, to facilitate the adoption of the methodologies across labs. More information on how to adhere to this format as well as downloadable templates (.doc or .xls) for the Reagents and Tools Table can be found in our author guidelines (section 'Structured Methods'):

15) Please order the manuscript sections like this, using these names:

Title page - Abstract - Keywords - Introduction - Results - Discussion - Methods - Data availability section - Acknowledgements (including the funding information) - Disclosure and Competing Interests Statement - References - Figure legends - Expanded View Figure legends

I look forward to seeing a revised version of your manuscript when it is ready. Please let me know if you have questions or comments regarding the revision.

Best,

Achim Breiling
Senior editor
EMBO reports

Referee #1:

This manuscript demonstrates that RGG containing proteins (Sbp1, Scd6/LSM14A) localize to granules upon treatment with hydroxyurea, Scd6 binds and regulates SRS2 in a manner that is regulated by arginine methylation of the RGG motif, that the RGG and LSM motifs are crucial for the process, and that NHEJ is influenced by LSM14A, suggesting the functional significance of this regulation. Overall the data generally support the conclusions that are drawn. In terms of overall presentation, the manuscript is a bit of a dense read and some of the supplementary figures should be moved to the main body of the manuscript to reflect their importance to the story. I do believe the manuscript will have impact to field, particularly to the specialized RGG protein and stress response niche. I do have several suggestions to polish the manuscript/study:

Major Points:

1. Fig. 1 F/G: were the delta RGG and LSM variants expressed at an equivalent level to the WT protein in these experiments?
2. Fig. 3G: The 6 data points for the delta LSM variant are literally spread evenly up and down the graph, making these data appear highly questionable as to whether one can draw a definitive conclusion from them.

Minor Points:

1. Abstract: the acronym NHEJ likely will need to be defined for the general reader.
2. Introduction, first paragraph: change gene expression to 'transcription' in the phrase 'Even if the contribution of gene expression to GSR..' as I assume this is what is meant here. Gene expression consists of synthesis, processing, translation and decay.

3. Pg. 3 Introduction: Since they are liquid-liquid phase condensates and ribonucleoproteins (RNPs) refers to any protein-RNA interaction, I think that referring to PBs and SGs as mRNPs is a bit misleading (especially the 'major mRNPs').
4. Pg. 3 Introduction: are PBs truly 'sites' of mRNA decay as stated? There are papers in the literature that would argue otherwise.
5. Pg. 3, three lines from bottom. Change LSM14 to LSM14A
6. Pg. 4 top - What is an 'LCS' - containing protein? The acronym has not been defined
7. Fig. S1 - there are a lot of important data in this figure that demonstrate the coordinated movement of Scd6 and Sbp1 to granules. They should be moved into the main body of the manuscript in my opinion. Likewise, a whole section of the Results is dedicated to Fig. S2 - thus I would suggest moving these data into the main body of the manuscript to assist the reader.
8. Fig. 1F should be flipped in the figure with panel G since G is discussed in the results section before F
9. Be sure to define all acronyms for the reader.
10. Pg. 8/Fig. S3D/4A: It would be interesting to complete the story and determine the functional relationship of Scd6 to the DNL4 mRNA
11. Fig. 3H/I: It might be optimal to calculate and compare Kd's for the methylated and unmethylated variants. Also the labels at the top of 3H do not line up with the wells of the EMSA gel.

Significance:

Overall the data generally support the conclusions that are drawn. In terms of overall presentation, the manuscript is a bit of a dense read and some of the supplementary figures should be moved to the main body of the manuscript to reflect their importance to the story. I do believe the manuscript will have impact to field, particularly to the specialized RGG protein and stress response niche

Referee #2:

The authors describe a novel role for the RGG-box containing RNA-binding protein Scd6 and its human ortholog LSM14A in the genotoxic stress to HU treatment. The proteins accumulate in granules upon HU treatment and may be involved in the translational repression of mRNAs coding for relevant factors mediating the HU response. For instance, they show that yeast Scd6 interact with SRS2 mRNA and represses translation during HU response. Interestingly, they also show that interaction with RNAs is dependent on arginine methylation in the LSM domain, which greatly affects interaction with the RNA though specificity needs to be shown. At the end, the authors performed initial analysis with the human ortholog LSM14A, indicating that the cellular function of the protein is evolutionary conserved.

The study is interesting and adds new aspects for understanding RGG-containing proteins and their molecular functions. While it has been previously known that Scd6 accumulates in granules and acts as a translational repressor via eIF4G, this study reveals the regulation of a particular target (SRS2) in context of a relevant stress response. The experiments are usually well performed, though some controls and descriptions need to be added.

Major points:

1. Fig. 2A, B. While there seems to be an effect on the lag phase, it could be revealing if the authors pls. calculate the doubling times for the strains and treatments (taking through the exponential growth phase). Furthermore, it would be good if the authors can show the rescue of phenotypes for deletion strains (ie. reintroduction of respective gene on ARS-CEN based plasmids or (if not available) with the OE plasmids).
2. Fig. 3H. The authors tested the 5'UTR of SRS2 for interaction with recombinant Scd6. Firstly, it is unclear why the authors have chosen the 5'UTR for investigation? Can the authors explain. Secondly, the affinities are relatively low (μM) and the gel shift assay lacks a negative control. The authors should test an unrelated RNA fragment of approximately the same size to control for specificity (negative control). It is unclear whether the protein could interact with any RNA-fragment through charged RNA backbone. Thirdly, it would be good if the authors could show a Coomassie gel for the recombinant protein used in those assays.
3. The Materials and Methods section lacks important information and requires further details to evaluate the study (see below 10 - 17).

Minor points:

Results:

4. The numbering of Figure S1, S2 is confused in the first part of the results section. The authors should check numbering. In general, numbering should follow in the order of the text - pls. check.
5. Pg. 5. CHX treatment leads to a decrease in Scd6-GFP and SBP-1 GFP granules. Essentially, CHX blocks translation elongation so the result indicates that puncta depend on active translation. The authors may want to add this liaison point towards the claim that mRNAs could be present in those puncta. How this results integrates with data shown in Fig. 5S5.
6. Fig. 2H. It would be helpful to the reader, if the authors could mark the respective fraction in the polysomes taken for analysis of relative enrichments. How was this relative enrichment was calculated needs further description.

7. Fig. S5B. 1% SDS treatment cause absence for Scd6 signal from the pellet fraction. Based on this result, I am not clear how based on this result they can claim for presence of higher order mRNA-protein complexes? Why does it exclude the possibility for Scd6 aggregates accumulating in the pellet? The authors need to explain/ modify this statement. Related to earlier findings that showed dependency of puncta upon CHX treatment, one wonders how this result matches to this earlier observation (ie. EDTA should disassemble ribosomes)? Can the authors explain?
8. Fig. 5E, F. For the RNA-seq, the authors compared polysomes with free RNAs (up to 80S) and found enrichment of LIG4 and RTEL1. However, the polysomal profiling mainly shows a slight shift of those mRNAs in higher polysomes; while there is no difference compared to free fractions. How can this be explained?
On the line, the authors should indicate clearly what fractions were pooled for RNA seq analysis. It is also not clear how the authors quantified percentage of RNA in individual fractions (have they spiked-in an RNA?)
- this needs to be stated in the M&M section.
9. At the end, it may be beneficial to the reader if the authors could provide a simple scheme depicting the model developed during this study.
10. Supplemental Data set (.xls)
The adjusted p-values are clustered and >0.05. Can the authors check and describe how those were calculated. How does it match with Volcano plots.

Materials and Methods:

11. A list of primers should be given with specification of their use.
12. The plasmids constructed for (over)expression of proteins/ production of recombinant proteins should be added. If published, references should be added accordingly.
13. RIP: the media for growing yeast cells should be added. Check also other section if defined.
14. RT-qPCR is not sufficiently described. RT kit needs specification, PCR reaction cycles should be given.
15. Quantification of mRNA levels in polysomes is unclear. How was the distribution of mRNA profiles determined? Have the authors added some RNA spikes to fractions?
16. The calculation for the enrichments in IPs is not described conclusively and should be added.
17. Polysomes fractionation (mammalian). It is indicated that the resultant supernatant was adjusted to 5M NaCl and 1 M MgCl₂. This seems to be very high - is this a typo? OR why such high concentrations have been chosen?

Significance:

The study appears solid and well done, except some weaknesses that need to be addressed (see above section). Overall, the interplay with translation could be better investigated and the involvement for interactions with eIF4G/ ribosomes could be better investigated but possibly beyond this study.

Essentially, the study adds a nice mix of experimental approaches to manifest the linkage between granules formation, translation and the physiological implications for genotoxic stress response. However, it is not clear how the chosen conditions reflect any natural conditions (yeast may never be exposed to 100-200 mM HU) and hence, it is unclear how far the observations reflect nature and occur like that. It is certainly a limitation that only one particular stress condition was investigated and it is unclear whether it is also seen with other genotoxic stress inducing agents.

Audience: The study will attract the interest of researchers working with cytoplasmic RBPs, translation and stress granules. The latter topic is currently on a high wave as many researchers jumped on this topic. The study may though remain within this circle as the analysis is rather specialised and constrained on one stress (genotoxic) and wider implications of the findings in a physiological context are unclear.

Referee #3:

In this manuscript, Rajyaguru, Vagner and colleagues address the role of yeast Scd6 in translation regulation during hydroxyurea (HU) stress response. They first show that Scd6 associates to puncta upon HU-treatment, in a manner that depends on the RGG and LSM domains of the protein. They then show that deletion of Scd6 has a mild effect on cell survival under HU-treatment, and over-expression of SCD6 restores the defects observed upon over-expression of SRS2 (a known regulator of the DNA damage response-DDR), suggesting that SRS2 is an SCD6 target, and that SCD6 negatively regulates SRS2. The authors further show that SCD6 binds to SRS2 mRNA and inhibits its translation. Binding to mRNA is somewhat regulated by methylation of the RGG domain, and methylation seems to be controlled by the LSM domain. Finally, the role of SCD6 as a translation regulator during HU-response is conserved, because its mammalian homolog, LSM14A, stimulates the translation of DDR factors under similar conditions. Altogether, this is a nice report showing a function of Scd6/LSM14A in regulation of translation upon HU treatment. However, there are some contradictions that need to be resolved, and the role of puncta in the whole picture is not clear.

Major comments:

1. Page 5 and Fig S2E-F: The CLHX experiment to conclude that mRNA is present in Sdc6 and Sbp1 puncta is rather indirect. The fact that RNase treatment of a granule-enriched pellet has no effect (Fig S5B) does not help. The authors should perform RNase treatment of intact cells and see that the puncta disappear.
2. Fig 2A-F: The effects of Scd6 and Sbp1 deletion upon HU-treatment are very small. A more convincing effect is observed upon over-expression of both SRS2 and SCD6. What is the effect of over-expression of SDC6 and SBP1 alone (i.e. without SRS2 over-expression)?
3. Fig 2E: Why is there an opposite effect of deletion of Scd6 and Sbp1 in the SRS2 over-expression background?
4. Page 7, top: '...indicating that Scd6 regulated the expression of SRS2 in a HU-dependent manner.' In my opinion, the results so far suggest that Scd6 and SRS2 are somehow functionally connected during HU-treatment. To substantiate the statement of the authors, they should provide a Western blot showing that the levels of SRS2 change upon Scd6 KO or OE during HU-treatment. This will also substantiate the results shown in Figs 2G-H.
5. Figs 3: How are the localization of Scd6 protein and SRS2 mRNA to granules, and the levels of SRS2 protein, in cells exposed to HU after deletion of Hmt1? This would substantiate a role of Hmt1 in vivo.
6. Fig 3C: Is the increased interaction of SRS2 mRNA with Scd6 due to increased levels of SRS2 mRNA upon HU treatment? See also comment below.
7. Fig 4A: There seems to be an enrichment of SRS2 mRNA both in the granule-enriched pellet and in the supernatant upon HU treatment in the Scd6-GFP context, suggesting increased SRS2 mRNA levels altogether. The enrichment in granules upon HU is difficult to see, as one should measure the distribution of the mRNA in the pellet relative to the supernatant. Can the authors represent the ratio pellet/supernatant normalized to a control transcript? A similar calculation can be done for the protein normalized to a control protein.
8. Fig 4B: Increased juxtaposition of SRS2 mRNA and Scd6 granules upon HU treatment does not really mean increased colocalization. Granules are likely significantly apart such that increased interactions between the two partners are not explained by increased juxtaposition. Please, comment, tune-down and provide examples where increased granule juxtaposition is associated with increased interaction.
9. Fig 4D: These results are in direct contradiction with those shown in Fig 1C.
10. Fig 5E: Can the authors provide a GO analysis of the up- and down- regulated transcripts?

Minor comments:

11. Figures S1 and S2 seem to be swapped. Please make sure that Figures and panels are arranged in the order they are mentioned in the main text.
12. Page 5, sentence: 'our results argue for the role of Scd6 and Sbp1 in HU-mediated stress response'. I do not agree, as no functional assays showing that these proteins affect HU-mediated stress response have been provided at this point of the story. Please, delete.
13. Page 6: The authors state 'Since scd6 and sbp1 showed tolerance to chronic HU exposure...'. Where is this shown?
14. Fig 2F: The rescue by SCD6 OE is not complete, as mentioned in the main text.
15. Figure 2G-H: Please, indicate in the figure what the authors consider 'translated' and 'untranslated' fractions.

Significance:

The findings provide a new function for Scd6/LSM14A in regulation of translation upon HU treatment. There are limitations regarding the strength of effects in some cases, and the integration of the role of granule formation. These findings are useful for scientists working on genotoxic stress, RNA-binding proteins and/or translation.

Full Revision

Manuscript number: RC-2024-02640, EMBOR-2024-60803V2-Q

Corresponding author(s): Purusharth I, Rajyaguru; Stephan Vagner

1. General Statements

This section is optional. Insert here any general statements you wish to make about the goal of the study or about the reviews.

In the manuscript titled, "Scd6/LSM14A proteins regulate the translation of specific mRNAs to modulate genotoxic stress-induced DNA damage response" we elucidate a conserved role of an RNA-binding protein with low-complexity sequences (RGG-motifs) in response to several genotoxic stresses. This work uncovers HU-stress mediated translation regulation of SRS2, Ligase IV and RTEL1 transcripts by Scd6 (yeast)/LSM14 (human). It further identifies RNP condensates and arginine methylation as sites and means of this regulation.

We heartily thank all three reviewers for their overall encouraging comments about the significance of this manuscript. Specifically, we appreciate their view that the manuscript provides new functional insights into the role of RGG-motif-containing RNA-binding protein in genotoxic stress response. They further agree that such knowledge will impact and interest the general audience of RNA biology and stress biology.

We have carefully noted all the comments raised by three reviewers. In accordance with that, we have added a significant amount of new data and removed some data to make the manuscript less dense and more streamlined. The new results and their analysis have helped us improve the manuscript, allowing us to provide a stronger mechanistic and functional insight underlying the findings presented in this work. Apart from the point-by-point response to each of the comments (mentioned in the later section), we summarize here, the major new results presented in the manuscript based on overall suggestions by the three reviewers and EMBO Reports editor.

- To address the comments about the wider implication of our result and the issue of only using one stress, we have now added Scd6 localization data in response to four new genotoxic stresses such as methyl methanesulphonate (MMS), Cisplatin, Zeocin and UV irradiation (now Figure 1). All the stresses except Zeocin induce the localization of Scd6 to puncta, indicating a general and robust involvement of Scd6 in DNA damage response pathways.
- We have further tested the genetic interaction of SCD6 and SRS2 in the presence of each of the additional four stresses (now Figure 2). Scd6 and SRS2 genetic interaction manifests itself in the presence of HU and MMS stress indicating that other stresses likely involve regulation of transcripts encoding distinct DNA damage proteins by Scd6 in granules.
- To address whether Scd6 puncta were indeed the sites of translation repression, we measured SRS2 mRNA abundance in polysomes. We observed that in cells expressing the Scd6 LSM domain deletion mutant, repression of SRS2 was defective upon HU treatment (now Figure 3G). This is consistent with the observation that in the presence of this mutant, SRS2 mRNA fails to localize to Scd6 puncta (Figure 3F).

- To substantiate the polysome data from cell lines, we provide western blot indicating increased RTEL1 (human homolog of SRS2) protein levels upon LSM14A knockdown in HU stress (now Figure 5G).
- To address the broad physiological implication of Scd6-mediated regulation of specific targets upon genotoxic stress, we provide evidence that localization of Scd6 to granules in HU, MMS and UV irradiation affects NHEJ-mediated DNA repair in yeast, providing a direct link between granule localization and DNA Damage Repair (now Figure 5I). Such regulation is conserved between yeast and humans (Figure 5H) providing strong functional and physiological relevance to the data presented in this manuscript.

Reviewer 1

Overall comments: This manuscript demonstrates that RGG containing proteins (Sbp1, Scd6/LSM14A) localize to granules upon treatment with hydroxyurea, Scd6 binds and regulates SRS2 in a manner that is regulated by arginine methylation of the RGG motif, that the RGG and LSM motifs are crucial for the process, and that NHEJ is influenced by LSM14A, suggesting the functional significance of this regulation. Overall, the data generally support the conclusions that are drawn. In terms of overall presentation, the manuscript is a bit of a dense read and some of the supplementary figures should be moved to the main body of the manuscript to reflect their importance to the story. I do believe the manuscript will have impact to field, particularly to the specialized RGG protein and stress response niche. I do have several suggestions to polish the manuscript/study.

Response: We thank this reviewer for the overall positive comments and constructive inputs. As per suggestion, the paper has been significantly remodeled to make it less dense and more streamlined. Since the study focused more on the role of Scd6 and its mechanism of action, we removed the data associated with Sbp1, which we think has brought clarity to the study.

Major Comment 1: Fig. 1 F/G: were the delta RGG and LSM variants expressed at an equivalent level to the WT protein in these experiments?

Response: We thank the reviewer for this comment. We have quantified the total fluorescence intensity of GFP from the existing microscopy images for WT and domain deletion mutants for Scd6 (Now Figure EV2B-D). This result indicates that the Scd6 Δ RGG and Scd6 Δ LSm protein levels are more or comparable WT.

Major Comment 2: Fig. 3G: The 6 data points for the delta LSM variant are literally spread evenly up and down the graph, making these data appear highly questionable as to whether one can draw a definitive conclusion from them.

Response: We agree with the reviewer that there is some scatter in the data points. To address this, we have performed additional experiments and added new data points to the existing results. Even though there is a spread in the points, except for one data point, all others show an increase in methylation of LSM domain deletion mutant compared to WT, which is statistically significant. The old blot and graph (Old Figures 3F

Full Revision

and 3G) have now been replaced with new ones (Figures 4I and 4J), which look more convincing. The result and conclusion derived from it remain unchanged.

Minor Comments

Comment 1: Abstract: the acronym NHEJ likely will need to be defined for the general reader.

Response: The acronym has been expanded in the abstract and explained in the introduction.

Comment 2: Introduction, first paragraph: change gene expression to 'transcription' in the phrase 'Even if the contribution of gene expression to GSR..' as I assume this is what is meant here. Gene expression consists of synthesis, processing, translation and decay.

Response: The required change has been made.

Comment 3: Pg. 3 Introduction: Since they are liquid-liquid phase condensates and ribonucleoproteins (RNPs) refer to any protein-RNA interaction, I think that referring to PBs and SGs as mRNPs is a bit misleading (especially the 'major mRNPs').

Response: The statement has been rewritten.

Comment 4: Introduction: are PBs truly 'sites' of mRNA decay as stated? There are papers in the literature that would argue otherwise.

Response: The statement has been modified with more citations.

Comment 5: Pg. 3, three lines from bottom. Change LSM14 to LSM14A

Response: The addition has been done.

Comment 6: Pg. 4 top - What is an 'LCS' - containing protein? The acronym has not been defined

Response: The acronym has been defined now. We have also defined acronyms at other places in the manuscript.

Comment 7: Fig. S1 - there are a lot of important data in this figure that demonstrate the coordinated movement of Scd6 and Sbp1 to granules. They should be moved into the main body of the manuscript in my opinion. Likewise, a whole section of the Results is dedicated to Fig. S2 - thus I would suggest moving these data into the main body of the manuscript to assist the reader.

Response: We thank the reviewer for pointing this out. To address this, the data associated with Scd6 in older Figures S1 and S2 are re-distributed to be part of Figures 2F, 2G and Figure EV2B-D.

Full Revision

Considering the several new insightful experimental results (summarised above) that have been added to the manuscript, we think the story is well-rounded and focused on Scd6. To streamline the manuscript, which was already a dense read, we have removed Sbp1-associated results from the manuscript.

Comment 8: Fig. 1F should be flipped in the figure with panel G since G is discussed in the results section before F

Response: Figure 1G is now Figure EV2B and is in the order mentioned in the text. As described above, Figure 1F has been removed from the manuscript.

Comment 9: Be sure to define all acronyms for the reader.

Response: All acronyms in the manuscript have been defined wherever applicable.

Comment 11: Fig. 3H/I: It might be optimal to calculate and compare Kd's for the methylated and unmethylated variants. Also, the labels at the top of 3H do not line up with the wells of the EMSA gel.

Response: We have calculated the Kd's for the EMSA, and it has been added to the results section. We have also aligned the labels at the top of the EMSA gel (now Figure 4K) to match with the wells.

Reviewer 2

Overall comments: The authors describe a novel role for the RGG-box containing RNA-binding protein Scd6 and its human ortholog LSM14A in the genotoxic stress to HU treatment. The proteins accumulate in granules upon HU treatment and may be involved in the translational repression of mRNAs coding for relevant factors mediating the HU response. For instance, they show that yeast Scd6 interacts with *SRS2* mRNA and represses translation during HU response. Interestingly, they also show that interaction with RNAs is dependent on arginine methylation in the LSM domain, which greatly affects interaction with the RNA though specificity needs to be shown. At the end, the authors performed initial analysis with the human ortholog LSM14A, indicating that the cellular function of the protein is evolutionary conserved. The study is interesting and adds new aspects for understanding RGG-containing proteins and their molecular functions. While it has been previously known that Scd6 accumulates in granules and acts as a translational repressor via eIF4G, this study reveals the regulation of a particular target (*SRS2*) in context of a relevant stress response. The experiments are usually well performed, though some controls and descriptions need to be added.

The study appears solid and well done, except some weaknesses that need to be addressed. Overall, the interplay with translation could be better investigated and the involvement for interactions with eIF4G/ ribosomes could be better investigated but possibly beyond this study.

Essentially, the study adds a nice mix of experimental approaches to manifest the linkage between granules formation, translation and the physiological implications for genotoxic stress response. However, it is not clear how the chosen conditions reflect any natural conditions (yeast may never be exposed to 100-200 mM

HU) and hence, it is unclear how far the observations reflect nature and occur like that. It is certainly a limitation that only one particular stress condition was investigated and it is unclear whether it is also seen with other genotoxic stress inducing agents.

Response: We thank the reviewer for the overall positive comments and constructive criticism. We have experimentally addressed the weaknesses of the study that were pointed out in the above paragraph. It has been summarized below followed by a point-wise response:

1. To address the interaction with eIF4G1, we have now provided evidence that indicates that Scd6-eIF4G1 interaction decreases in response to HU stress (Figure 4F), indicating an eIF4G1 independent translation repression of *SRS2* mRNA.
2. With regard to the suitability of using 200 mM HU for yeast experiments, we would like to politely point out that numerous experiments involving the use of 200 mM HU in yeast have been reported in the literature. Some very recent examples are cited here (<https://doi.org/10.15252/emboj.2022113104>, <https://doi.org/10.1038/s44318-024-00161-x>, <https://doi.org/10.1073/pnas.2404470121>, <https://doi.org/10.1038/s44319-023-00055-9>) arguing that such studies with model systems help understand fundamental biological processes such as DNA damage response. Besides, our observations extend to human cell lines where we have used 10 mM HU, indicating that the regulation of DNA damage response mediated by SRS2 uncovered in yeast helped us identify a similar regulation in humans.
3. In the current version of the manuscript, we have tested the localization of Scd6 in response to five different genotoxins: Methyl methanesulphonate (MMS), hydroxyurea (HU), UV irradiation, cisplatin and zeocin (Figure 1). We also uncover the genetic interaction between SCD6 and SRS2 in the above-mentioned stresses and observe the most potent genetic interaction upon HU stress (Figure 2A-B). Moreover, to address the broad physiological implication of Scd6-mediated regulation of specific targets upon genotoxic stress, we provide evidence that localization of Scd6 to granules in genotoxic stress in HU, MMS and UV irradiation affects NHEJ-mediated DNA repair in yeast, providing a direct link between granule localization and DNA Damage Repair (now Figure 5I). Such regulation is conserved between yeast and humans (Figure 5H), providing strong functional and physiological relevance to the data presented in this manuscript.

Major Comment 1: Fig. 2A, B. While there seems to be an effect on the lag phase, it could be revealing if the authors pls. calculate the doubling times for the strains and treatments (taking through the exponential growth phase). Furthermore, it would be good if the authors can show the rescue of phenotypes for deletion strains (ie. reintroduction of respective gene on ARS-CEN based plasmids or (if not available) with the OE plasmids.

Response: We thank the reviewer for this remark. However, based on comments associated with poor strength of the phenotype (Reviewer 3, Major comment 2), we have removed data associated with the older Figure 2A and 2B. The physiological relevance of Scd6-mediated regulation in DNA damage response is now better assessed by the data demonstrating the role of Scd6 in NHEJ activity upon various genotoxic stresses (Figure 5I). This clearly indicates the broad physiological relevance of Scd6 function in genotoxic stress response.

However, to address this reviewer's comment, we have analyzed the effect of complementing the deletion strain with *Scd6* on the CEN plasmid. We observe that Δ *scd6* shows tolerance to HU stress as previously seen, which is rescued almost entirely upon complementation with WT *SCD6*. We could add back the original growth phenotype data and complementation data if needed.

Major Comment 2 (part 1): Fig. 3H. The authors tested the 5'UTR of *SRS2* for interaction with recombinant *Scd6*. Firstly, it is unclear why the authors have chosen the 5'UTR for investigation. Can the authors explain.

Response: We thank the reviewer for this important comment. During experimentation and analysis, we assayed *Scd6* binding to two different fragments of *SRS2* mRNA: 5' and 3'UTR of the same lengths (200 bases). We used the UTR fragments because numerous reports are indicating the role of UTRs in the regulation by RNA binding proteins (<https://doi.org/10.1093/bfpg/els056>, <https://doi.org/10.1126/science.aad9868>, <https://doi.org/10.1093/jxb/erae073>). RNA EMSAs with purified *Scd6* and *in vitro* transcribed UTR RNA fragments revealed a significantly better binding of *Scd6* with the 5' UTR fragment of *SRS2* mRNA compared to the 3' UTR. Therefore, we proceeded with the 5' UTR fragment for further analysis. We have now added this as a supplementary figure panel and explanation in the manuscript text (Figure EV3B and EV3C).

Major Comment 2 (part 2): Secondly, the affinities are relatively low (μ M), and the gel shift assay lacks a negative control. The authors should test an unrelated RNA fragment of approximately the same size to control for specificity (negative control). It is unclear whether the protein could interact with any RNA fragment through a charged RNA backbone.

Response: Our *in vivo* data suggests that the binding of *Scd6* with *SRS2* mRNA is condition and RNA-specific and is regulated by methylation (now Figure 4C, EV3A and 4D). As the reviewer mentioned, *Scd6*, in principle, could bind to any RNA molecule due to the affinity of an RNA-binding protein (with positively charged amino acids such as arginine) to RNA molecule. Nevertheless, the significant difference in the binding of *Scd6* to the 5'UTR and 3'UTR fragments acts as a relative control for EMSA. The aim of the *in vitro* experiment (EMSA) was to establish the difference, if any, in the binding affinities of unmethylated vs methylated *Scd6*, like the *in vivo* data, where we observe significantly increased binding to *SRS2* mRNA upon decreased *Scd6* methylation.

Major Comment 2 (part 3): Thirdly, it would be good if the authors could show a Coomassie gel for the recombinant protein used in those assays.

Response: The Coomassie gel, provided earlier as part of the supplementary data, has now been added to the main figure (Figure 4H) for better clarity.

Major Comment 3: Methods and Materials: The Materials and Methods section lacks important information and requires further details to evaluate the study (see below 11 – 17)

Response: The comment has been duly noted.

Minor Comments

Results:

Comment 4: The numbering of Figure S1, S2 is confused in the first part of the results section. The authors should check numbering. In general, numbering should follow in the order of the text - pls. check.

Response: We have ensured that the numbering follows the order of the text throughout the manuscript.

Comment 5: Pg. 5. CHX treatment leads to a decrease in Scd6-GFP and SBP-1 GFP granules. Essentially, CHX blocks translation elongation, so the result indicates that puncta depends on active translation. The authors may want to add this liaising point towards the claim that mRNAs could be present in those puncta. How this results integrates with data shown in Fig. S5B.

Response: We thank the reviewer for this comment. Since granules are dynamic structures that depend on active translation, CHX treatment leads to the dissociation of Scd6 granules. This indicates that most of the mRNAs present in these granules could be recycled for translation in polysomes. This strategy has been used in multiple research articles for similar deductions ([10.1091/mbc.E08-05-0499](https://doi.org/10.1091/mbc.E08-05-0499), <https://doi.org/10.1083/jcb.151.6.1257>, <https://doi.org/10.1093/nar/gku582>). We have now modified the text in the manuscript to accommodate this point. It has been previously reported that core components of stress granules, once formed, are stable and resistant to RNase, EDTA and NaCl treatment *ex vivo* (<https://doi.org/10.1016/j.cell.2015.12.038>), even though these structures have RNA. Figure S5B (now EV2A) indicates that the granule-enriched fraction indeed behaves like physiological RNA granule cores and not protein aggregates.

Comment 6: Fig. 2H. It would be helpful to the reader, if the authors could mark the respective fraction in the polysomes taken for analysis of relative enrichments. How was this relative enrichment calculated needs further description.

Response: The modification has been made (now Figure 2D) and added to the methods and materials.

Comment 7: Fig. S5B. 1% SDS treatment cause absence for Scd6 signal from the pellet fraction. Based on this result, I am not clear how based on this result they can claim for presence of higher order mRNA-protein complexes? Why does it exclude the possibility for Scd6 aggregates accumulating in the pellet? The authors need to explain/ modify this statement. Related to earlier findings that showed dependency of puncta upon CHX treatment, one wonders how this result matches to this earlier observation (ie.EDTA should disassemble ribosomes)? Can the authors explain?

Response: The very stable β -zipper interactions present in prion-like domains, which leads to aggregation, are resistant to 1-2% SDS treatment (<https://doi.org/10.1016/j.cell.2015.12.038>). Hence, we think that solubilization upon 1% SDS treatment indicates that these are not aggregates. EDTA and NaCl can disrupt interactions, which are stabilized mainly by electrostatic forces. Our observations (now Figure EV2A)

indicate that Scd6 could be part of the more stable mRNP condensate core structure and is, therefore, resistant to these treatments. Such observations have been previously reported, for example, stress granule cores in yeast are not affected by EDTA and NaCl treatments (<https://doi.org/10.1016/j.cell.2015.12.038>).

Comment 8 (part 1): Fig. 5E, F. For the RNA-seq, the authors compared polysomes with free RNAs (up to 80S) and found enrichment of LIG4 and RTEL1. However, the polysomal profiling mainly shows a slight shift of those mRNAs in higher polysomes; while there is no difference compared to free fractions. How can this be explained?

Response: We observed a shift from lower polysome fractions (11-12-13) (not from free fractions) to higher polysome fractions (14-15), indicating an increased number of ribosomes translating the RTEL1 mRNA.

Comment 8 (part 2): On the line, the authors should indicate clearly what fractions were pooled for RNA seq analysis. It is also not clear how the authors quantified percentage of RNA in individual fractions (have they spiked-in an RNA?) - this needs to be stated in the M&M section.

Response: We have added the requested information in the Materials and Methods section. Fractions 13 to 17 were pooled for RNAseq analysis. The % of RNA in each fraction was calculated as described in Panda AC et al. Bio Protoc . 2017 Feb 5;7(3):e2126. doi: [10.21769/BioProtoc.2126](https://doi.org/10.21769/BioProtoc.2126)

Comment 9: At the end, it may be beneficial to the reader if the authors could provide a simple scheme depicting the model developed during this study.

Response: We thank the reviewer for this comment. We have included a model derived from our study as a new figure (Figure 6).

Comment 10: Supplemental Data set (.xls) The adjusted p-values are clustered and >0.05 . Can the authors check and describe how those were calculated. How does it match with Volcano plots.

Response: The adjusted p-values are indeed >0.05 . The p-values (and not the adjusted p-values) are plotted in the Volcano plot (now Fig. 5E)

Materials

and

Methods:

Comment 11: A list of primers should be given with specification of their use.

Response: The list has been added in the supplementary files (Table EV3)

Comment 12: The plasmids constructed for (over)expression of proteins/ production of recombinant proteins should be added. If published, references should be added accordingly.

Response: The list has been added in the supplementary files (Table EV4)

Full Revision

Comment 13: RIP: the media for growing yeast cells should be added. Check also other section if defined.

Response: The information has been added wherever required.

Comment 14: RT-qPCR is not sufficiently described. RT kit needs specification, PCR reaction cycles should be given.

Response: The information has been added

Comment 15: Quantification of mRNA levels in polysomes is unclear. How was the distribution of mRNA profiles determined? Have the authors added some RNA spikes to fractions?
See above.

Response: The % of RNA in each fraction was calculated as described in Panda AC et al. Bio Protoc . 2017 Feb 5;7(3):e2126. doi: [10.21769/BioProtoc.2126](https://doi.org/10.21769/BioProtoc.2126). Details have now been added in the Materials and Methods section.

Comment 16: The calculation for the enrichments in IPs is not described conclusively and should be added.

Response: The calculation has now been elaborated and added to the Materials and Methods section.

Comment 17: Polysomes fractionation (mammalian). It is indicated that the resultant supernatant was adjusted to 5M NaCl and 1 M MgCl₂. This seems to be very high - is this a typo? OR why such high concentrations have been chosen?

Response: The sentence has been removed. There was no need for such an adjustment.

Review 3

Overall comments: In this manuscript, Rajyaguru, Vagner and colleagues address the role of yeast Scd6 in translation regulation during hydroxyurea (HU) stress response. They first show that Scd6 associates to puncta upon HU-treatment, in a manner that depends on the RGG and LSM domains of the protein. They then show that deletion of Scd6 has a mild effect on cell survival under HU-treatment, and over-expression of SCD6 restores the defects observed upon over-expression of SRS2 (a known regulator of the DNA damage response-DDR), suggesting that SRS2 is an SCD6 target, and that SCD6 negatively regulates SRS2. The authors further show that SCD6 binds to SRS2 mRNA and inhibits its translation. Binding to mRNA is somewhat regulated by methylation of the RGG domain, and methylation seems to be controlled by the LSM domain. Finally, the role of SCD6 as a translation regulator during HU-response is conserved, because its mammalian homolog, LSM14A, stimulates the translation of DDR factors under similar conditions. Altogether, this is a nice report showing a function of Scd6/LSM14A in regulation of translation

upon HU treatment. However, there are some contradictions that need to be resolved, and the role of puncta in the whole picture is not clear.

The findings provide a new function for Scd6/LSM14A in regulation of translation upon HU treatment. There are limitations regarding the strength of effects in some cases, and the integration of the role of granule formation. These findings are useful for scientists working on genotoxic stress, RNA-binding proteins and/or translation.

Response: We thank the reviewer for sharing the overall positive comments. We have carefully noted the concern raised by this reviewer about the role of granules in regulation. We believe these comments have greatly improved the manuscript. We have experimentally addressed the concerns, and new results are summarized below followed by a point-wise response.

To address the broad physiological implication of Scd6-mediated regulation of specific targets upon genotoxic stress, we provide evidence that localization of Scd6 to granules upon genotoxic stresses such as HU, MMS and UV irradiation affect NHEJ-mediated DNA repair in yeast, providing a direct link between granule localization and DNA damage repair (now Figure 5I). Such regulation is conserved between yeast and humans (Figure 5H), providing strong functional and physiological relevance to the data presented in this manuscript.

The major concern raised by this reviewer was the lack of clarity about the role of Scd6 granules in *SRS2* mRNA regulation. To address this, we measured *SRS2* mRNA abundance in polysomes. We observed that in cells expressing the Scd6 Lsm domain deletion mutant, repression of *SRS2* translation was defective upon HU treatment (now Figure 3G). This is consistent with the observation that in the presence of this mutant, *SRS2* mRNA fails to localize to Scd6 puncta (Figure 3F). This provides evidence that *SRS2* mRNA localization to Scd6 granules is vital for its repression.

Major Comment 2: Fig 2A-F: The effects of Scd6 and Sbp1 deletion upon HU-treatment are very small. A more convincing effect is observed upon over-expression of both *SRS2* and *SCD6*. What is the effect of over-expression of *SCD6* and *SBP1* alone (i.e. without *SRS2* over-expression)?

Response: We thank the reviewer for this comment. We agree with the concern about the weak phenotype and hence have removed these data from the manuscript. Instead, we have added another functional read-out for the role of Scd6 in DNA damage response (NHEJ activity) in response to various stresses (Figure 5I). This data provides robust evidence supporting the role of Scd6 in DNA damage response. Please also see our response to comment 1 from Reviewer 2.

Major Comment 3: Fig 2E: Why is there an opposite effect of deletion of Scd6 and Sbp1 in the *SRS2* over-expression background?

Response: As explained earlier, considering the several new insightful experimental results (summarised above) added to the manuscript, we think the story is well-rounded and focused on Scd6. To streamline the manuscript, which was already a dense read, as pointed out by another reviewer, we have removed Sbp1-associated results from the manuscript.

Major Comment 4: Page 7, top: '...indicating that Scd6 regulated the expression of SRS2 in a HU-dependent manner.' In my opinion, the results so far suggest that Scd6 and SRS2 are somehow functionally connected during HU-treatment. To substantiate the statement of the authors, they should provide a Western blot showing that the levels of SRS2 change upon Scd6 KO or OE during HU-treatment. This will also substantiate the results shown in Figs 2G-H.

Response: We thank the reviewer for this comment. Detecting SRS2 protein has been technically challenging. The SRS2 construct used in this study is untagged. Unfortunately, the commercial SRS2 antibody has been discontinued. We requested several groups who have used SRS2 antibody in their past studies but they have either closed down their labs or are unable to find an aliquot to share. We have tried tagging SRS2 with 6xHis/1XFLAG/3xFLAG tags at N and C-terminal, but unfortunately, the protein was undetectable in the Western blot analysis using either of the tag-specific antibodies. We have also tried western blot analysis using SRS2-GFP strain, but the protein does not get detected by anti-GFP antibody, probably because of very low expression.

Since we will not be able to provide western blots for Srs2 protein levels due to technical challenges, we have provided western data for RTEL1 (human homolog of Srs2) protein levels upon Lsm14A knockdown in the presence and absence of HU. This validates the polysome data we have of RTEL1 regulation by LSM14A and would, by extension, substantiate the SRS2 polysome data.

Major Comment 6: Fig 3C: Is the increased interaction of SRS2 mRNA with Scd6 due to increased levels of SRS2 mRNA upon HU treatment? See also comment below.

Response: Based on RT-qPCR of total RNA, *SRS2* mRNA levels do not increase, which has now been added as a Supplementary figure (Figure EV2F). Moreover, the quantification of *SRS2* mRNA from the FISH data also does not support increased mRNA levels (Figure 3D, left panel).

Major Comment 7: Fig 4A: There seems to be an enrichment of SRS2 mRNA both in the granule-enriched pellet and in the supernatant upon HU treatment in the Scd6-GFP context, suggesting increased SRS2 mRNA levels altogether. The enrichment in granules upon HU is challenging to see, as one should measure the distribution of the mRNA in the pellet relative to the supernatant. Can the authors represent the ratio pellet/supernatant normalized to a control transcript? A similar calculation can be done for the protein normalized to a control protein.

Response: As mentioned earlier, RT-qPCR data with *SRS2* mRNA levels in total lysate has been added to supplementary data (Figure EV2F). Based on RT-qPCR of total RNA, *SRS2* mRNA levels do not seem to increase.

The quantification of *SRS2* mRNA and Scd6 protein enrichment is done such that the supernatant and pellet fractions are separately normalized to their respective controls (Scd6GFP, untreated sample) and, therefore, do not represent the mRNA distribution but relative mRNA enrichment. However as per the recommendation by the reviewer, the data has been replotted as a ratio of supernatant and pellet with the addition of two more data points and has been added in the main figure (Figure 3F). The overall conclusion remains that the *SRS2* mRNA is enriched in Scd6 granules upon HU treatment.

Major Comment 8: Fig 4B: Increased juxtaposition of SRS2 mRNA and Scd6 granules upon HU treatment does not really mean increased colocalization. Granules are likely significantly apart such that increased interactions between the two partners are not explained by increased juxtaposition. Please comment, tune down and provide examples where increased granule juxtaposition is associated with increased interaction.

Response: The increased interaction between Scd6 protein and SRS2 mRNA upon HU stress has been demonstrated through RNA immunoprecipitation experiment (old Figure 3C, now Figure 4C). We have also provided evidence for co-enrichment of Scd6 protein and SRS2 mRNA in granule enriched fractions (Figure 3F). We believe that the juxtaposition term is leading to misinterpretation of the data, and therefore, we have replaced the term juxtaposition with percentage area overlap to demonstrate that the SRS2 mRNA foci overlap with Scd6GFP foci up to an average of 43.5% in HU stress. This means that the SRS2 mRNA is interacting with Scd6 in the granules. We think that the FISH data adds to the existing observation and all the data together supports our interpretation. Moreover, there are a few examples which indicate increased juxtaposition as an increase in interaction. For example, overlap between stress granules and P-bodies increases interaction and exchange between their components (doi: [10.1016/j.chom.2008.03.004](https://doi.org/10.1016/j.chom.2008.03.004)). A more recent example is the interaction between RNase L-dependent bodies (RLBs) and P bodies, which are observed as juxtaposing or partially overlapping granules ([10.1074/jbc.RA119.011638](https://doi.org/10.1074/jbc.RA119.011638)). However, we have modified the text and the figure (now Figure 3) as recommended by the reviewer along with more quantification.

We would also like to point out that we have provided new data with the LSM domain deletion mutant of Scd6 (Figure 3G), which fails to repress SRS2 mRNA translation. This mutant is defective in directing SRS2 mRNA to Scd6 puncta since it itself does not localize to puncta (Figure 3F). This provides important proof for SRS2 repression in Scd6 granules.

Major Comment 9: Fig 4D: These results are in direct contradiction with those shown in Fig 1C.

Response: We thank the reviewer for this comment. Figure 1C (now part of Figure 1A and 1B) demonstrates that Scd6 localization to puncta, when expressed from a CEN plasmid, significantly increases upon HU stress. The same trend is visible in Figure 4D (now Figure EV2E) where Scd6 is expressed from a 2 μ plasmid; however, it is insignificant. The data in Figure 1C and 4D (now Figure 1A and EV2E respectively) are rather inconsistent with each other than being contradictory. Nevertheless, we understand this reviewer's concern and address it below.

The initial localization experiments used Scd6 expressed from CEN plasmid or genomically tagged Scd6. Since both these versions of Scd6 are not detectable using western blotting (due to the overall low copy number of Scd6 in yeast cells), we used Scd6 expressed from 2 μ plasmid. Localization to condensates by liquid-liquid phase separation is a concentration-driven phenomenon. Therefore, when Scd6 is expressed from a 2 μ plasmid amounting to increased protein levels, its localization to puncta increases even in the absence of stress, which is visible in the quantitation provided in the figure (Figure EV2E) as compared to Figure 1A. We have now analysed the percentage granular localization (granule intensity) of Scd6 (2 μ), which significantly increases upon HU stress (Figure 3D, right panel). Thus although the number of Scd6 granules does not increase upon HU stress when expressed from a 2 μ plasmid, there is a significant increase in localization of Scd6 to granule upon HU stress (Figure 3D, right panel). Also to avoid confusion for the

Full Revision

readers, Figure 4D is now shifted to Figure EV2E, and the Scd6 granule/cell quantification is now replaced with percentage granular localization of Scd6.

Major comment 10: Fig 5E: Can the authors provide a GO analysis of the up- and down-regulated transcripts?

Response: We have now provided a GO analysis (Table EV2). However, due to the low number of regulated genes, only a few GO terms with weak scores appeared in the analysis.

Minor comments:

Comment 11: Figures S1 and S2 seem to be swapped. Please make sure that Figures and panels are arranged in the order they are mentioned in the main text.

Response: We thank the reviewer for pointing it out. Based on suggestions; to streamline the manuscript and make it easier to understand, we have removed Sbp1-associated results from the manuscript. The data associated with Scd6 in older Figures S1 and S2 are part of Figures 2F, 2G and Figure EV2B-D.

Comment 12: Page 5, sentence: 'our results argue for the role of Scd6 and Sbp1 in HU-mediated stress response'. I do not agree, as no functional assays showing that these proteins affect HU-mediated stress response have been provided at this point of the story. Please, delete.

Response: We have removed the sentence from the existing paragraph.

Comment 13: Page 6: The authors state 'Since Dscd6 and Dsbp1 showed tolerance to chronic HU exposure...'. Where is this shown?

Response: The growth curve in older Figure 2A and 2B and the plating assay in older Figure 2C was done with hydroxyurea in the media/plate. Hence, we state that deletion of either SCD6 or SBP1 shows tolerance to chronic (or continuous) HU stress. But this data has now been removed based on comments. See our response to comment 1 from Reviewer 2.

Comment 14: Fig 2F: The rescue by SCD6 OE is not complete, as mentioned in the main text.

Response: We have now included the quantification of the spot assay in presented in the older Figure 2F (now Figure 2C) to show that the rescue by SCD6 overexpression is complete (provided as Fig EV1B).

Comment 15: Figure 2G-H: Please, indicate in the figure what the authors consider 'translated' and 'untranslated' fractions.

Response: The fractions have now been labelled to indicate the missing information in Figure 2G (now Figure 2D).

3. Description of analyses that authors prefer not to carry out

Review 1

Minor Comment 10: Pg. 8/ Fig. S3D/4A: It would be interesting to complete the story and determine the functional relationship of Scd6 to the DNL4 mRNA

Response: It is indeed an interesting observation and is currently being pursued as part of another story. We believe it is beyond the scope of the current manuscript.

Review 3

Major Comment 1: Page 5 and Fig S2E-F: The CLHX experiment to conclude that mRNA is present in Scd6 and Sbp1 puncta is rather indirect. The fact that RNase treatment of a granule-enriched pellet has no effect (Fig S5B) does not help. The authors should perform RNase treatment of intact cells and see that the puncta disappear.

Response: We thank the reviewer for this comment. Cycloheximide treatment is a well-accepted assay to detect the presence of mRNA in granules. Since granules are dynamic structures, and these depend on active translation, CHX treatment leads to the dissociation of Scd6 and Sbp1 granules. This indicates that granule assembly depends on the availability of mRNA derived from translating ribosomes. The observation that Scd6 puncta are sensitive to cycloheximide but not to RNase A treatment is not surprising. It indeed is consistent with the properties of some of the condensates reported in the literature. For example, stress granule cores that are sensitive to cycloheximide, like Scd6 puncta, are resistant to RNase treatment in lysate, indicating that once formed, these structures are quite stable (<https://doi.org/10.1016/j.cell.2015.12.038>). It is interpreted to suggest that the RNAs in these condensates are protected by the RNA-binding proteins. Also, subsequently in the study, we perform smFISH (Figure 3A-B), granule enrichment (Figure 3F), and RNA immunoprecipitation (Figure 4C) which put together demonstrates the presence of mRNA in Scd6 granules.

Full Revision

Major Comment 5: Figs 3: How are the localization of Scd6 protein and *SRS2* mRNA to granules, and the levels of Srs2 protein, in cells exposed to HU after deletion of Hmt1? This would substantiate the role of Hmt1 in vivo.

Response: Hmt1 is a major methyltransferase in yeast with many protein substrates, several of which are RNA-binding proteins. Therefore the readouts associated with Hmt1 deletion can be complicated. Our initial investigations also point in this direction. We think addressing the precise role of arginine mono- and/or di- methylation of Scd6 in genotoxic stress response is an interesting new direction. We hope to pursue this in a future project.

Dear Dr. Rajyaguru,

Thank you for the submission of your revised manuscript to our editorial offices. I have now received the reports from two of the three referees that were asked to re-evaluate the study, you will find below. As you will see, both referees now support the publication of the study in EMBO reports. Referee #3 decline to re-assess the revised manuscript. However, referee #1 also assessed your responses to the points of referee #3 and indicated that the concerns of this referee have been adequately addressed in the revised manuscript. Referee #1 and referee #2 have remaining concerns and suggestions to improve the study, I ask you to address in a final revised manuscript. Please also provide a final p-b-p-response regarding the remaining points of the referee.

Moreover, I have these editorial requests I also ask you to address:

- Please provide a more compact title with not more than 100 characters (including spaces).
- Please provide an abstract with not more than 175 words, written in present tense.
- Please reduce the number of keywords to five and order the manuscript sections like this, using these names:
Title page - Abstract - Keywords - Introduction - Results - Discussion - Methods - Data availability section - Acknowledgements (including the funding information) - Disclosure and Competing Interests Statement - References - Figure legends - Expanded View Figure legends
- Thus, please add the funding information to the Acknowledgements.
- Please provide individual production quality figure files as .eps, .tif, .jpg (one file per figure), of main figures and EV figures. Please upload these as separate, individual files upon re-submission.
- Please remove now the referee access information from the Data Availability section and make sure the dataset is public latest upon online publication of the paper. Moreover, please provide a direct link for the dataset.
- Please use our reference format:
<http://www.embopress.org/page/journal/14693178/authorguide#referencesformat>
- Please check again that the number "n" for how many independent experiments were performed, their nature (biological versus technical replicates), the bars and error bars (e.g. SEM, SD) and the test used to calculate p-values is indicated in the respective figure legends. Please also check that all the p-values are explained in the legend, and that these fit to those shown in the figure. Please provide statistical testing where applicable. Please avoid the phrase 'independent experiment', but clearly state if these were biological or technical replicates. Please also indicate (e.g. with n.s.) if testing was performed, but the differences are not significant. In case n=2, please show the data as separate datapoints without error bars and statistics. See also:
<http://www.embopress.org/page/journal/14693178/authorguide#statisticalanalysis>
- If n<5, please show single datapoints for diagrams. Moreover:
 - Please note that the exact p values are not provided in the legends of figures 5H, EV2 C, E
 - Please indicate the statistical test used for data analysis in the legends of figures 1B, C; 3F, 5E
 - Please indicate what */ **/ ***/ **** represents; if this represents p value(s), please indicate the exact p value in the legend(s) of figure(s) 1B, C; 2E, G; 3B, D, F, G; 4C, E, G, J, L; 5C, G, H, I; EV1 A, B; EV2 C, D; EV3 A, C.
 - Please indicate what */ **/ ***/ **** represents; if this represents p value(s), please indicate the statistical test used and where appropriate, and the exact p value in the legend(s) of figure(s) EV2 D
 - Please note that information related to n is missing in the legends of figures 3D, 5E, EV2 D, F; EV3 A, C
 - Please note that the error bars are not defined in the legends of figures 5F, EV2 D, EV3 C
 - Please note that the white arrows are not defined in the legend of figure 1A, 2F. This needs to be rectified.
 - Please note that the dotted borders are not defined in the legend of figure 3A. This needs to be rectified.
- Table EV1 and Table EV2 are datasets and need to be named and uploaded as Dataset EV1 and Dataset EV2. Please also update the callouts for these items accordingly. Please add legends for these as a separate TAB in the respective excel sheets.
- Please move tables EV3, EV4, EV5 and EV6 to the 'Reagents and Tools Table'. Please update their callouts (see 'Reagents and Tools Table') and make sure that the tools table is called out where appropriate in the Methods section.
- Please remove the section/legends of the EV tables and Table EV6 from the manuscript text file.
- Thank you for providing the source data. Please upload the SD as one folder per figure, grouping together separate excel files

for all panels for one figure (and ZIPed together).

In addition, I would need from you uploaded separately:

Best,

Referee #1:

The authors modified the abstract, introduction and other parts of the manuscript. The manuscript comes along more focused and they added new data supporting the study. Importantly, they expanded the scope by showing accumulation of Scd6 in cytoplasmic puncta upon various genotoxic stress treatments, while in the original version only application of hydroxyurea (HU) was tested. They also add other data (e.g. synthetic screen) and removed some figures. Overall, the seen effects are often not very profound which led them to remove some weaker parts (figures) - this is fine but would not have been necessary to be fair. The M&M section has been revised accordingly and a final figure summarising the results has been added.

While most points were adequately addressed, there is still some confusion regarding RNA seq and the quantification of RNA in polysomes.

Comment 8: Although the authors somewhat clarified the point somewhat in their response, I could not find a clear description of the fractions taken for consideration in the M&M section. It still seems that they used free RNA up to 80S ribosomes vs. heavy polysomes for RNA seq in the M&M section while they mention to have used fractions 13-17 for polysomes in their response to referees. Hence, they should declare what their understanding of heavy polysomes is - fraction 13-17? Accordingly, fractions between 80S and heavy polysomes have not been considered, right? Hence, the authors profiled free RNA vs. heavy polysomes but the rationale for comparing those is not explained and should be added.

Comment 8 (part 2) & comment 15. The authors should add the mentioned reference used for quantification of RNA in polysomes to the manuscript (Panda et al). Here, I am not clear why the authors used 1 µg for RNA from each polysomal fraction for cDNA synthesis? They should rather have used same fraction (e.g. 0.5 mL) as the amount of RNA varies across polysomal fractions and would screw-up quantification (and according to Panda et al). As such, this is all still confusing and the authors should clarify this point to remove doubts on this quantification.

Referee #2:

The authors have adequately addressed the concerns raised in the original round of peer review through Review Commons. I find the manuscript to be improved and convincing. There is one minor typo that I noted:

Minor Point:

1. Abstract: polysome profiling should be two words (not polysomeprofiling)

Rev_Com_number: RC-2024-02640
New_manu_number: EMBOR-2024-60803V3
Corr_author: Rajyaguru
Title: Scd6/LSM14A proteins regulate specific mRNA to modulate genotoxic stress-induced DNA damage response

Reviewers' Comments: Point-by-point response

We would like to thank both Reviewer 1 and Reviewer 2 for reviewing the manuscript again and supporting its publication. The comments provided had greatly improved the manuscript leading to a positive outcome.

Following are the p-b-p responses for the latest set of comments.

Reviewer 1

The authors modified the abstract, introduction and other parts of the manuscript. The manuscript comes along more focused and they added new data supporting the study. Importantly, they expanded the scope by showing accumulation of Scd6 in cytoplasmic puncta upon various genotoxic stress treatments, while in the original version only application of hydroxyurea (HU) was tested. They also add other data (e.g. synthetic screen) and removed some figures. Overall, the seen effects are often not very profound which led them to remove some weaker parts (figures) - this is fine but would not have been necessary to be fair. The M&M section has been revised accordingly and a final figure summarising the results has been added.

While most points were adequately addressed, there is still some confusion regarding RNA seq and the quantification of RNA in polysomes.

Comment 8: Although the authors somewhat clarified the point somewhat in their response, I could not find a clear description of the fractions taken for consideration in the M&M section. It still seems that they used free RNA up to 80S ribosomes vs. heavy polysomes for RNA seq in the M&M section while they mention to have used fractions 13 -17 for polysomes in their response to referees. Hence, they should declare what their understanding of heavy polysomes is - fraction 13-17? Accordingly, fractions between 80S and heavy polysomes have not been considered, right? Hence, the authors profiled free RNA vs. heavy polysomes but the rationale for comparing those is not explained and should be added.

Comment 8 (part 2) & comment 15. The authors should add the mentioned reference used for quantification of RNA in polysomes to the manuscript (Panda et al). Here, I am not clear why the authors used 1 µg for RNA from each polysomal fraction for cDNA synthesis? They should rather have used same fraction (e.g. 0.5 mL) as the amount of RNA varies across polysomal fractions and would screw-up quantification (and according to Panda et al). As such, this is all still confusing and the authors should clarify this point to remove doubts on this quantification.

Response

We have now corrected the M&M section by adding the following statements:

- heavy polysome fractions (13 to 17)
- The read counts on RNAs extracted from polysome fractions (13-17) depend on both mRNA abundance and their translation rates. Differential translation can be characterized by the differences between the changes in total mRNA levels and the levels of mRNAs engaged in ribosomes (heavy polysome fractions).
- The % of RNA in each fraction was calculated as described in Panda AC et al. Bio Protoc . 2017 Feb 5;7(3):e2126. doi: [10.21769/BioProtoc.2126](https://doi.org/10.21769/BioProtoc.2126)

- RNA from the same volume (300 μ l) of each polysome fraction (17 fractions) was isolated.....

Reviewer 2

The authors have adequately addressed the concerns raised in the original round of peer review through Review Commons. I find the manuscript to be improved and convincing. There is one minor typo that I noted:

Minor comment:

1. Abstract: polysome profiling should be two words (not polysomeprofiling)

Response: The correction has been made.

Dr. Purusharth Rajyaguru
Indian Institute of Science
Department of Biochemistry
Lab FE#16, Department of Biochemistry,
New biological sciences building, Indian Institute of science
Bangalore, Karnataka 560012
India

Dear Dr. Rajyaguru,

I am very pleased to accept your manuscript for publication in the next available issue of EMBO reports. Thank you for your contribution to our journal.

Yours sincerely,
